# SALIENT IMAGENET: HOW TO DISCOVER SPURIOUS FEATURES IN DEEP LEARNING?

**Sahil Singla & Soheil Feizi**
University of Maryland, College Park
{ssingla,sfeizi}@umd.edu

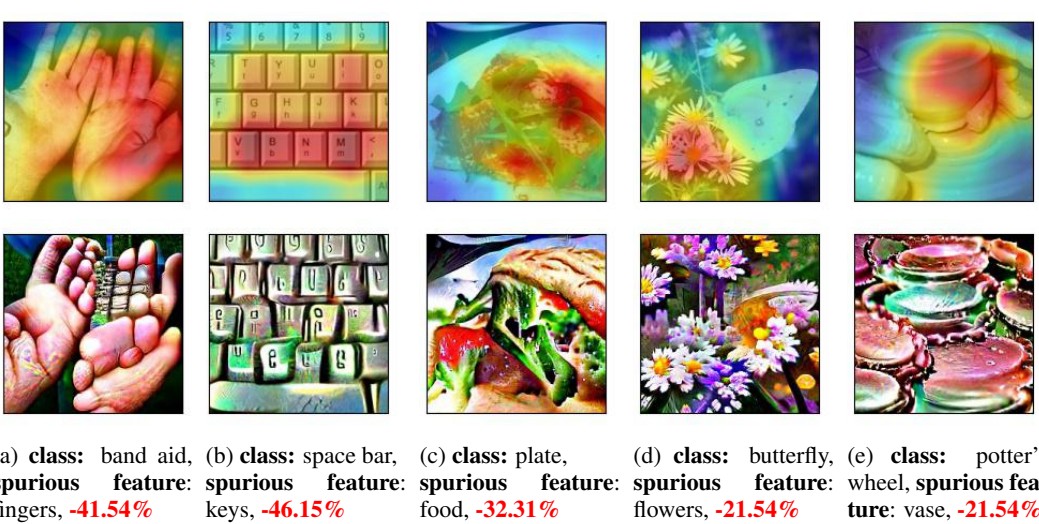

(a) **class:** band aid, **spurious feature**: fingers, **-41.54%**  (b) **class:** space bar, **spurious feature**: keys, **-46.15%**  (c) **class:** plate, **spurious feature**: food, **-32.31%**  (d) **class:** butterfly, **spurious feature**: flowers, **-21.54%**  (e) **class:** potter's wheel, **spurious feature**: vase, **-21.54%**

Figure 1: Examples of spurious features discovered using our framework for a Standard Resnet-50 model. In the top row, red color highlights regions with spurious features. The bottom row shows images generated by visually amplifying these features. Adding a small amount of gaussian noise (with $\sigma = 0.25$) to spurious regions (red regions) significantly reduces the model accuracy (shown in dark red in captions) for a subset of 65 images with the shown class labels.

## ABSTRACT

Deep neural networks can be unreliable in the real world especially when they heavily use *spurious* features for their predictions. Focusing on image classifications, we define *core features* as the set of visual features that are always a part of the object definition while *spurious features* are the ones that are likely to *co-occur* with the object but not a part of it (e.g., attribute "fingers" for class "band aid"). Traditional methods for discovering spurious features either require extensive human annotations (thus, not scalable), or are useful on specific models. In this work, we introduce a *general* framework to discover a subset of spurious and core visual features used in inferences of a general model and localize them on a large number of images with minimal human supervision. Our methodology is based on this key idea: to identify spurious or core *visual features* used in model predictions, we identify spurious or core *neural features* (penultimate layer neurons of a robust model) via limited human supervision (e.g., using top 5 activating images per feature). We then show that these neural feature annotations *generalize* extremely well to many more images *without* any human supervision. We use the activation maps for these neural features as the soft masks to highlight spurious or core visual features. Using this methodology, we introduce the *Salient Imagenet* dataset containing core and spurious masks for a large set of samples from Imagenet. Using this dataset, we show that several popular Imagenet models rely heavily on various spurious features in their predictions, indicating the standard accuracy alone is not sufficient to fully assess model perfor-

mance. Code and dataset for reproducing all experiments in the paper is available at `https://github.com/singlasahil14/salient_imagenet`.

# 1 INTRODUCTION

The growing use of deep learning systems in sensitive applications such as medicine, autonomous driving, law enforcement and finance raises concerns about their trustworthiness and reliability in the real world. A key reason for the lack of reliability of deep models is their reliance on spurious input features (i.e., features that are not essential to the true label) in their inferences. For example, a convolutional neural network (CNN) trained to classify camels from cows associates green pastures with cows and fails to classify pictures of cows in sandy beaches correctly since most of the images of cows had green pastures in the training set (Beery et al., 2018; Arjovsky et al., 2020). Similarly, Bissoto et al. (2020) discovered that skin-lesion detectors use spurious visual artifacts for making predictions. The list of such examples goes on (de Haan et al., 2019; Singla et al., 2021).

Most of the prior work on discovering spurious features (Nushi et al., 2018; Zhang et al., 2018; Chung et al., 2019) requires humans to first come up with a possible list of spurious features. This is often followed by an expensive human-guided labeling of visual attributes which is not scalable for datasets with a large number of classes and samples such as ImageNet. In some cases, this may even require the construction of new datasets with unusual visual attributes to validate the hypothesis (such as cows in sandy beaches discussed previously). To address these limitations, recent works (Singla et al., 2021; Wong et al., 2021) use the neurons of robust models as visual attribute detectors thereby circumventing the need for manual annotations. However, their applicability is either limited to robust models which achieve low clean accuracy or to models that achieve low accuracy on the training dataset. We discuss limitations of these methods in more detail in Section 2.

In this work, we introduce a *general* methodology to discover a subset of spurious and core (non-spurious) visual attributes used in model inferences and localize them on a large number of images with minimal human supervision. Our work builds upon the prior observations (Engstrom et al., 2019) that for a robust model, neurons in the penultimate layer (called *neural features*) often correspond to human-interpretable visual attributes. These visual attributes can be inferred by visualizing the *heatmaps* (e.g. via CAM, Zhou et al. (2016)) that highlight the importance of image pixels for those neural features (top row in Figure 1). Alternatively, one can amplify these visual attributes in images by maximizing the corresponding neural feature values via a procedure called the *feature attack* (Engstrom et al. (2019); Singla et al. (2021), bottom row in Figure 1).

Our framework (shown in Figure 2) is based on this key idea: to identify spurious or core *visual attributes* used for predicting the class $i$, we identify spurious or core *neural features* via limited human supervision. We define a core visual attribute as an attribute for class $i$ that is always a part of the object definition and spurious otherwise. To annotate a neural feature as core or spurious, we show only the top-5 images (with a predicted label of $i$) that maximally activate that neural feature to Mechanical Turk workers. We then show that these neural feature annotations *generalize* extremely well to top-$k$ (with $k \gg 5$) images with label $i$ that maximally activate that neural feature. For example, on Imagenet, in Section 3.4, we show a generalization accuracy of $95\%$ for $k = 65$ (i.e., a 13 fold generalization). Thus, by using neural features and their corresponding neural activation maps (heatmaps), with *limited* human supervision, we can identify a large subset of samples that contain the relevant spurious or core visual attributes. The neural activation maps for these images can then be used as soft segmentation masks for these visual attributes. We emphasize that the usual method of obtaining such a set would require the manual detection and segmentation of a visual attribute across all images in the training set and is therefore not scalable.

We apply our proposed methodology to the Imagenet dataset: we conducted a Mechanical Turk study using 232 classes of Imagenet and 5 neural features per class. For each neural feature, we obtained its annotation as either core (non-spurious) or spurious for the label. Out of the $232 \times 5 = 1,160$ neural features, 160 were deemed to be spurious by the workers. For 93 classes, at least one spurious neural feature was discovered. Next, for each (class=$i$, feature=$j$) pair, we obtained 65 images with class $i$ showing highest activations of the neural feature $j$ and computed their neural activation maps. The union of these images is what we call the *Salient Imagenet* dataset. The dataset contains $52,521$ images with around 226 images per class on average. Each instance in the dataset is of the form $(\mathbf{x}, y, \mathcal{M}^c, \mathcal{M}^s)$ where $y$ is the ground truth class and $\mathcal{M}^s, \mathcal{M}^c$ represent the set of

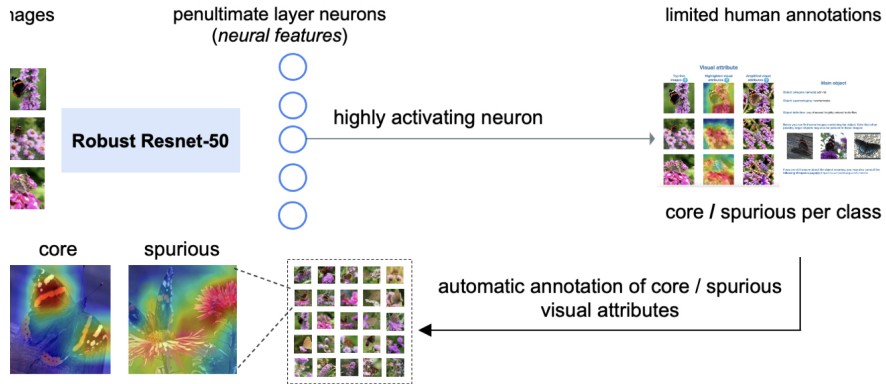

Figure 2: Our framework for constructing the *Salient Imagenet* dataset. We annotate *neural features* as core (non-spurious) or spurious using minimal human supervision. We next show that these annotations *generalize* to many more new samples. We highlight core and spurious regions on the input samples using activation maps of these neural features.

spurious and core masks, respectively (obtained from neural activation maps of neural features). This dataset can be used to test the sensitivity of *any pretrained model* to different image features by corrupting images using their relevant masks and observing the drop in the accuracy.

Ideally, we expect our trained models to show a low drop in accuracy for corruptions in spurious regions and a high drop for that of the core regions. However, using a standard Resnet-50 model, we discover multiple classes for which the model shows a trend contradictory to our expectations: a significantly higher *spurious drop* compared to the *core drop*. As an example, for the class drake (Fig. 5), we discover that corrupting any of spurious features results in a significantly higher drop ($> 40\%$) compared to corrupting any of core features ($< 5\%$). We show many more examples in Appendix L. For various standard models (Resnet-50, Wide-Resnet-50-2, Efficientnet-b4, Efficientnet-b7), we evaluate their accuracy drops due to corruptions in spurious or core regions at varying degrees of corruption noise and find no significant differences between their spurious or core drops suggesting that none of the models significantly differentiates between core and spurious visual attributes in their predictions (Fig. 6). In summary, our methodology has the following steps:

- **Step 1:** We select neural features as penultimate-layer neurons of a pretrained robust network and visualize them using a few highly activating samples (Section 3.2).
- **Step 2:** Using Mechanical Turk, we annotate neural features as core or spurious (Section 3.3).
- **Step 3:** Using neural feature annotations, we automatically annotate core and spurious visual attributes on many more samples without any human supervision (Section 3.3).
- **Step 4:** Using another Mechanical Turk study, we show that our methodology generalizes extremely well to identify core/spurious visual attributes on new samples (Section 3.4).
- **Step 5:** Applying steps 1-4 to Imagenet, we develop a new dataset called *Salient Imagenet* whose samples, in addition to the class label, are annotated by core and/or spurious masks (Section 4).
- **Step 6:** Using such a richly annotated dataset, we assess *core/spurious accuracy* of standard models by adding small noise to spurious/core visual attributes, respectively (Section 5).

To the best of our knowledge, this is the first work that introduces a general methodology to discover core and spurious features with limited human supervision by leveraging neurons in the penultimate layer of a robust model as interpretable visual feature detectors. Also, this is the first work that releases a version of Imagenet with annotated spurious and core image features. We believe that this work opens future research directions to leverage richly annotated datasets to develop deep models that mainly rely on core and meaningful features in their inferences.

## 2 RELATED WORK

**Robustness and Interpretability**: In recent years, there have been many efforts towards post-hoc interpretability techniques for trained neural networks. Most of these efforts have focused on *local*

*explanations* where decisions about single images are inspected (Zeiler & Fergus, 2014; Yosinski et al., 2016; Dosovitskiy & Brox, 2016; Mahendran & Vedaldi, 2016; Nguyen et al., 2016; Zhou et al., 2018; Olah et al., 2018; Adebayo et al., 2018; Chang et al., 2019; Carter et al., 2019; Yeh et al., 2019; Sturmfels et al., 2020; O'Shaughnessy et al., 2019; Verma et al., 2020). These include saliency maps (Simonyan et al., 2014; Smilkov et al., 2017; Sundararajan et al., 2017; Singla et al., 2019), class activation maps (Zhou et al., 2016; Selvaraju et al., 2019; Bau et al., 2020; Ismail et al., 2019; 2020), surrogate models to interpret local decision boundaries such as LIME (Ribeiro et al., 2016), methods to maximize neural activation values by optimizing input images (Nguyen et al., 2015; Mahendran & Vedaldi, 2015) and finding influential inputs (Koh & Liang, 2017). However, recent work suggests that class activation or saliency maps may not be sufficient to narrow down visual attributes and methods for maximizing neural activations often produce noisy visualizations (Olah et al., 2017; 2018) with changes imperceptible to humans. To address these limitations, recent works (Tsipras et al., 2018; Engstrom et al., 2019) show that for robust (Madry et al., 2018) or adversarially trained models (in contrast to standard models), activation maps are qualitatively more interpretable and optimizing an image directly to maximize a certain neuron produces human perceptible changes in the images which can be useful for visualizing the learned features.

**Failure explanation**: Most of the prior works on failure explanation either operate on tabular data where interpretable features are readily available (Zhang et al., 2018; Chung et al., 2019), language data where domain-specific textual queries can be easily generated (Wu et al., 2019; Ribeiro et al., 2020), images where visual attributes can be collected using crowdsourcing (Nushi et al., 2018; Xiao et al., 2021; Plumb et al., 2021) or photorealistic simulation (Leclerc et al., 2021). However, collecting visual attributes via crowdsourcing can be expensive for large datasets. Moreover, these methods require humans to hypothesize about the possible failures and one could miss critical failure modes when the visual attributes used by model are different from the ones hypothesized by humans.

To address these limitations, recent works (Wong et al., 2021; Singla et al., 2021) use the neurons of robust models as the visual attribute detectors for discovering failure modes, thus avoiding the need for crowdsourcing. However, Wong et al. (2021) cannot be used to analyze the failures of standard (non-robust) models which achieve significantly higher accuracy and are more widely deployed in practice. Moreover, to discover the spurious feature, they ask MTurk workers to identify the common visual attribute across multiple images without highlighting any region of interest. However, different images may have multiple common attributes even if they come from different classes and this approach may not be useful in such cases. Barlow (Singla et al., 2021) learns decision trees using misclassified instances to discover leaf nodes with high failure concentration. However, the instances could be correctly classified due to some spurious feature (e.g., background) and Barlow will discover no failure modes in such cases. In contrast, our approach circumvents these limitations and discovers the spurious features for standard models even when they achieve high accuracy.

## 3 A GENERAL FRAMEWORK FOR DISCOVERING SPURIOUS FEATURES

Consider an image classification problem $\mathcal{X} \to \mathcal{Y}$ where the goal is to predict the ground truth label $y \in \mathcal{Y}$ for inputs $\mathbf{x} \in \mathcal{X}$. For each class $i \in \mathcal{Y}$, we want to identify and localize a set of *core or spurious visual attributes* in the training set that can be used by neural networks to predict $i$:

- **Core Attributes** are the set of visual features that are always a part of the object definition.
- **Spurious Attributes** are the ones that are likely to *co-occur* with the object, but not a part of it (such as food, vase, flowers in Figure 1).

A more formal discussion of these abstract definitions can be found in Appendix G. Note that localizing these visual attributes on a large number of images requires significant human annotations and can be very expensive. In this paper, we propose a general methodology to tackle this issue.

### 3.1 NOTATION AND DEFINITIONS

For a trained neural network, the set of activations in the penultimate layer (adjacent to logits) is what we call the *neural feature vector*. Each element of this vector is called a *neural feature*. For an input image and a neural feature, we can obtain the *Neural Activation Map* or *NAM*, similar to CAM (Zhou et al., 2016), that provides a soft segmentation mask for highly activating pixels (for the neural feature) in the image (details in Appendix A.1). The corresponding *heatmap* can then be

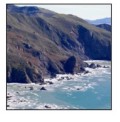 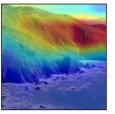 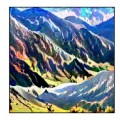 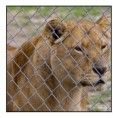 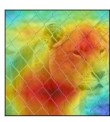 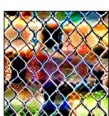

| Image | Heatmap | Feature Attack | Image | Heatmap | Feature Attack |

Figure 3: On the left, the heatmap suffices to explain that the focus is on ground. But on the right, the heatmap covers both cougar and wires. Feature attack clarifies that the focus is on wires.

obtained by overlaying the NAM on top of the image so that the red region highlights the highly activating pixels (Appendix A.2). The *feature attack* (Singla et al., 2021) is generated by optimizing the input image to increase the values of the desired neural feature (Appendix G).

## 3.2 EXTRACTING, VISUALIZING AND SELECTING NEURAL FEATURES

Prior works (Engstrom et al., 2019; Singla et al., 2021) provide evidence that neural features of a robust model encode human interpretable visual attributes. Building on these works, to identify spurious or core visual attributes used for predicting class $i$, we identify neural features encoding these attributes. For each image in the Imagenet training set (Deng et al., 2009), we extract the neural feature vector using a robust Resnet-50 model. The robust model was adversarially trained using the $l_2$ threat model of radius 3. Each neural feature is visualized using the following techniques:

- **Most activating images**: By selecting 5 images that maximally activate the neural feature, the common visual pattern in these images can be interpreted as the visual attribute encoded in it.
- **Heatmaps (NAMs)**: When there are multiple common visual attributes, heatmaps highlight the region of interest in images and can be useful to disambiguate; e.g., in Figure 3 (left panels), heatmap disambiguates that the focus of the neural feature is on the ground (not water or sky).
- **Feature attack**: In some cases, heatmaps may highlight multiple visual attributes. In such cases, the feature attack can be useful to disambiguate; e.g., in Figure 3 (right panels), the feature attack indicates that the focus of the neural feature is on the wires (not cougar).

However, the neural feature vector can have a large size (2048 for the Resnet-50 used in this work) and visualizing all of these features per class to determine whether they are core or spurious can be difficult. Thus, we select a small subset of these features that are highly predictive of class $i$ and annotate them as core or spurious for $i$. Prior work (Singla et al., 2021) selects this subset by considering the top-$k$ neural features with the highest mutual information (MI) between the neural feature and the *model failure* (a binary variable that is 1 if the model misclassifies $\mathbf{x}$ and 0 otherwise). However, the model could be classifying all the images correctly using some spurious features. In this case, the MI value between the (spurious) neural feature and the model failure will be small and thus this approach will fail to discover such critical failure modes.

In this work, we select a subset of neural features *without* using the model failure variable. To do so, we first select a subset of images (from the training set) on which the robust model predicts the class $i$. We compute the mean of neural feature vectors across all images in this subset denoted by $\bar{\mathbf{r}}(i)$. From the weight matrix $\mathbf{w}$ of the last linear layer of the robust model, we extract the $i^{th}$ row $\mathbf{w}_{i,:}$ that maps the neural feature vector to the logit for the class $i$. Next, we compute the hadamard product $\bar{\mathbf{r}}(i) \odot \mathbf{w}_{i,:}$. Intuitively, the $j^{th}$ element of this vector $(\bar{\mathbf{r}}(i) \odot \mathbf{w}_{i,:})_j$ captures the mean contribution of neural feature $j$ for predicting the class $i$. This procedure leads to the following definition:

**Definition 1** *The Neural Feature Importance of feature $j$ for class $i$ is defined as:* $\mathrm{IV}_{i,j} = (\bar{\mathbf{r}}(i) \odot \mathbf{w}_{i,:})_j$. *For class $i$, the neural feature with $k^{th}$ highest IV is said to have the feature rank $k$.*

We thus select the neural features with the highest-5 importance values (defined above) per class.

## 3.3 MECHANICAL TURK STUDY FOR DISCOVERING SPURIOUS AND CORE FEATURES

To determine whether a neural feature $j$ is core or spurious for class $i$, we conducted a crowd study using Mechanical Turk (MTurk). We show the MTurk workers two panels: One panel visualizes the

feature $j$ and the other describes the class $i$. To visualize $j$, we show the 5 most activating images (with predicted class $i$), their heatmaps highlighting the important pixels for $j$, and the feature attack visualizations (Engstrom et al., 2019). For the class $i$, we show the object names (Miller, 1995), object supercategory (from Tsipras et al. (2020)), object definition and the wikipedia links. We also show 3 images of the object from the Imagenet validation set. We then ask the workers to determine whether the visual attribute is a part of the main object, some separate objects or the background. They were also required to provide reasons for their answers and rate their confidence on a likert scale from 1 to 5. The design for this study is shown in Figure 9 in Appendix C. Each of these tasks (also called Human Intelligence Tasks or HITs) were evaluated by 5 workers. The HITs for which the majority of workers selected either separate objects or background as their responses were deemed to be spurious and the ones with main object as the majority answer were deemed to be core. We note that our proposed approach might not be effective for discovering *non-spatial* spurious signals (such as gender/race) that are not visualizable using heatmaps/feature attacks.

Because conducting a study for all 1000 Imagenet classes can be expensive, we selected a smaller subset of 232 classes (denoted by $\mathcal{T}$) by taking a union of 50 classes with the highest and 50 with the lowest accuracy across multiple models (details in Appendix B). For each class, we selected 5 neural features with the highest feature importance values resulting in $232 \times 5 = 1160$ HITs. For 160 HITs, workers deemed the feature to be spurious for the label. Thus, for each class $i \in \mathcal{T}$, we obtain a set of core and spurious neural features denoted by $\mathcal{C}(i)$ and $\mathcal{S}(i)$, respectively.

## 3.4 GENERALIZATION OF VISUAL ATTRIBUTES ENCODED IN NEURAL FEATURES

We show that the visual attribute inferred for some neural feature $j$ (via visualizing the top-5 images with predicted class $i$ as discussed in the previous section) *generalizes* extremely well to top-$k$ (with $k \gg 5$) images with the ground truth label $i$ that maximally activate feature $j$. That is, we can obtain a large set of images containing the visual attribute (encoded in neural feature $j$) by only inspecting a small set of images. To obtain the desired set, denoted by $\mathcal{D}(i, j)$, we first select training images with label $i$ (Imagenet contains $\approx 1300$ images per label). Next, we select $k = 65$ images (5% of 1300) from this set with the highest activations of the neural feature $j$. We use the Neural Activation Maps (NAMs) for these images as soft segmentation masks to highlight the desired visual attributes.

To validate that the NAMs focus on the desired visual attributes, we conducted another Mechanical Turk study. From the set $\mathcal{D}(i, j)$, we selected 5 images with the highest and 5 with the lowest activations of feature $j$. We show the workers two panels: The first panel shows images with the highest 5 activations (along with their heatmaps) and the second shows images with the lowest 5 activations (and heatmaps). We then ask them to determine if the focus of heatmaps in both visualizations was on the same visual attribute, different attributes or if either of the two visualizations was unclear. For each spurious feature (160 total), we obtained answers from 5 workers each. For 152 visualizations (95%), majority of the workers selected *same* as the answer, thereby validating that the NAMs focus on the same visual attribute. The design for the study is shown in Figure 10 in Appendix D.

We emphasize that although using $k = 65$ significantly increases the size of the set (i.e., a 13 fold increase over the manually annotated set), the value of $k$ can be changed depending on the number of training images with the desired properties. One can quickly search over various possible values of $k$ by visualizing the 5 images (and their heatmaps) with the highest and lowest activations of the relevant neural feature (in the set of $k$ images) until they both focus on the desired visual attribute.

## 4 THE SALIENT IMAGENET DATASET

Recall that for each class $i \in \mathcal{T}$, we obtain a set of core and spurious neural features denoted by $\mathcal{C}(i)$ and $\mathcal{S}(i)$, respectively (Section 3.3). Then, using the generalization property (Section 3.4), we obtain the set $\mathcal{D}(i, j)$ for each $i \in \mathcal{T}$ and $j \in \mathcal{C}(i) \cup \mathcal{S}(i)$. Each $\mathcal{D}(i, j)$ contains 65 images along with the NAM (per image) acting as the soft segmentation mask for the visual attribute encoded in feature $j$. We note these soft segmentation masks overlap with the desired visual attribute and may not cover it completely. However, since we corrupt these regions using Gaussian noise which *preserves the content* of the original images (discussed in Section 5), they can be useful for evaluating the model sensitivity to various spurious/core visual attributes. If $j \in \mathcal{C}(i)$, these are called *core masks*, otherwise *spurious masks*. The union of all these datasets $\mathcal{D}(i, j)$ is what we call the *Salient*

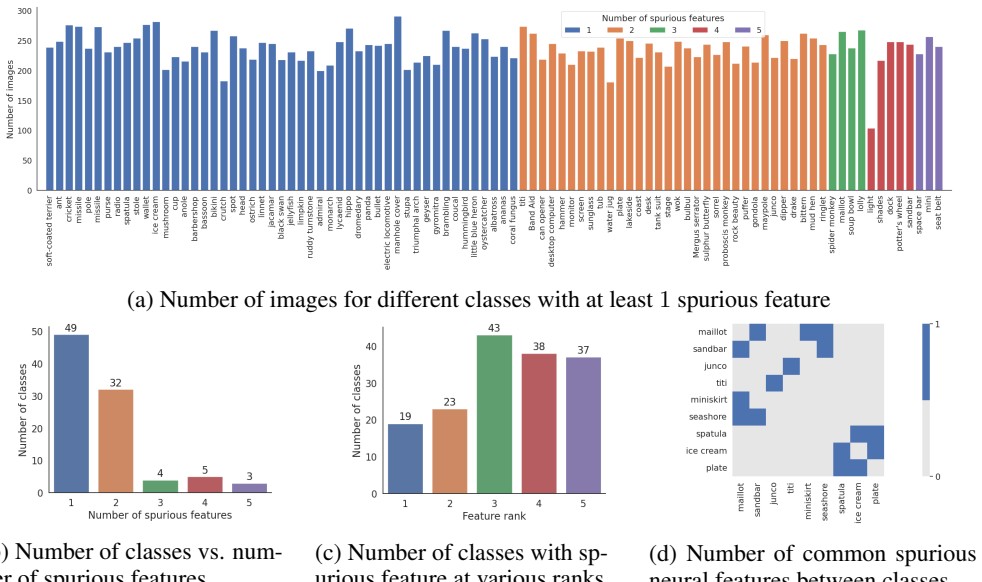

(a) Number of images for different classes with at least 1 spurious feature

(b) Number of classes vs. number of spurious features

(c) Number of classes with spurious feature at various ranks

(d) Number of common spurious neural features between classes

Figure 4: In the top row, we show number of images for different classes with at least 1 spurious feature in the Salient Imagenet dataset. Bottom row shows various plots using the dataset.

*Imagenet* dataset. Each instance in the dataset is of the form $(\mathbf{x}, y, \mathcal{M}^c, \mathcal{M}^s)$ where $y$ is the ground truth label while $\mathcal{M}^s/\mathcal{M}^c$ represent the set of spurious/core masks for the image $\mathbf{x}$. The dataset and anonymized Mechanical Turk study results are also available at the associated github repository.

The dataset contains $52,521$ images with around $226$ images per class on average. From the MTurk study (Section 3.3), $160$ features were discovered to be spurious and $1,000$ to be core for relevant classes. We visualize several spurious features ($59$ total) in Appendix J. Examples of background and foreground spurious features are in Appendix J.1 and J.2, respectively. For 93 classes, we discover at least one spurious feature (the number of images per class shown in Fig. 4a). We show a histogram of the number of classes vs. the number of spurious features in Fig. 4b. For 3 classes (namely space bar, miniskirt and seatbelt) all 5 features were found to be spurious. This finding suggests that in safety critical applications, it can be unsafe to just use the test accuracy to assess the model performance because the model may be highly accurate on the benchmark for *all the wrong reasons*. We plot the histogram of the number of classes with spurious feature at rank $k$ (from 1 to 5) in Fig. 4c. We discover that across all feature ranks, a significant number of spurious features are discovered (minimum 19) and a larger number of spurious features are discovered at higher feature ranks (4 and 5). This suggests that inspecting a larger number of features per class (i.e., $> 5$) may be necessary for a more thorough analysis of spurious features. In Fig. 4d, we show the number of shared spurious features between different classes (diagonal is zeroed out for brevity). By visualizing common neural features, we diagnose that tree branches are a confusing visual attribute between titi and junco (Appendix K.1); food between plate and icecream (Appendix K.2).

## 5 EVALUATING DEEP MODELS USING THE SALIENT IMAGENET DATASET

In this section, we use the Salient Imagenet dataset to evaluate the performance of several pretrained models on Imagenet. In particular, each set $\mathcal{D}(i, j)$ can be used to test the sensitivity of *any trained model* to the visual attribute encoded with feature $j$ for predicting the class $i$. We can either blackout (fill with black color) or corrupt (using some noise model) the region containing the attribute (using the soft segmentation masks) for all images in this set and evaluate the accuracy of the model on these corrupted images. However, because the existing Imagenet pretrained models have not explicitly been trained on images with black regions, these images can be characterized as out of distribution of the training set. Thus, it may be difficult to ascertain whether changes in accuracy are due to the distribution shift or the removal of spurious features. Instead, we choose to corrupt the region of interest using Gaussian noise to preserve the content of the original images (which can be useful

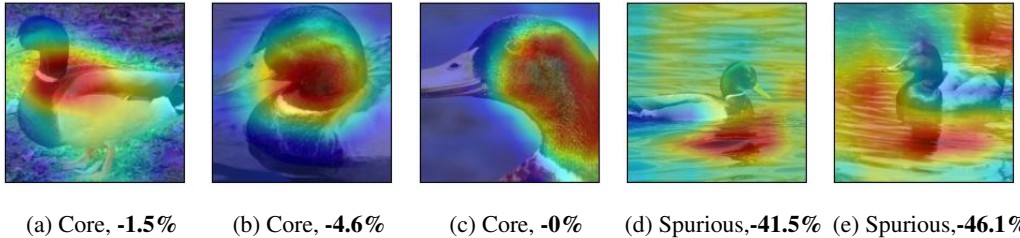

(a) Core, **-1.5%**     (b) Core, **-4.6%**     (c) Core, **-0%**     (d) Spurious,**-41.5%**  (e) Spurious,**-46.1%**

Figure 5: Each image denotes the heatmap for different neural features for the class drake. Using a standard Resnet-50 model, (clean accuracy of 95.4%), we observe a drop of *at least* $41.5\%$ by adding gaussian noise with $\sigma = 0.25$ to spurious masks while a drop of *at most* $4.6\%$ for core masks. This shows that the model heavily relies on spurious features in its predictions for this class.

when the soft segmentation masks are inaccurate). Such Gaussian noise is also known to occur in the real world due to sensor or electronic circuit noise. Since the segmentation masks we derive are soft (i.e., their values lie between 0 and 1, not binary), we use these masks to control the degree of the Gaussian noise corruption across the original image. That is, given image $\mathbf{x}$ and mask $\mathbf{m}$, we add noise $\mathbf{z} \sim \mathcal{N}(\mathbf{0}, \mathbf{I})$ using the equation below (examples in Appendix I):

$$\mathbf{x} + \sigma\left(\mathbf{z} \odot \mathbf{m}\right), \tag{1}$$

where $\sigma$ is a hyperparameter that can be used to control the degree of corruption in the image. The above equation ensures that regions where the mask $\mathbf{m}$ has values close to 1 are corrupted by noise with the desired standard deviation $\sigma$ and the ones with values close to 0 suffer little changes.

## 5.1 CORE AND SPURIOUS ACCURACY

We now introduce concepts of *core* and *spurious* accuracy to evaluate deep models. Informally, the core accuracy is the accuracy of the model only due to the core features of the images and the spurious accuracy is due to the spurious regions. To compute the core accuracy for class $i$, we first obtain the sets $\mathcal{D}(i, j)$. For some image $\mathbf{x} \in \mathcal{D}(i, j)$, let $\mathbf{m}^{(j)}$ be the mask obtained for some feature $j$. Next, we take the union of all these sets for $j \in \mathcal{S}(i)$ denoted by $\mathcal{DS}(i)$. Note that for some images in the union, we may have multiple spurious masks but to evaluate the core accuracy, we want to obtain a single mask per image that covers all the spurious attributes. Thus, for the image $\mathbf{x}$, the $(p, q)$-th element of the single spurious mask $\mathbf{s}$ (or, the single core mask $\mathbf{c}$), is computed by taking the element-wise max over all the relevant masks:

$$\mathbf{s}_{p,q} = \max_{j \in \mathcal{S}(i),\, \mathbf{x} \in \mathcal{D}(i,j)} \mathbf{m}^{(j)}_{p,q}, \qquad \mathbf{c}_{p,q} = \max_{j \in \mathcal{C}(i),\, \mathbf{x} \in \mathcal{D}(i,j)} \mathbf{m}^{(j)}_{p,q} \tag{2}$$

**Definition 2** *(Core Accuracy) We define the Core Accuracy or* $\mathrm{acc}^{(\mathcal{C})}$ *as follows:*

$$\mathrm{acc}^{(\mathcal{C})} = \frac{1}{|\mathcal{T}|} \sum_{i \in \mathcal{T}} \mathrm{acc}^{(\mathcal{C})}(i), \quad \mathrm{acc}^{(\mathcal{C})}(i) = \frac{1}{|\mathcal{DS}(i)|} \sum_{\mathbf{x} \in \mathcal{DS}(i)} \mathbb{1}\left(h\left(\mathbf{x} + \sigma\left(\mathbf{z} \odot \mathbf{s}\right)\right) = y\right)$$

We acknowledge that our definition of core accuracy is *incomplete* because the set of spurious visual attributes discovered using our framework may not cover *all* spurious attributes in the dataset. Finally, note that the defined core accuracy is a function of the noise parameter ($\sigma$) used to corrupt spurious regions. The Spurious Accuracy or $\mathrm{acc}^{(\mathcal{S})}(i)$ can be computed similarly by replacing $\mathcal{DS}(i)$ with $\mathcal{DC}(i) = \cup_{j \in \mathcal{C}(i)} \mathcal{D}(i, j)$, and $\mathbf{s}$ with $\mathbf{c}$ in equation 2. Observe that $\mathcal{DS}(i) \neq \mathcal{DC}(i)$ in general because $\mathcal{D}(i, j)$ may contain different sets of images for different $j$'s. Thus, the standard accuracy of the model (i.e. without adding any noise) on the two sets pf $\mathcal{DS}(i)$ and $\mathcal{DC}(i)$ can be different. We want our trained models to show a low degradation in performance when noise is added to spurious regions (i.e. high core accuracy) and a high degradation when corrupting core regions (i.e. low spurious accuracy).

## 5.2 RESULTS FROM EVALUATING DEEP MODELS ON THE SALIENT IMAGENET DATASET

In this section, we test the sensitivity of several models to various spurious/core features.

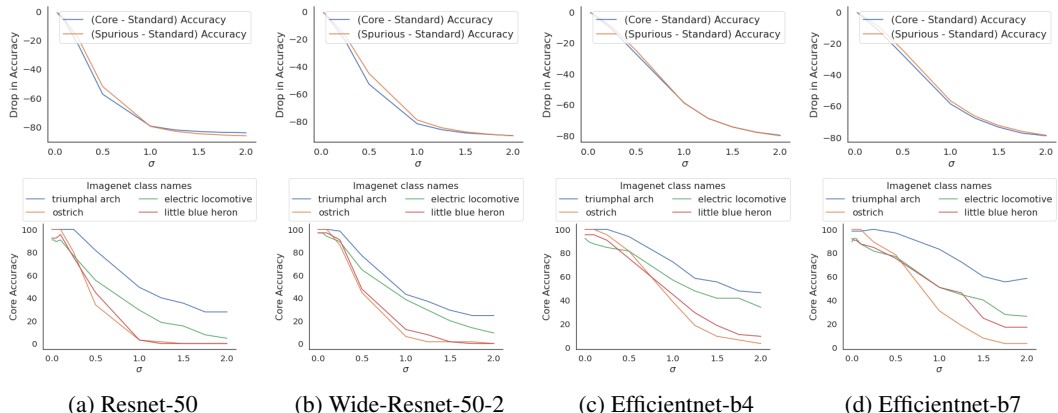

|              |                   |                 |                  |
| :----------: | :---------------: | :-------------: | :--------------: |
| (a) Resnet-50 | (b) Wide-Resnet-50-2 | (c) Efficientnet-b4 | (d) Efficientnet-b7 |

Figure 6: In the top row, we observe a similar drop in accuracy irrespective of whether noise is added to spurious or core regions suggesting that trained models do not differentiate between the spurious and core regions for predicting the desired class. In the bottom row, we show core accuracy vs. noise parameter $\sigma$ for several classes. In particular, for both classes "triumphal arch" and "ostrich", the model has a standard accuracy of $100\%$ (at $\sigma = 0$). However, the core accuracy for triumphal arch is high even at $\sigma = 2.0$ ($\approx 40\%$) while that for ostrich is almost $0\%$.

**Testing model sensitivity to spurious features**. We test the sensitivity of a standard Resnet-50 model to different spurious and core features (one feature at a time) by corrupting images from the datasets $\mathcal{D}(i, j)$ and evaluating the drop in model accuracy. We use $\sigma = 0.25$ (equation 1) because we observe that it preserves the content of the images (Examples in Appendix I) so we would expect the model prediction to remain unchanged. In Figure 1, we show multiple examples of spurious features discovered using this procedure. In Figure 5, we show a class (namely drake) on which the model has a high clean accuracy ($95.4\%$). However, it exhibits a large drop in performance when any of the spurious regions (e.g. water) are corrupted while a small drop is observed when any of the core regions (e.g. bird's head) are corrupted. This highlights that **the model relies on mostly spurious features in its predictions** for this class. Out of the 93 classes with at least one spurious feature, we discovered 34 classes such that the accuracy drop due to some spurious feature was at least $20\%$ higher than due to some core feature. In Appendix L, we show visualizations of core and spurious features for 15 of these classes. On 11 of these 15 classes, the model has $> 95\%$ accuracy. Since the $l_2$ norm of noise perturbations can be different for different features (e.g. due to different $l_2$ norms of their activation maps), we also evaluate accuracy drops due to corrupting spurious/core features when the mean $l_2$ perturbations are similar and find similar trends (details in Appendix L).

**Comparing core and spurious accuracy of trained models**. Next, we compute the core and spurious accuracy of four standard (i.e. not adversarially trained) pretrained models namely, Resnet-50 (He et al., 2015), Wide Resnet-50-2 (Zagoruyko & Komodakis, 2016), Efficientnet-b4 and Efficientnet-b7 (Tan & Le, 2019). In Figure 6 (top row), we plot the drop in accuracy when noise is added to spurious regions (i.e., core accuracy - standard accuracy) and also to core regions (i.e., spurious accuracy - standard accuracy). We observe a similar drop in performance for all values of $\sigma$ and for all trained models. This suggests that trained models do not differentiate between spurious and core regions of the images for predicting an object class. In some sense, this is expected because in the standard Empirical Risk Minimization (ERM) paradigm, models have never explicitly been trained to make such differentiation, suggesting that providing additional supervision during training (for example, segmenting core/spurious regions as provided in Salient Imagenet or diversifying the training dataset) may be essential to train models that achieve high core accuracy. In Figure 6 (bottom row), we plot the core accuracy for 4 different classes as the noise level $\sigma$ increases. In particular, consider the two classes "triumphal arch" and "ostrich" that both have standard accuracy of $100\%$ (at $\sigma = 0$). However, the core accuracy for triumphal arch is high even at $\sigma = 2.0$ ($\approx 40\%$) while that for ostrich is almost $0\%$. This provides further evidence that **the standard accuracy alone is not a reliable measure for model performance in the real world** because the core accuracy for two classes with the same standard accuracy can be very different. We believe having richly annotated training datasets such as Salient Imagenet can lead to training reliable deep models that mainly rely on core and informative features in their predictions.

## ACKNOWLEDGEMENTS

This project was supported in part by NSF CAREER AWARD 1942230, a grant from NIST 60NANB20D134, HR001119S0026-GARD-FP-052, ONR YIP award N00014-22-1-2271, Army Grant W911NF2120076 and AWS Machine Learning Research Award. We would like to thank Besmira Nushi, Eric Horvitz and Shital Shah for helpful discussions.

## REPRODUCIBILITY

We provide the design details of our Mechanical Turk studies in Appendix Sections C and D. The code for user studies, experiments and the dataset is available at `https://github.com/singlasahil14/salient_imagenet`.

## ETHICS STATEMENT

This paper introduces a scalable framework for discovering spurious features in predictions made by deep neural networks. Our approach can be useful for various downstream tasks such as model debugging and developing improved models that rely on core features in their predictions. We introduce a new dataset called Salient ImageNet that includes annotations of various core and spurious visual attributes for a subset of images in the ImageNet training set. We do *not* collect any new samples for constructing our dataset. As this work paves the way to develop reliable models, we do not see any foreseeable negative consequences associated with our work.

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

# Appendix

## A  VISUALIZING THE NEURAL FEATURES OF A ROBUST MODEL

### A.1  NEURAL ACTIVATION MAP (NAM)

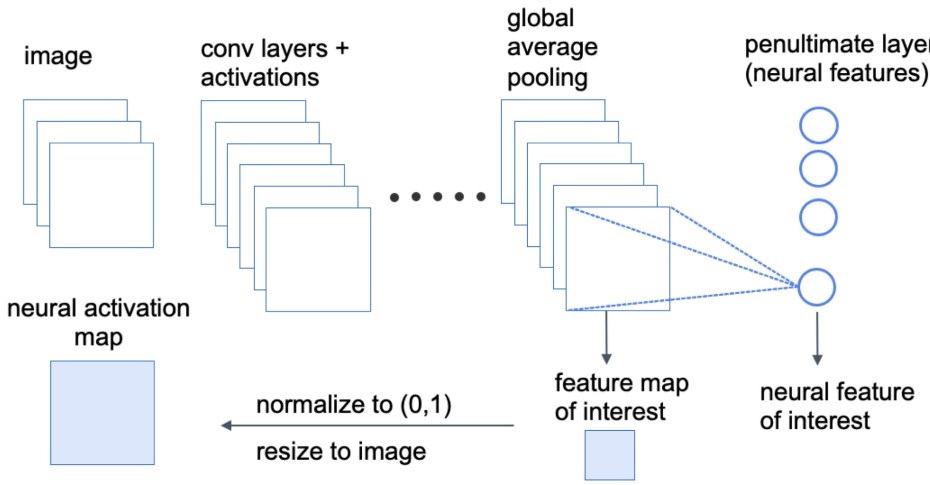

Figure 7: Neural activation map generation

Figure 7 describes the Neural Activation Map generation procedure. To obtain the neural activation map for feature $j$, we select the feature map from the output of the tensor of the previous layer (i.e the layer before the global average pooling operation). Next, we simply normalize the feature map between 0 and 1 and resize the feature map to match the image size, giving the neural activation map.

### A.2  HEATMAP

Heatmap can be generated by first converting the neural activation map (which is grayscale) to an RGB image (using the jet colormap). This is followed by overlaying the jet colormap on top of the original image using the following few lines of code:

```
import cv2

def compute_heatmap(img, fam):
    hm = cv2.applyColorMap(np.uint8(255 * nam),
        cv2.COLORMAP_JET)
    hm = np.float32(hm) / 255
    hm = hm + img
    hm = hm / np.max(hm)
    return hm
```

### A.3  FEATURE ATTACK

In Figure 8, we illustrate the procedure for the feature attack. We select the feature we are interested in and simply optimize the image to maximize its value to generate the visualization. $\rho$ is a hyperparameter used to control the amount of change allowed in the image. For optimization, we use gradient ascent with step size $= 40$, number of iterations $= 25$ and $\rho = 500$.

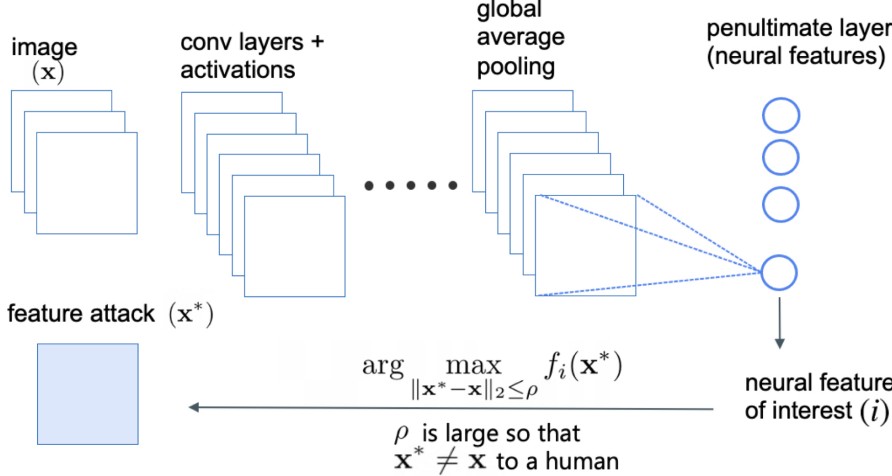

Figure 8: Feature attack generation

## B    SELECTING THE CLASSES FOR DISCOVERING SPURIOUS FEATURES

Because conducting a Mechanical Turk (MTurk) study for discovering the spurious features for all 1000 classes of Imagenet can be expensive, we selected a smaller subset of classes as follows. Using some pretrained neural network $h$, for each class $i$ in Imagenet, we obtain groups of images with the label $i$ (called label grouping) and prediction $i$ (i.e $h$ predicts $i$, called prediction grouping) giving 2000 groups. For each group, we compute their accuracy using the network $h$. For each grouping (label/prediction), we selected 50 classes with the highest and 50 with the lowest accuracy giving 100 classes per grouping and take the union. We used two pretrained neural networks: standard and robust Resnet-50 resulting in total 232 classes.

## C    MECHANICAL TURK STUDY FOR DISCOVERING SPURIOUS FEATURES

The design for the Mechanical Turk study is shown in Figure 9. We showed the workers two panels side by side. The left panel visualizes a neuron (from a robust resnet-50 model trained on Imagenet) and the right panel describes an object class from Imagenet.

The left panel visualizing the neuron is shown in Figure 9a. To visualize the neuron, we first select the subset of images for which the robust model predicts the object class (given on the right). We show three sections: Top five images (five images with highest activation values of the neuron in the subset), Highlighted visual attributes (heatmaps computed using the neural activation maps as described in Section A.2) and Amplified visual attributes (feature attack computed by optimizing the top five images to maximize the neuron as described in Section A.3).

The right panel describing the object class is shown in Figure 9b. We show the object category name (also called synset) from the wordnet heirarchy (Miller, 1995), object supercategory (from Tsipras et al. (2020)), object definition (also called gloss) and the relevant wikipedia links describing the object. We also show 3 images of the object from the Imagenet validation set.

The questionnaire is shown in Figure 9c. We ask the workers to determine whether they think the visual attribute (given on the left) is a part of the main object (given on the right), some separate object or the background of the main object. We also ask the workers to provide reasons for their answers and rate their confidence on a likert scale from 1 to 5. The visualizations for which majority of workers selected either separate object or background as the answer were deemed to be spurious. In total, we had 205 unique workers, each completing 5.66 tasks (on average). Workers were paid $0.1 per HIT, with an average salary of $8 per hour.

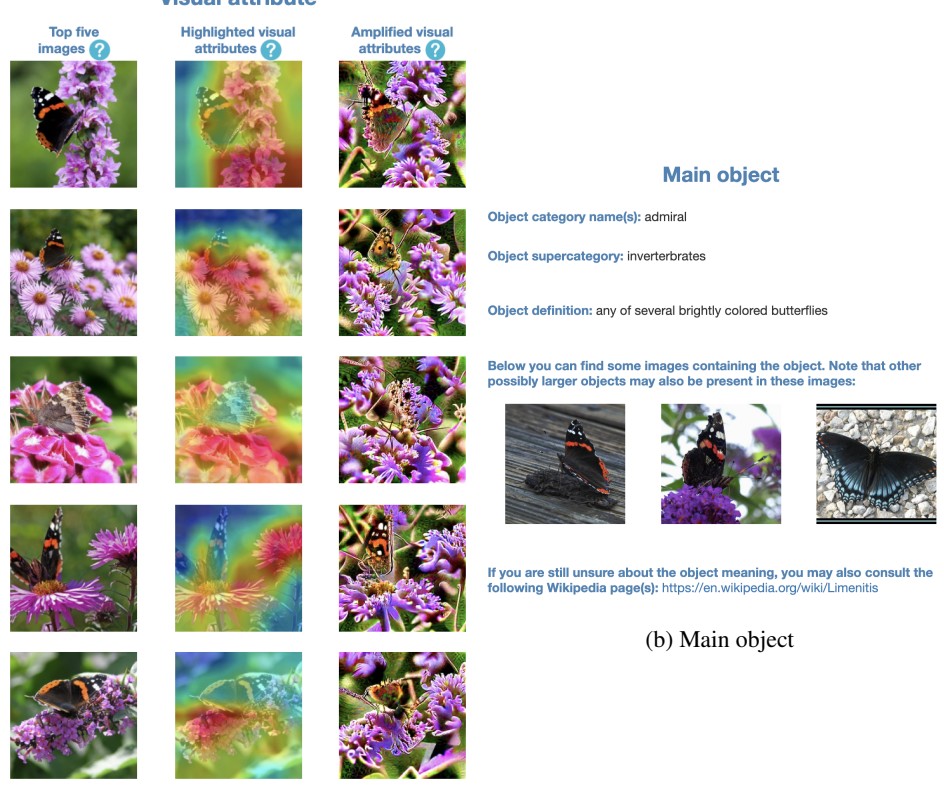

(a) Visual attribute

(b) Main object

**Q1.** Which of the following best describes the focus of the **red region in the Highlighted visual attributes**?

- Amplified visual attributes can be useful in some cases when the **red region overlaps over multiple objects** such as in these examples: overlap between cougar and wires, overlap between groom and flowers.
- You can ignore Amplified visual attributes when the **red region does not overlap over multiple objects.**
- The **main object** has the following **name(s)**: admiral
- The **main object** belongs to the following **supercategory**: invertebrates

○ Focus is on the **background of the main object.**
**Examples:** greenery in the background of grasshopper, water in the background of albatross, flower in the background of bee, baseball pitch in the background of player.

○ Focus is on some **other object (not a part of the main object).**
**Examples:** baby is not a part of bassinet (baby's bed), legs are not a part of maillot (dress for dancers), tree branch is not a part of bird, graffiti is not a part of freight car, flowers are not a part of groom, wires are not a part of cougar.

○ Focus is on **some part of the main object.**
**Examples:** nose of a cougar, whiskers of a cat, legs of a lobster, structure of a steel bridge.

**Q2.** How confident are you about your answer to Q1?

| (1) Not confident at all | (2) Slightly confident | (3) Somewhat confident | (4) Fairly confident | (5) Completely confident |

**Q3.** Please give reason(s) for your answer to Q1? This question is mandatory for completing this HIT. Detailed answers will receive a bonus.

(c) Questionnaire

Figure 9: Mechanical Turk study for discovering spurious features

## C.1 QUALITY CONTROL

Only allowed workers with at least $95\%$ approval rates and minimum of $1000$ HITs were allowed to complete our tasks. Additionally, short and generic answers to Q3 were rejected.

## D MECHANICAL TURK STUDY FOR VALIDATING HEATMAPS

The design for the Mechanical Turk study is shown in Figure 10. Given a subset of the training set (constructed as described in Section 3.4), our goal is to validate whether the heatmaps focus on the same visual attribute for all images in the subset.

Recall that this subset was constructed for each spurious feature discovered using the previous crowd study (Section C). For each spurious feature, we already know both the object class and the neuron index of the robust model. We construct the subset of training set by first selecting the subset of images for which the *label is the object class*. Note that this is different from the previous crowd study where the prediction of the robust model is the object class. Next, we select the top-65 images from this subset where the desired neuron has the highest activation values.

We showed the workers two panels side by side, each panel visualizing the same neuron. The left panel visualizing the neuron is shown in Figure 10a. The visualization shows the images with the highest-5 neural activations in the subset and corresponding heatmaps. The right panel visualizing the neuron is shown in Figure 10b. That shows the images with the lowest-5 neural activations in the subset and corresponding heatmaps.

The questionnaire is shown in Figure 10c. We ask the workers to determine whether they think the focus of the heatmap is on the same object (in both the left and right panels), different objects or whether they think the visualization in either of the sections is unclear . They were given the option to choose from these four options: same, different, Section A is unclear and Section B is unclear. Same as in the previous study (Section C), we ask the workers to provide reasons for their answers and rate their confidence on a likert scale from $1$ to $5$.

The visualizations for which majority of workers selected same as the answer were deemed to be validated i.e for this subset of $65$ images, we assume that the neural activation maps focus on the same visual attribute.

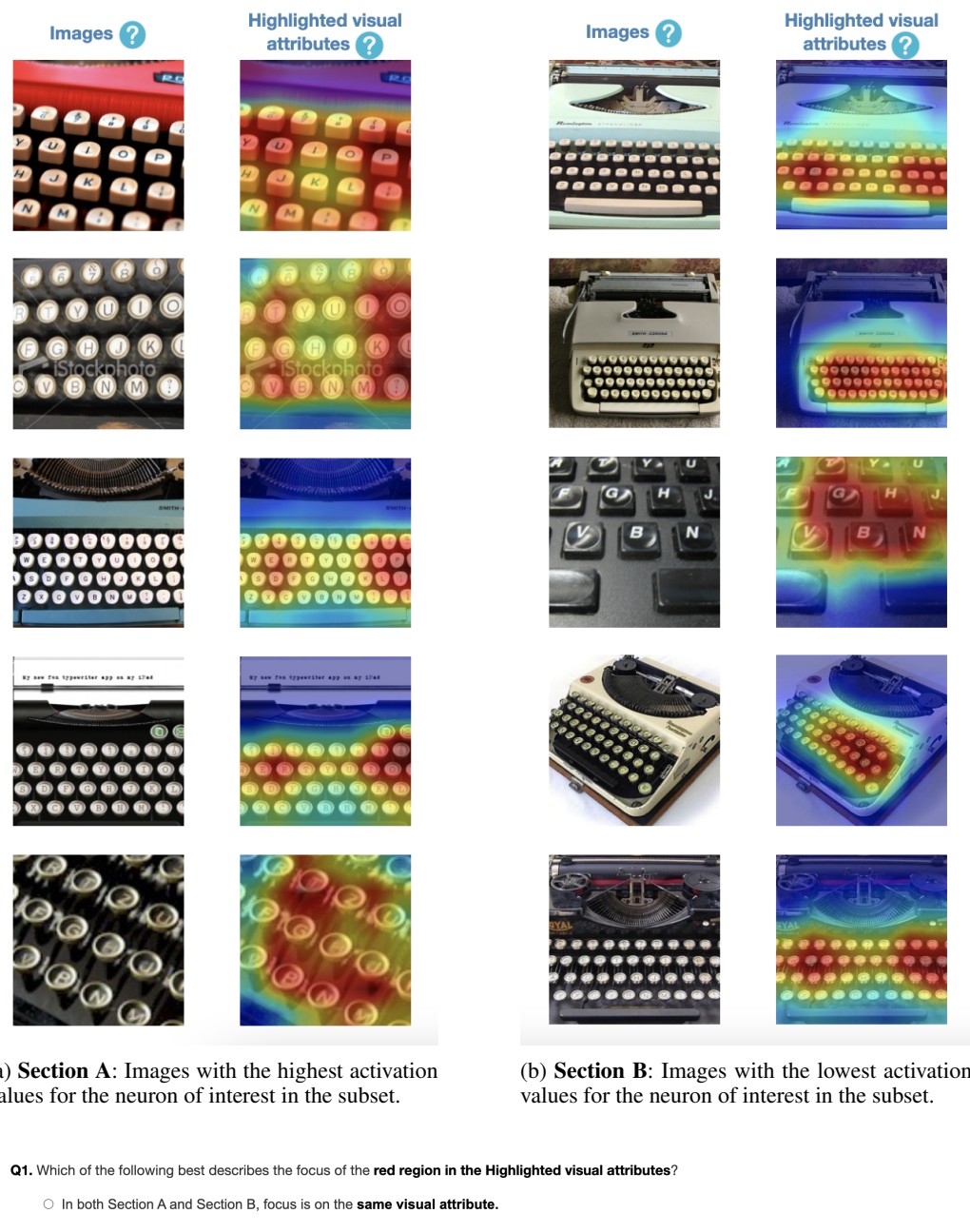

(a) **Section A**: Images with the highest activation values for the neuron of interest in the subset.

(b) **Section B**: Images with the lowest activation values for the neuron of interest in the subset.

**Q1.** Which of the following best describes the focus of the **red region in the Highlighted visual attributes**?

○ In both Section A and Section B, focus is on the **same visual attribute.**

○ In Section B, focus is on **different visual attribute** compared to Section A.

○ The focus of red region is **unclear in Section A.**

○ The focus of red region is **unclear in Section B.**

**Q2.** How confident are you about your answer to Q1?

(1) Not confident at all — (2) Slightly confident — (3) Somewhat confident — (4) Fairly confident — (5) Completely confident

**Q3.** Describe the visual attributes in Section A and Section B for your answer to Q1? This question is mandatory for completing this HIT. Detailed answers will receive a bonus.

(c) Questionnaire.

Figure 10: Mechanical Turk study for validating heatmaps

## E    DISCOVERING SPURIOUS VISUAL ATTRIBUTES IN OTHER IMAGE DATASETS USING OUR FRAMEWORK

In this work, we consider the problem of classification on Imagenet and introduce a methodology to discover core/spurious features. Our framework can be applied to discover the spurious/core visual attributes of other image classification datasets using the following steps:

(i) Train a robust model using the $l_2$ norm of some large radii (we use 3 in this work).

(ii) To discover spurious visual attributes for some class $i$ in the dataset, select the subset of images from the training dataset on which the robust model predicts $i$.

(iii) Select neural features from the penultimate layer of the robust model with the $k$ highest importance values (as defined in Section 1). We use $k = 5$ in this work. We note that other layers of the network may also provide informative neural features in some applications.

(iv) Visualize each neural feature using $m$ highly activating images, their heatmaps and feature attack (Section 3.2). We use $m = 5$ in this work. Note that both $k$ and $m$ may be changed depending on the problem at hand.

(v) Create a Mechanical Turk HIT per (class=$i$, neural feature=$j$) pair such that the HIT contains both the visualization for feature $j$ and also the details about the class $i$ (similar to Figure 9) and obtain worker annotations.

The HITs for which the majority of the workers annotate the neural feature to be spurious for class are deemed to be the spurious visual attributes. Then, simply one can use any Heatmap methods such as CAM to automatically highlight core and spurious masks on many other samples in the dataset. At this point, we recommend another MTurk study to validate the quality of highlighted core and spurious masks on new samples.

## F    ANALYZING THE REASONS PROVIDED BY MECHANICAL TURK WORKERS

Understanding the reasoning behind the answers of MTurk workers is a general and challenging problem in all crowd studies. To understand the reasons behind the answers provided by our MTurk workers, we have taken the following steps: (i) workers were required to give reasons for their answers to complete each task and (ii) they were explicitly instructed to use heatmaps (called highlighted visual attributes in the study) for their answers and use feature attack visualization (called amplified visual attributes in the study) for their answers when the heatmaps are unclear. For the crowd study for discovering spurious features (Section 3.3), we have observed more than 1200 mentions of the word "highlighted" and only 29 mentions of the word "amplified" suggesting that the workers use highlighted visual attributes (heatmaps) significantly more than the amplified visual attributes (feature attack). Some example descriptions provided by workers are given below in Figures 11 and 12.

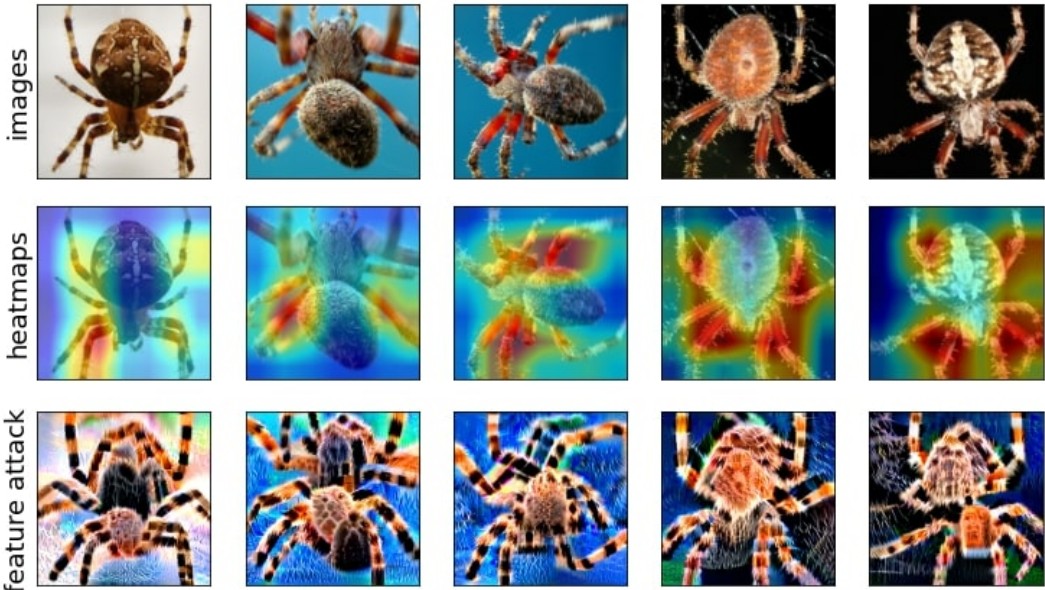

Figure 11: Visualization of feature **625** for class **barn spider** (class index: **73**).
Reasons for the answers provided by workers (all workers answered main object): (i) The red region in the Highlighted visual attributes focuses on the leg of the barn spider that is the main object, (ii) Legs of spider are mainly focused. (iii) Focus on spider, (iv) The red is on the legs of the barn spider, (v) Focus is on another part of the main object, such as the barn spider's legs.

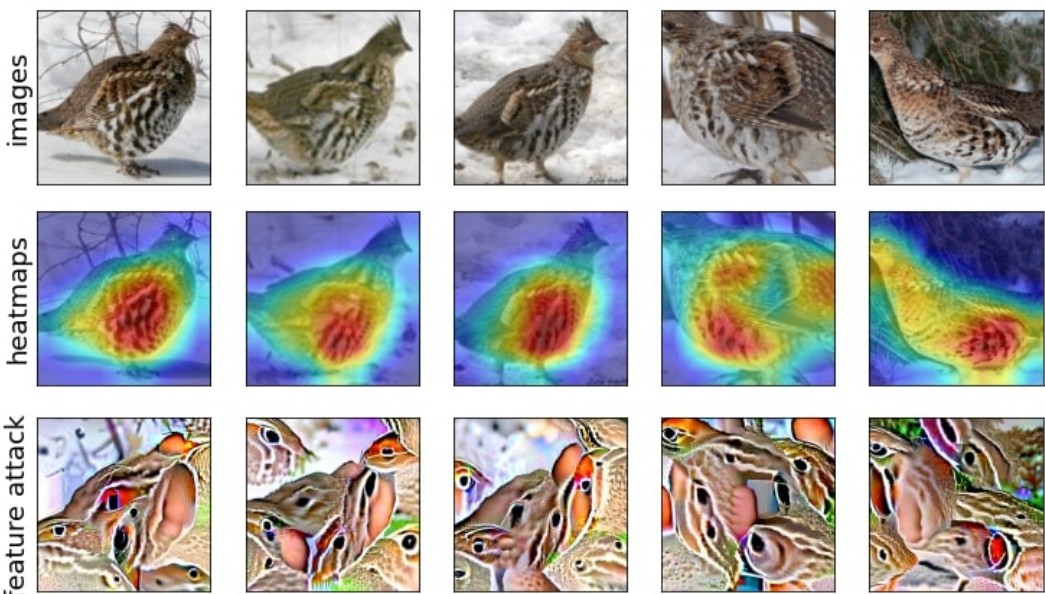

Figure 12: Visualization of feature **957** for class **ruffed grouse** (class index: **82**).
Reasons for the answers provided by workers (all workers answered main object): (i) The red region in the Highlighted visual attributes focuses on the feathers of the ruffed grouse that is the main object, (ii) focus is highlighting the bird's wing, (iii) The focus is on the feathers of the bird, (iv) Focus is on the bird, (v) All of the red parts are within the object

## G  FORMAL DEFINITIONS OF CORE AND SPURIOUS VISUAL ATTRIBUTES

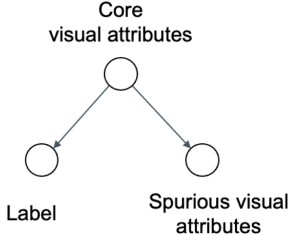

Figure 13: A simple graphical model relationship between core, spurious and label variables.

In the main text, we have explained definitions of core and spurious visual attributes used in our studies as following:

- **Core Attributes** are the set of visual features that are always a part of the object definition.
- **Spurious Attributes** are the ones that are likely to *co-occur* with the object, but not a part of it (such as food, vase, flowers in Figure 1).

In fact, defining these abstract concepts for Mturkers was quite challenging: For example, in our first user studies on Mechanical Turk, we have observed that defining spurious visual attributes as attributes that could be removed without changing the label of the object resulted in workers labeling attributes such as "limbs" to be spurious for the class "dog" (because a dog can still be recognized when the limbs are removed). Thus, for the purpose of our user studies, we defined the core visual attributes as the attributes that are a part of the relevant object definition. With this definition, "limbs" would be considered as a core feature for the class "dog". Similarly, we have defined spurious visual attributes as the attributes that are not a part of the object definition.

Although not necessary in our empirical Mechanical Turk studies, one can define core and spurious visual attributes formally through a simple graphical model depicted in Figure 13 (as explained in Khani & Liang (2021)) Let us consider $L$, $C$ and $S$ as label, core and spurious random variables, respectively. This graphical model indicates that given $C$, $L$ and $S$ are conditionally independent. That is, spurious variables are not essential in the object definition. This graphical model can be viewed through the lens of the Reichenbach's Common Cause Principle which indicates that since there is a correlation (and not causation) relationship between $S$ and $L$, they have a common cause that renders them conditionally independent. We emphasize that even this more formal definition of core/spurious features is *incomplete* since in practice there may be other confounding factors ignored in the representation of Figure 13.

# H    ADDITIONAL RESULTS FROM EVALUATING DEEP MODELS ON THE SALIENT IMAGENET DATASET

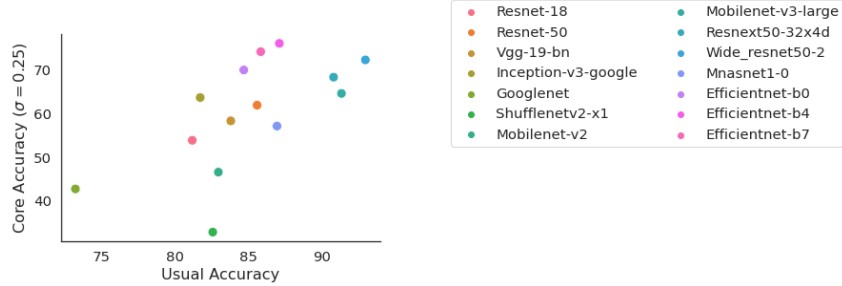

Figure 14: Comparing between the core accuracy (using $\sigma = 0.25$) and usual accuracy of different Imagenet trained models.

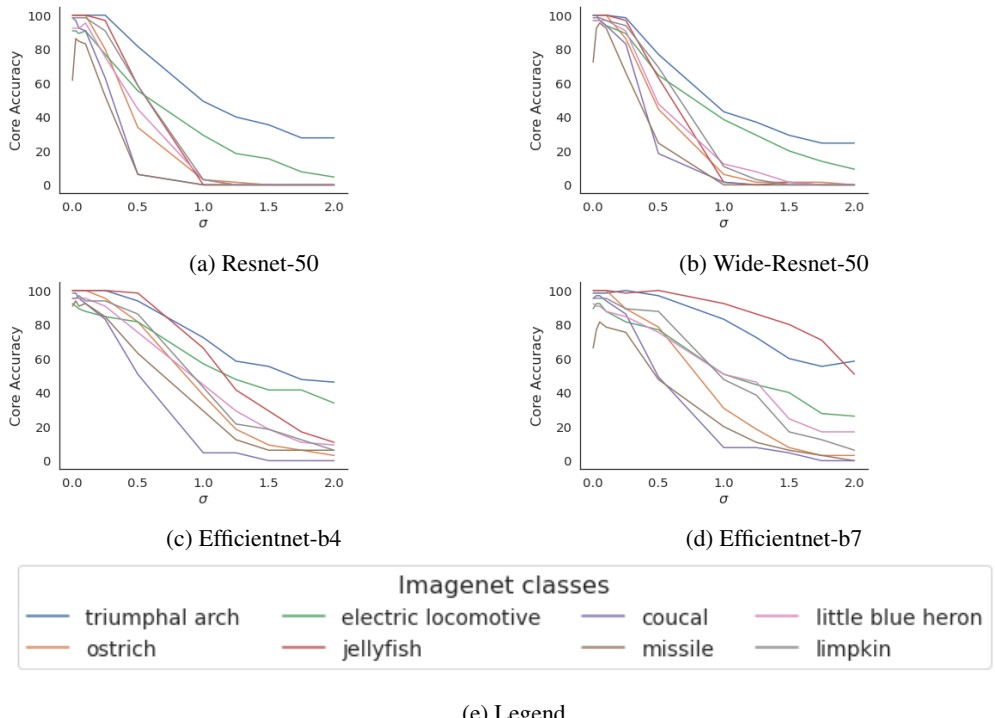

Figure 15: We plot the core accuracy for different classes as noise level $\sigma$ increases. For some classes, core accuracy is high even at large $\sigma$, but for most it decreases rapidly (e.g., triumphal arch, jellyfish on Efficientnet-b7).

# I EXAMPLES OF IMAGES FROM THE SALIENT IMAGENET DATASET CORRUPTED BY ADDING GAUSSIAN NOISE TO THE SPURIOUS REGIONS

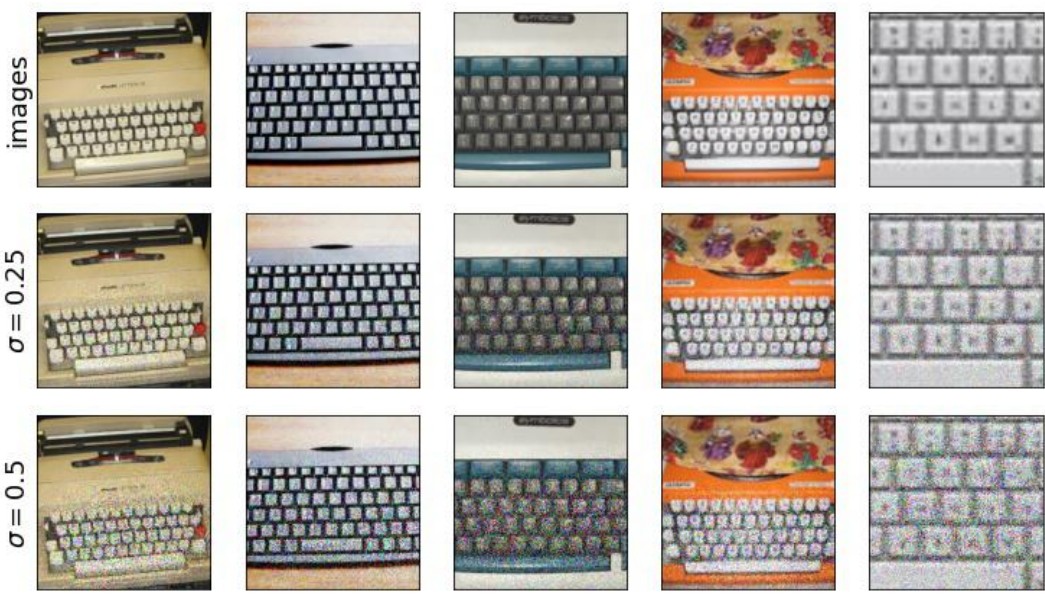

Figure 16: Images randomly sampled from the set $\mathcal{D}(i, j)$ where $i = 810$ (class index) and $j = 325$ (feature index). Class name: space bar.

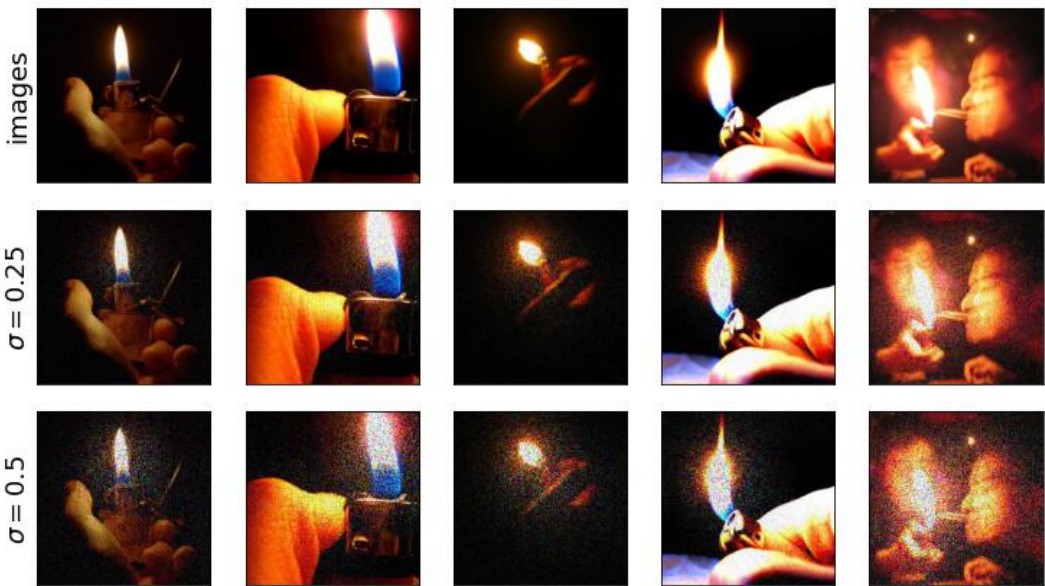

Figure 17: Images randomly sampled from the set $\mathcal{D}(i, j)$ where $i = 626$ (class index) and $j = 1986$ (feature index). Class name: lighter.

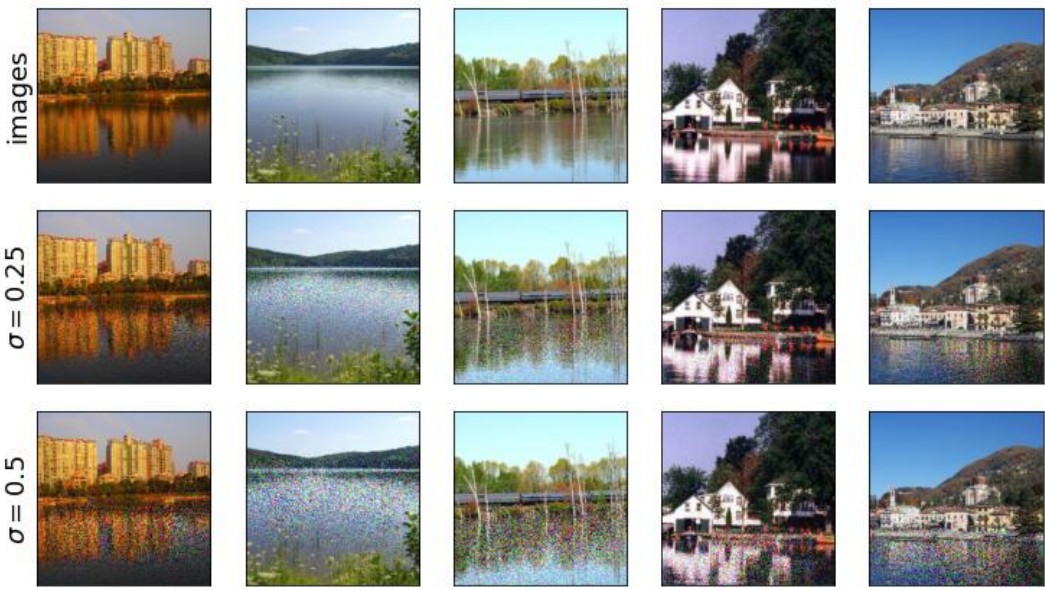

Figure 18: Images randomly sampled from the set $\mathcal{D}(i, j)$ where $i = 975$ (class index) and $j = 516$ (feature index). Class name: lakeside, lakeshore.

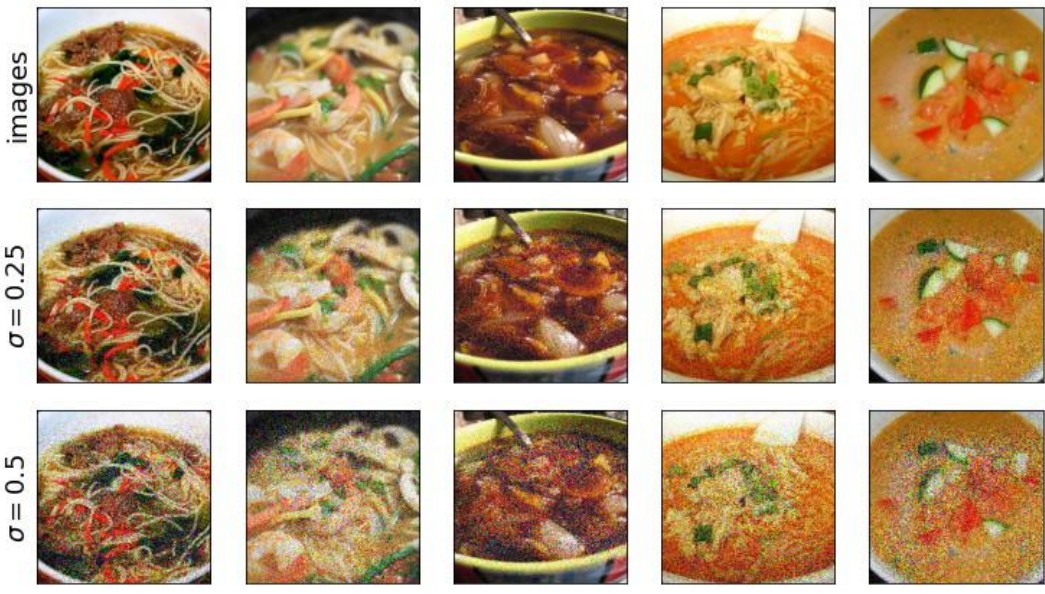

Figure 19: Images randomly sampled from the set $\mathcal{D}(i, j)$ where $i = 809$ (class index) and $j = 895$ (feature index). Class name: soup bowl.

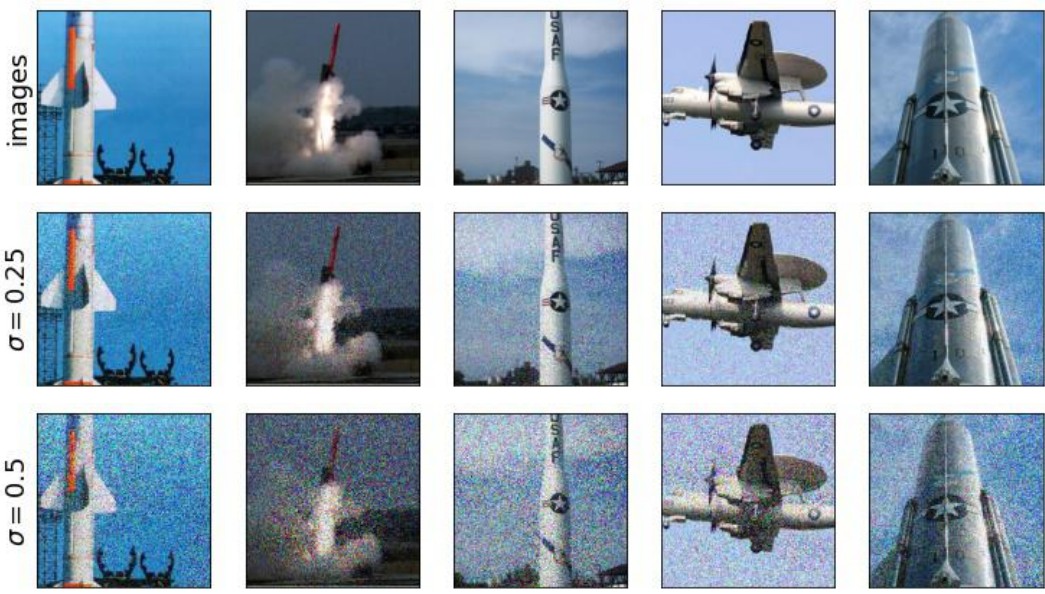

Figure 20: Images randomly sampled from the set $\mathcal{D}(i, j)$ where $i = 657$ (class index) and $j = 961$ (feature index). Class name: missile.

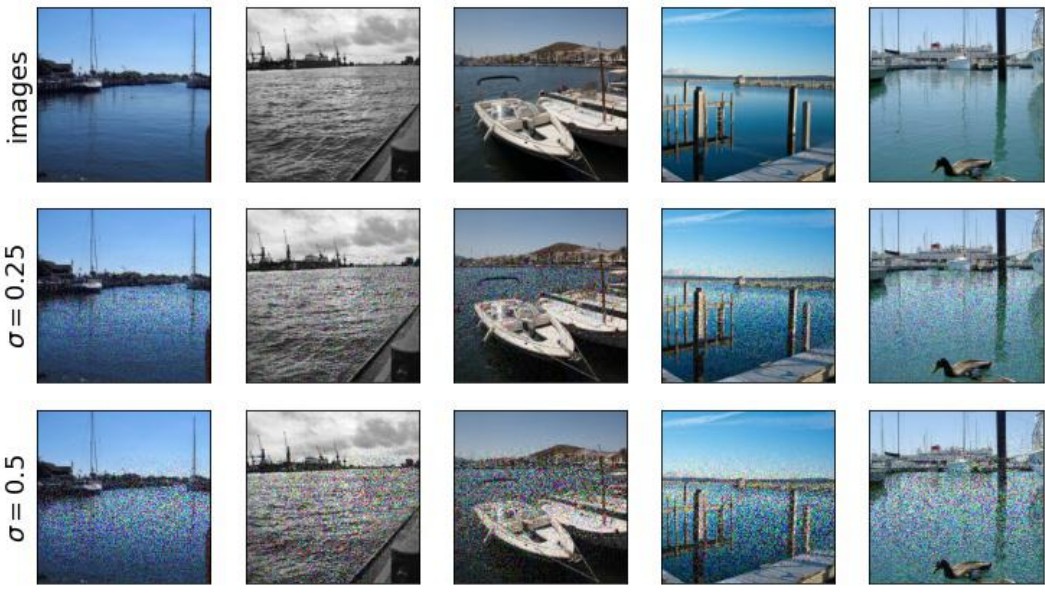

Figure 21: Images randomly sampled from the set $\mathcal{D}(i, j)$ where $i = 536$ (class index) and $j = 76$ (feature index). Class name: dock.

## J EXAMPLES OF SPURIOUS FEATURES

### J.1 BACKGROUND SPURIOUS FEATURES

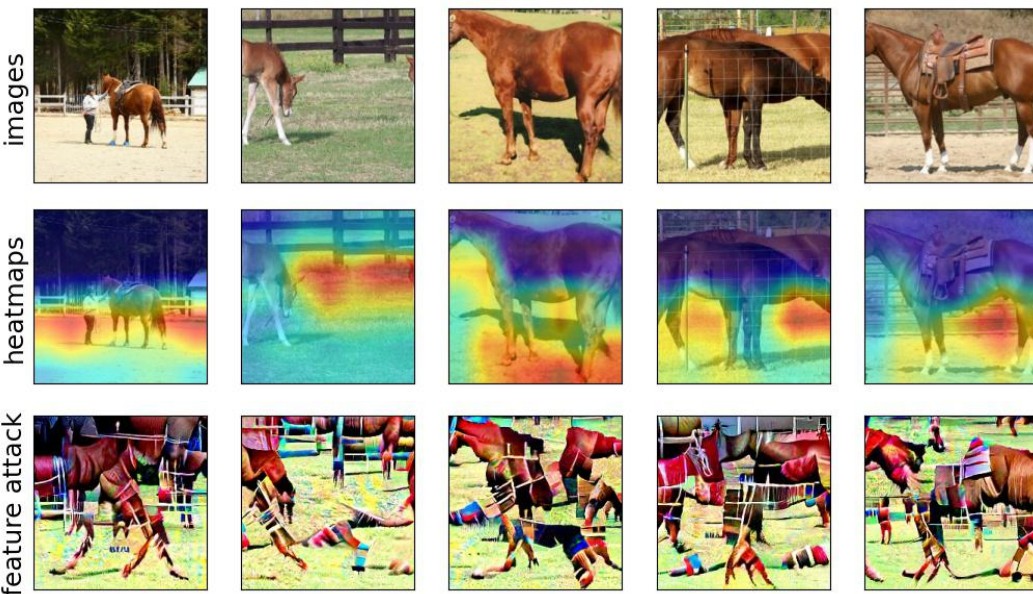

Figure 22: Visualization of feature **230** for class **sorrel** (class index: **339**).
Train accuracy using Standard Resnet-50: **99.385**%.
Drop in accuracy when gaussian noise is added to the highlighted (red) regions: **-12.308**%.

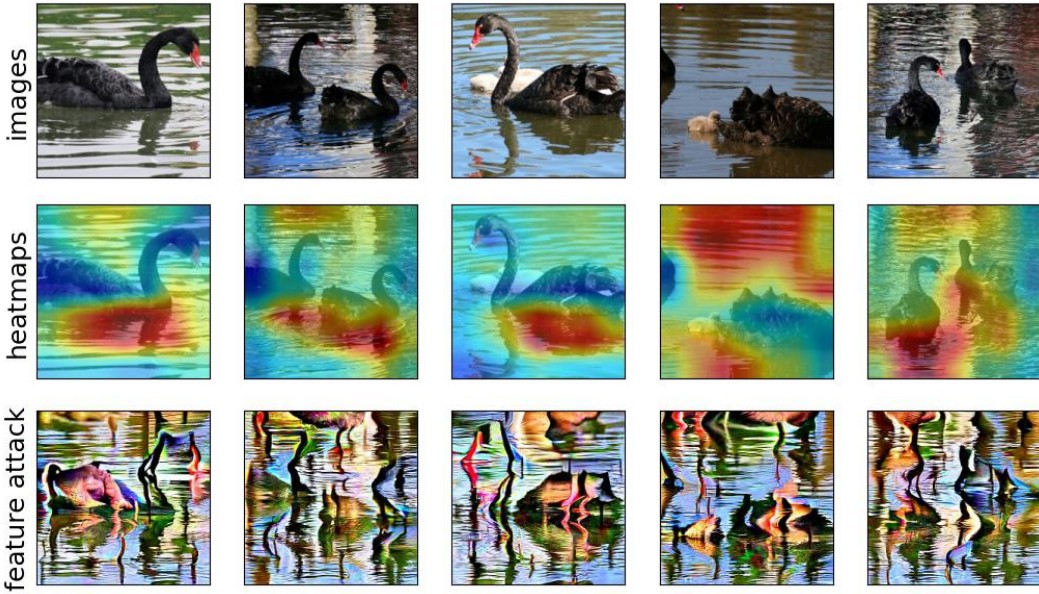

Figure 23: Visualization of feature **925** for class **black swan** (class index: **100**).
Train accuracy using Standard Resnet-50: **98.769**%.
Drop in accuracy when gaussian noise is added to the highlighted (red) regions: **-4.615**%.

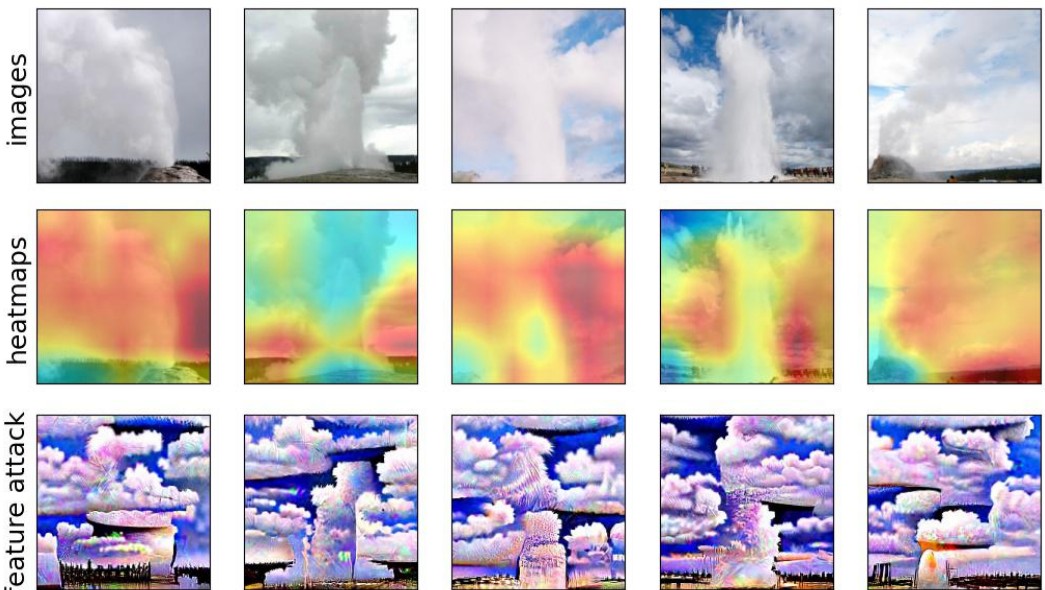

Figure 24: Visualization of feature **223** for class **geyser** (class index: **974**).
Train accuracy using Standard Resnet-50: **97.923**%.
Drop in accuracy when gaussian noise is added to the highlighted (red) regions: **-20.0**%.

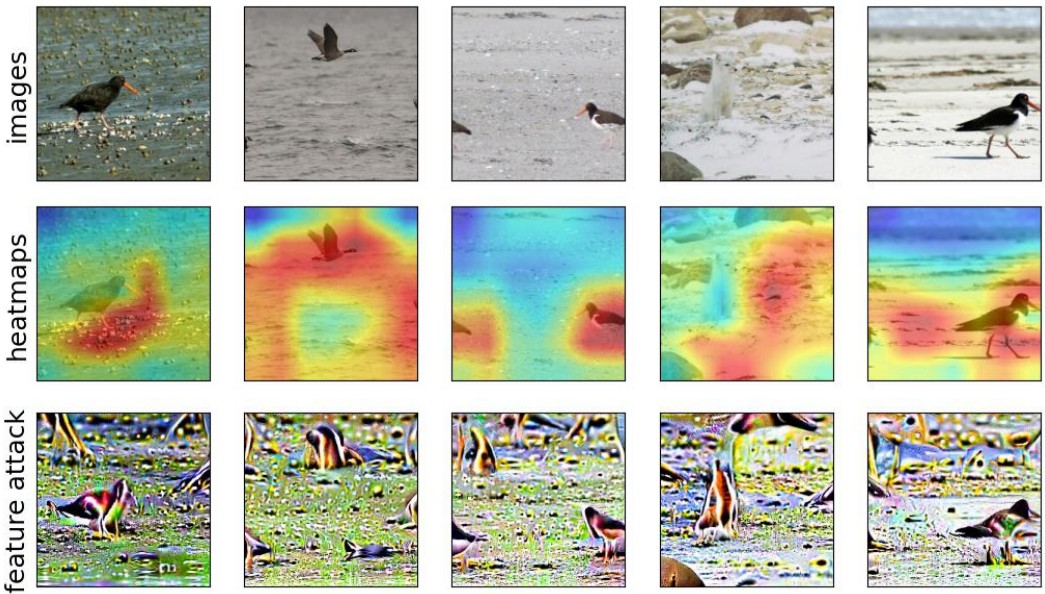

Figure 25: Visualization of feature **1468** for class **oystercatcher** (class index: **143**).
Train accuracy using Standard Resnet-50: **97.769**%.
Drop in accuracy when gaussian noise is added to the highlighted (red) regions: **-21.539**%.

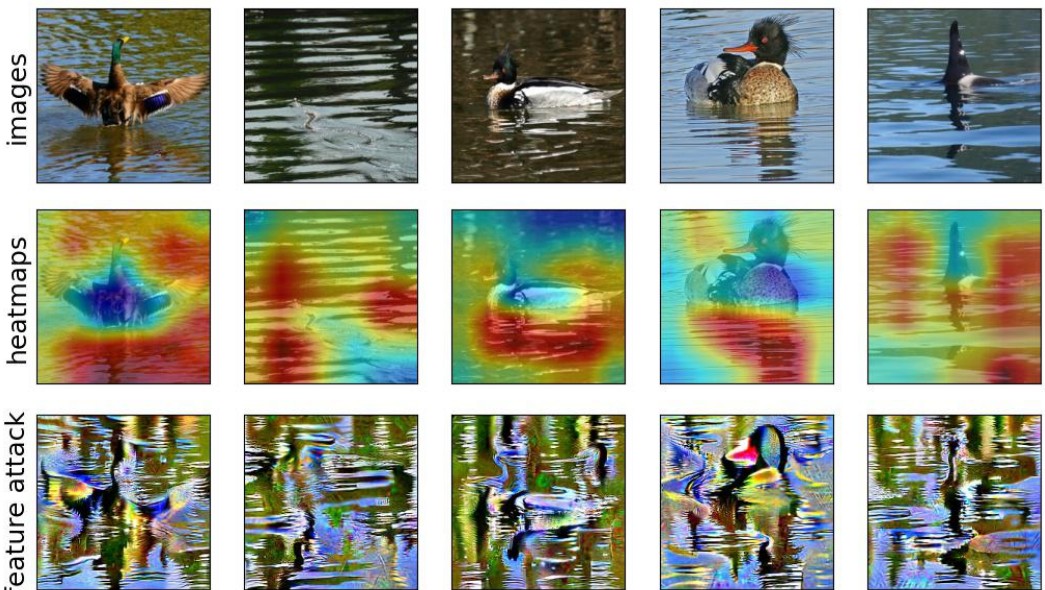

Figure 26: Visualization of feature **341** for class **red breasted merganser** (class index: **98**).
Train accuracy using Standard Resnet-50: **97.195**%.
Drop in accuracy when gaussian noise is added to the highlighted (red) regions: **-12.308**%.

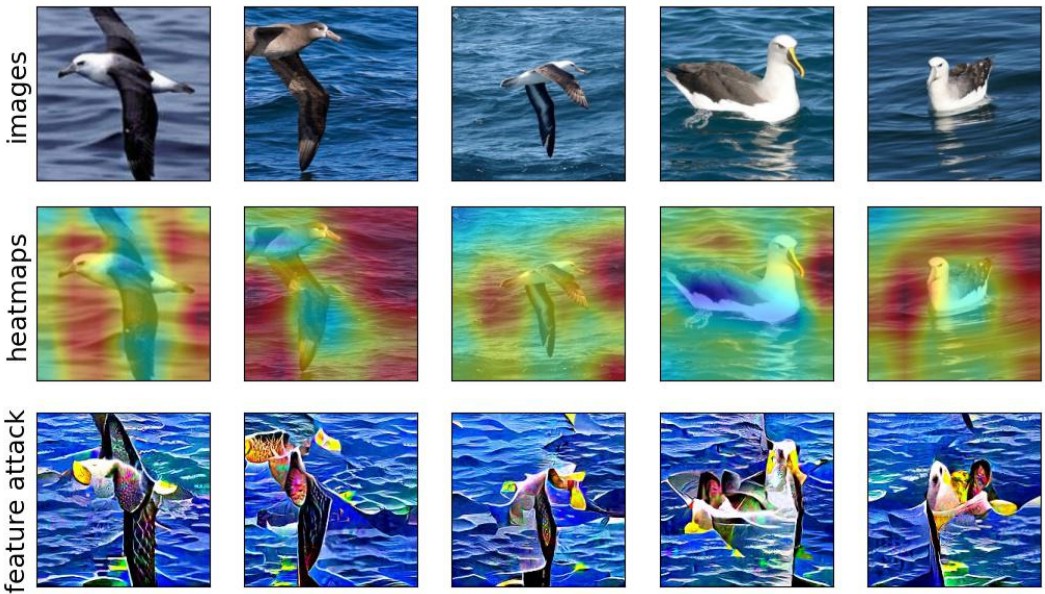

Figure 27: Visualization of feature **1697** for class **albatross** (class index: **146**).
Train accuracy using Standard Resnet-50: **96.615**%.
Drop in accuracy when gaussian noise is added to the highlighted (red) regions: **-20.0**%.

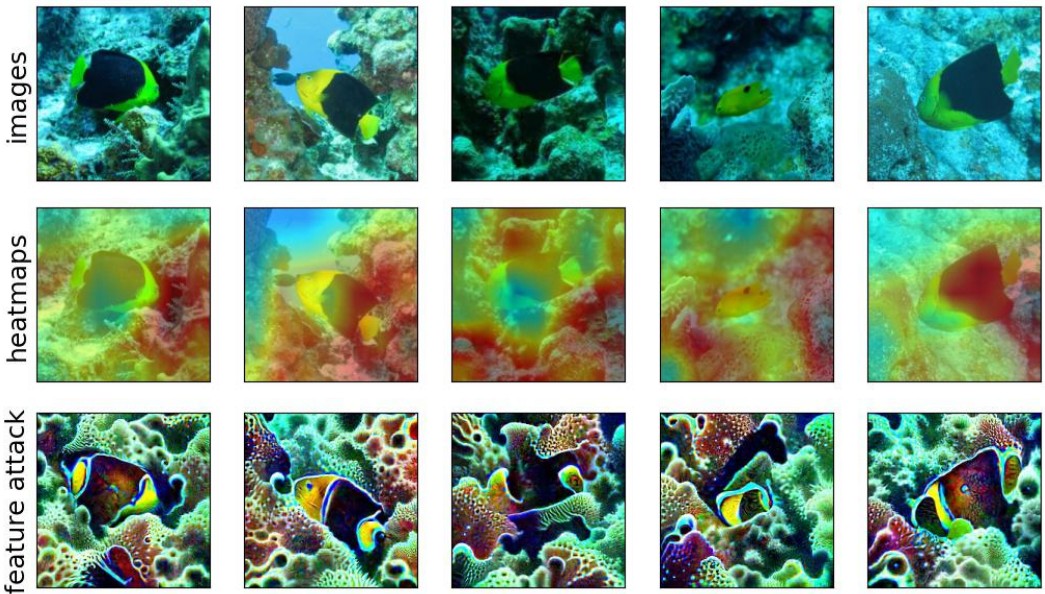

Figure 28: Visualization of feature **981** for class **rock beauty** (class index: **392**).
Train accuracy using Standard Resnet-50: **96.285**%.
Drop in accuracy when gaussian noise is added to the highlighted (red) regions: **-32.308**%.

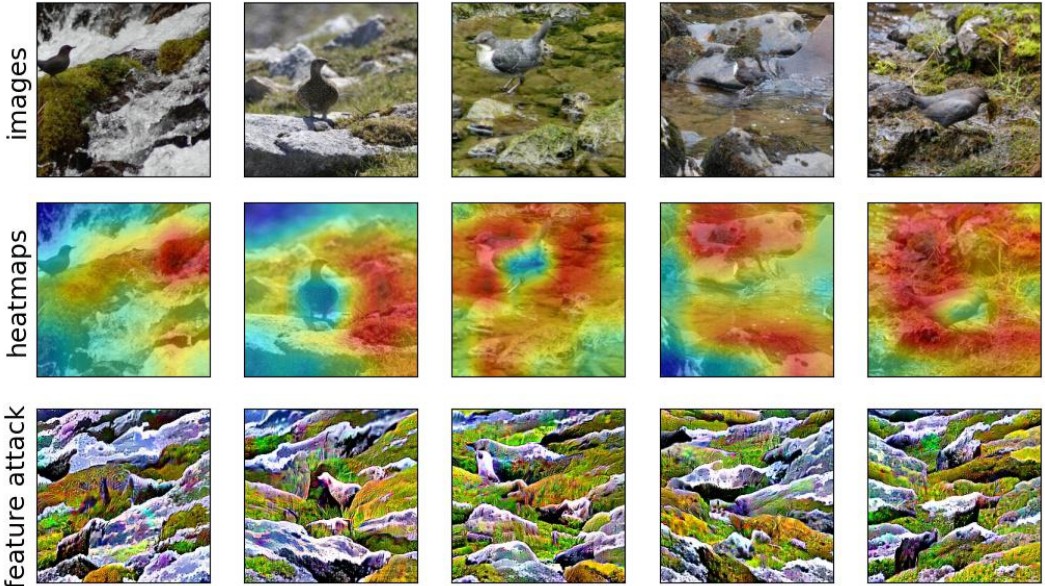

Figure 29: Visualization of feature **820** for class **water ouzel** (class index: **20**).
Train accuracy using Standard Resnet-50: **96.231**%.
Drop in accuracy when gaussian noise is added to the highlighted (red) regions: **-15.385**%.

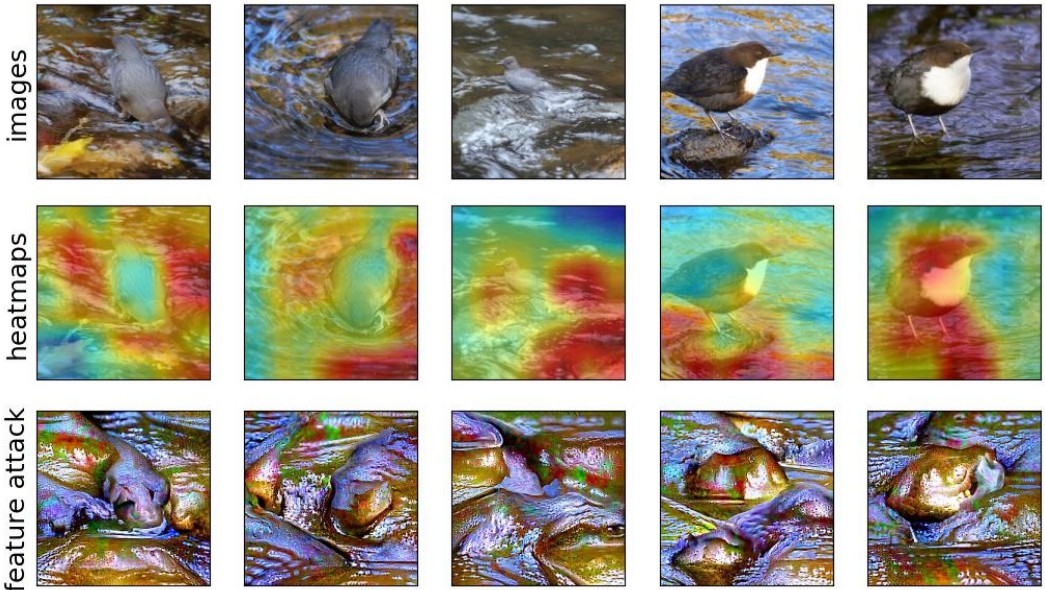

Figure 30: Visualization of feature **535** for class **water ouzel** (class index: **20**).
Train accuracy using Standard Resnet-50: **96.231**%.
Drop in accuracy when gaussian noise is added to the highlighted (red) regions: **-16.923**%.

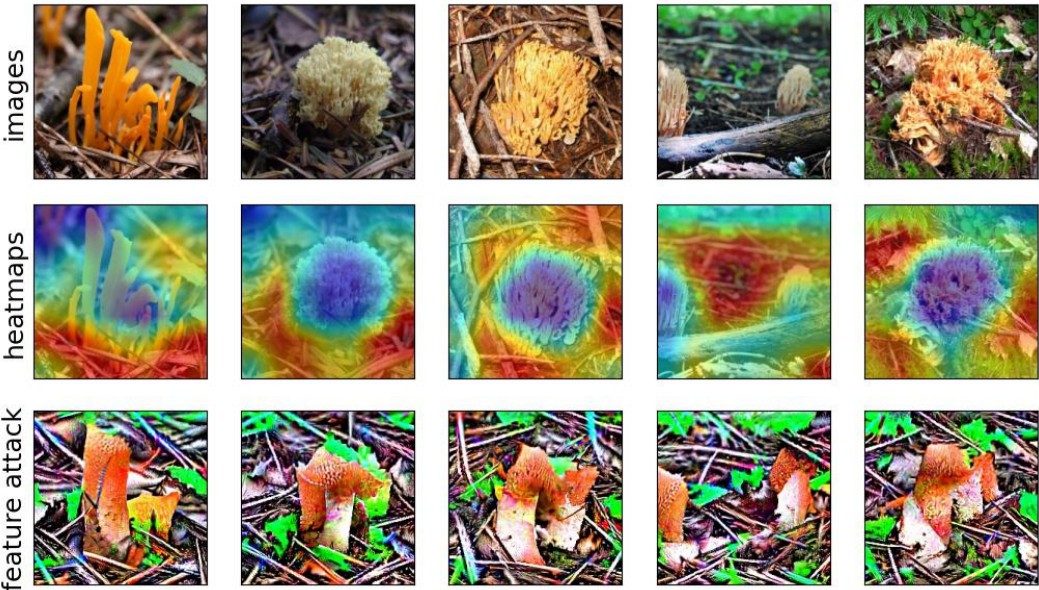

Figure 31: Visualization of feature **1475** for class **coral fungus** (class index: **991**).
Train accuracy using Standard Resnet-50: **93.308**%.
Drop in accuracy when gaussian noise is added to the highlighted (red) regions: **-9.231**%.

## J.2 FOREGROUND SPURIOUS FEATURES

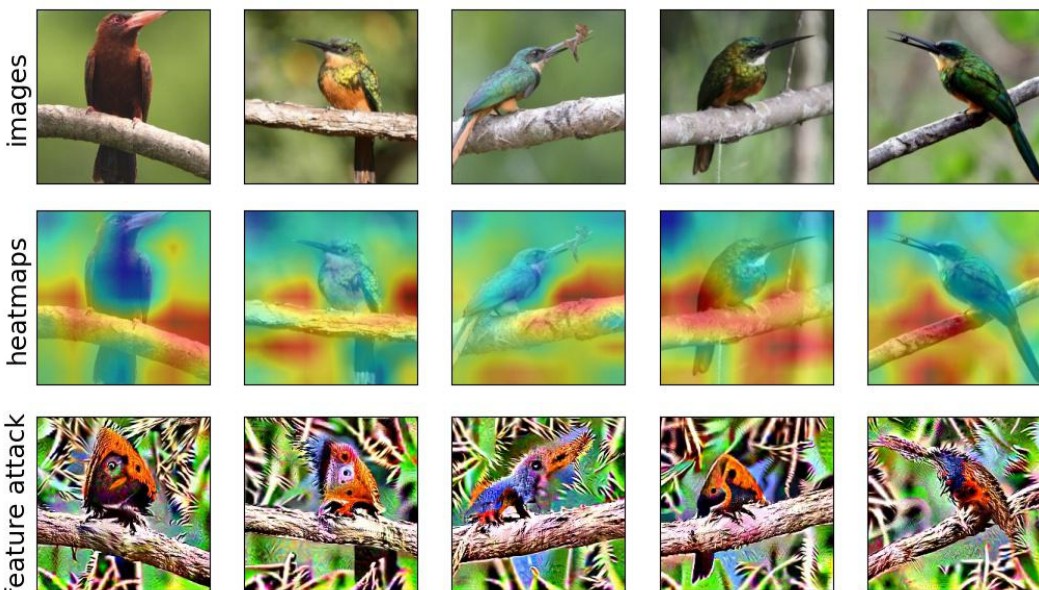

Figure 32: Visualization of feature **1556** for class **jacamar** (class index: **95**).
Train accuracy using Standard Resnet-50: **97.000**%.
Drop in accuracy when gaussian noise is added to the highlighted (red) regions: **-21.538**%.

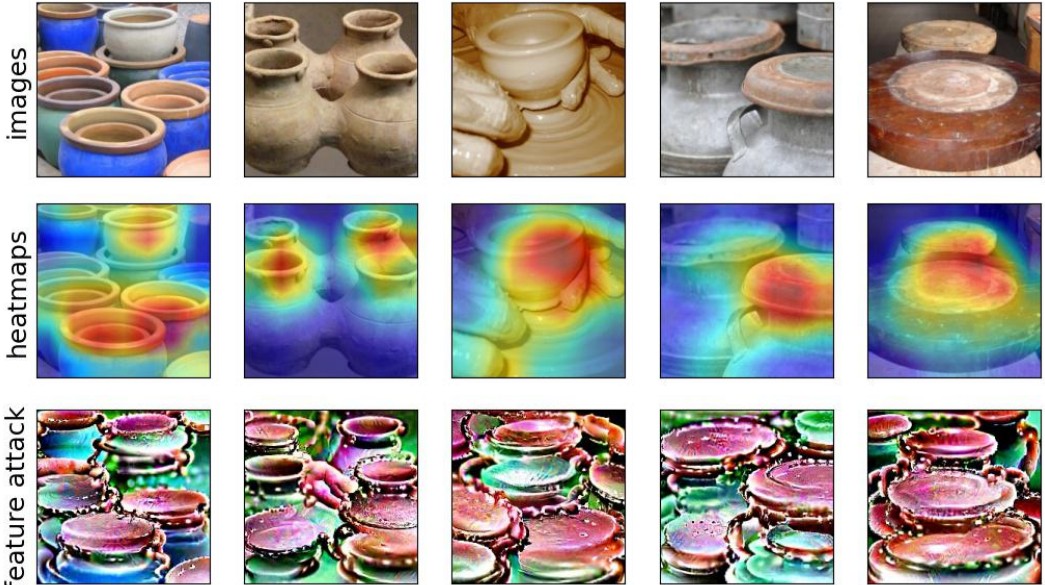

Figure 33: Visualization of feature **189** for class **potter's wheel** (class index: **739**).
Train accuracy using Standard Resnet-50: **95.615**%.
Drop in accuracy when gaussian noise is added to the highlighted (red) regions: **-21.538**%.

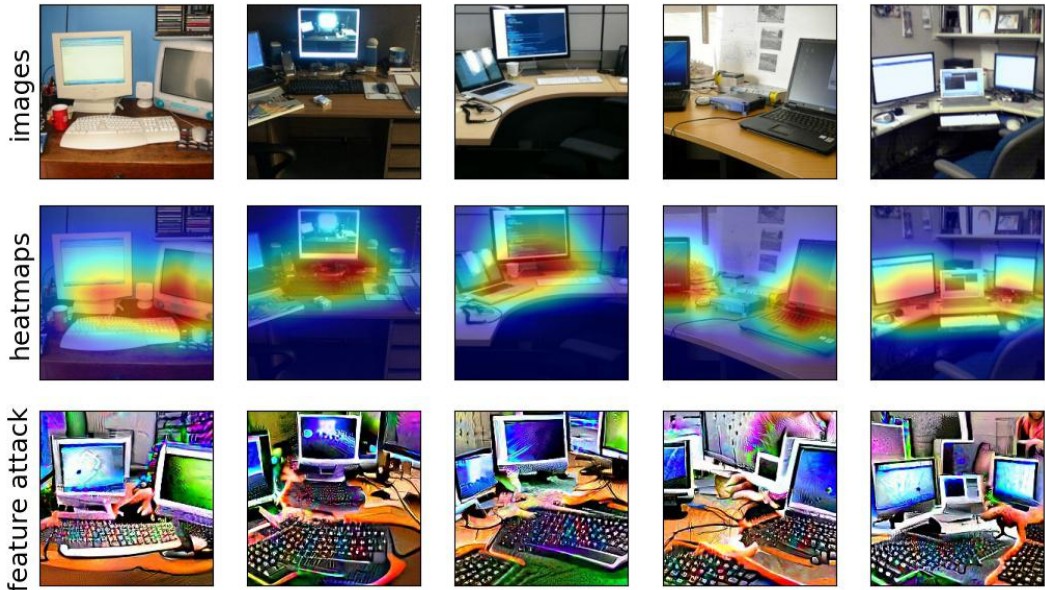

Figure 34: Visualization of feature **1832** for class **desk** (class index: **526**).
Train accuracy using Standard Resnet-50: **69.769**%.
Drop in accuracy when gaussian noise is added to the highlighted (red) regions: **-7.692**%.

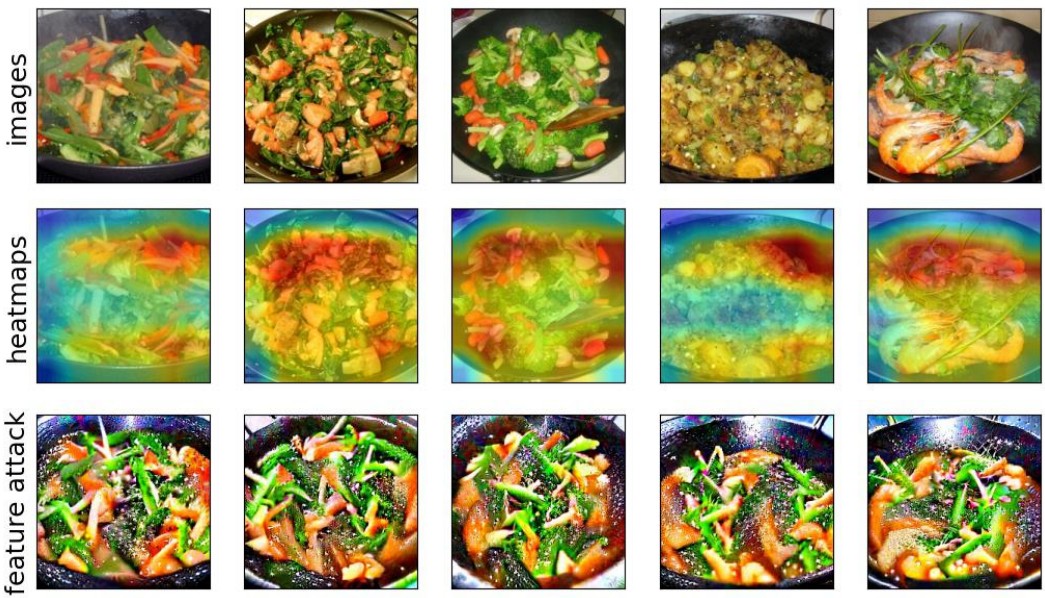

Figure 35: Visualization of feature **895** for class **wok** (class index: **909**).
Train accuracy using Standard Resnet-50: **71.077**%.
Drop in accuracy when gaussian noise is added to the highlighted (red) regions: **-46.154**%.

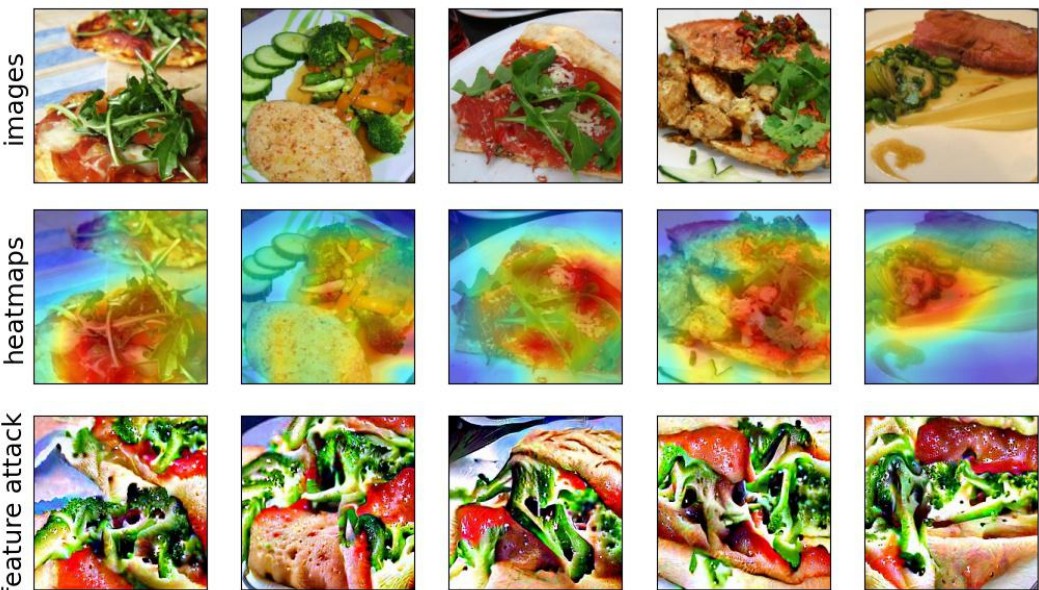

Figure 36: Visualization of feature **43** for class **plate** (class index: **923**).
Train accuracy using Standard Resnet-50: **64.538**%.
Drop in accuracy when gaussian noise is added to the highlighted (red) regions: **-32.308**%.

## J.3 SPURIOUS FEATURES BY FEATURE RANKS

### J.3.1 SPURIOUS FEATURES AT FEATURE RANK 1

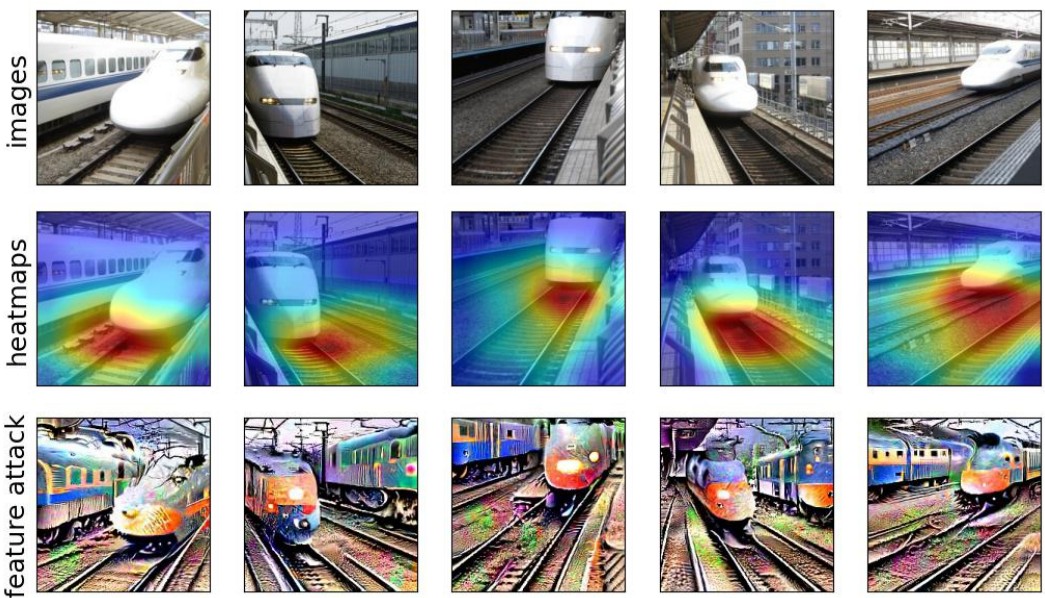

Figure 37: Visualization of feature **56** for class **bullet train** (class index: **466**).
Train accuracy using Standard Resnet-50: **98.077**%.
Drop in accuracy when gaussian noise is added to the highlighted (red) regions: **-1.538**%.

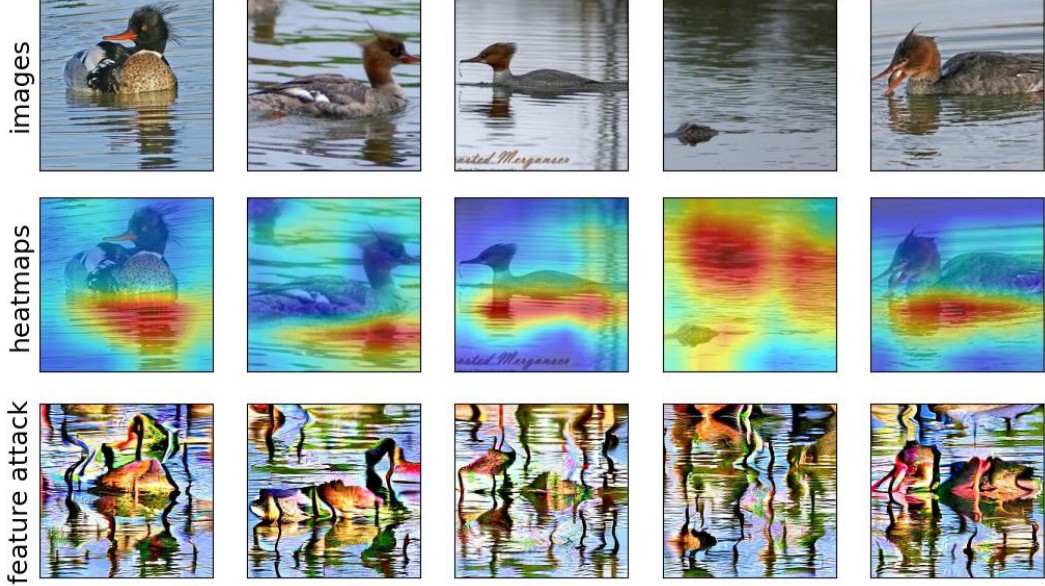

Figure 38: Visualization of feature **925** for class **red breasted merganser** (class index: **98**).
Train accuracy using Standard Resnet-50: **97.195**%.
Drop in accuracy when gaussian noise is added to the highlighted (red) regions: **-3.077**%.

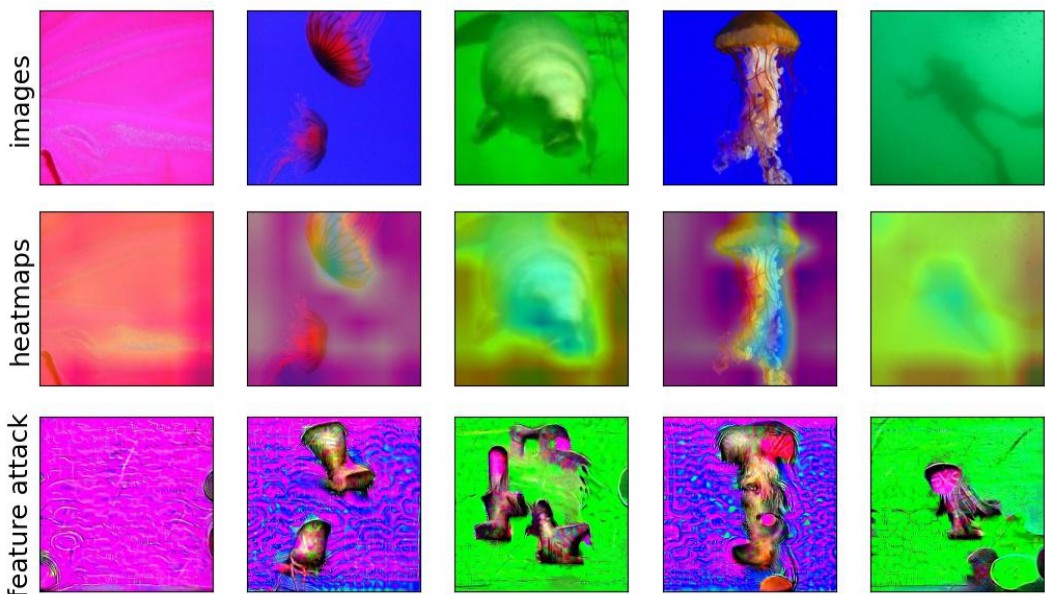

Figure 39: Visualization of feature **1406** for class **jellyfish** (class index: **107**).
Train accuracy using Standard Resnet-50: **98.077**%.
Drop in accuracy when gaussian noise is added to the highlighted (red) regions: **-4.615**%.

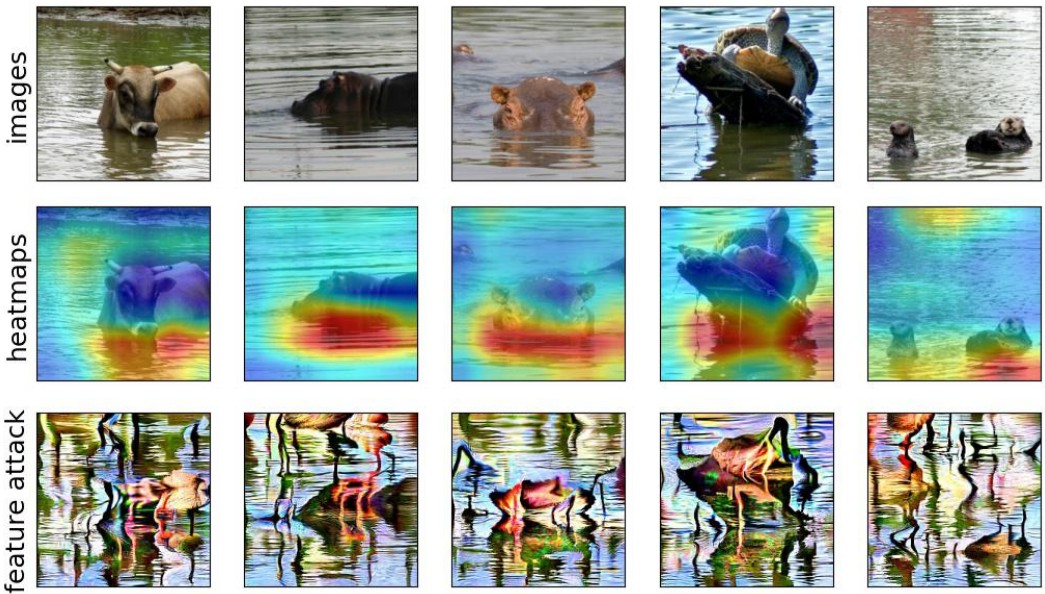

Figure 40: Visualization of feature **925** for class **hippopotamus** (class index: **344**).
Train accuracy using Standard Resnet-50: **97.846**%.
Drop in accuracy when gaussian noise is added to the highlighted (red) regions: **-27.693**%.

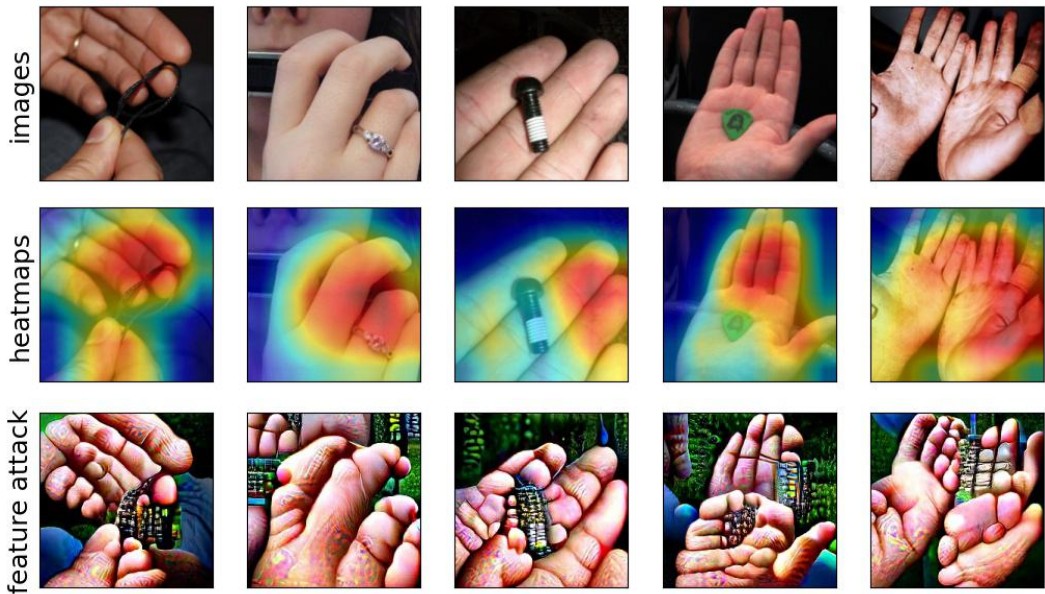

Figure 41: Visualization of feature **123** for class **band aid** (class index: **419**).
Train accuracy using Standard Resnet-50: **81.000**%.
Drop in accuracy when gaussian noise is added to the highlighted (red) regions: **-41.538**%.

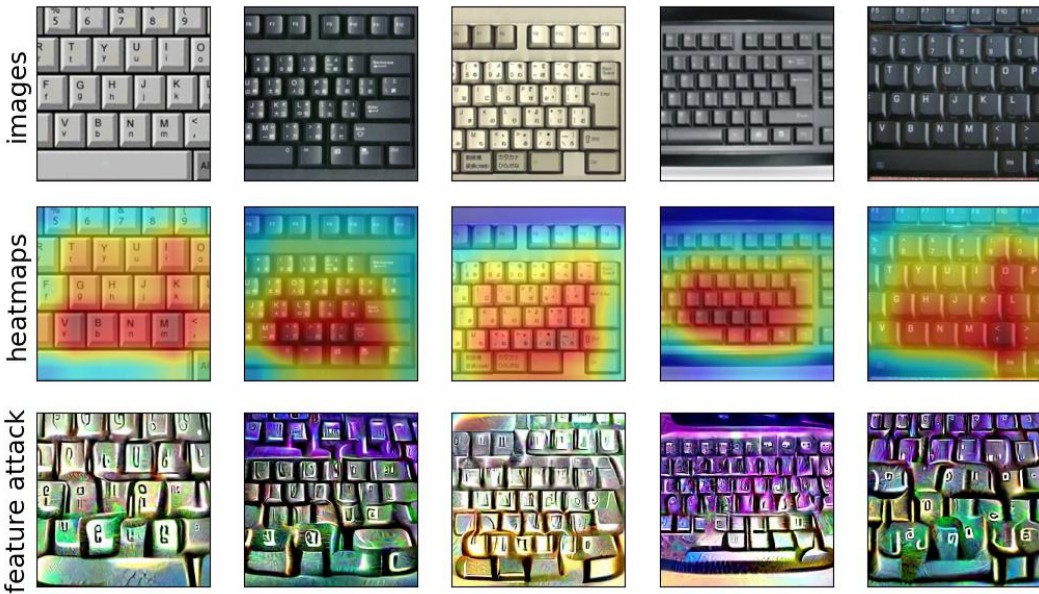

Figure 42: Visualization of feature **325** for class **space bar** (class index: **810**).
Train accuracy using Standard Resnet-50: **59.043**%.
Drop in accuracy when gaussian noise is added to the highlighted (red) regions: **-46.154**%.

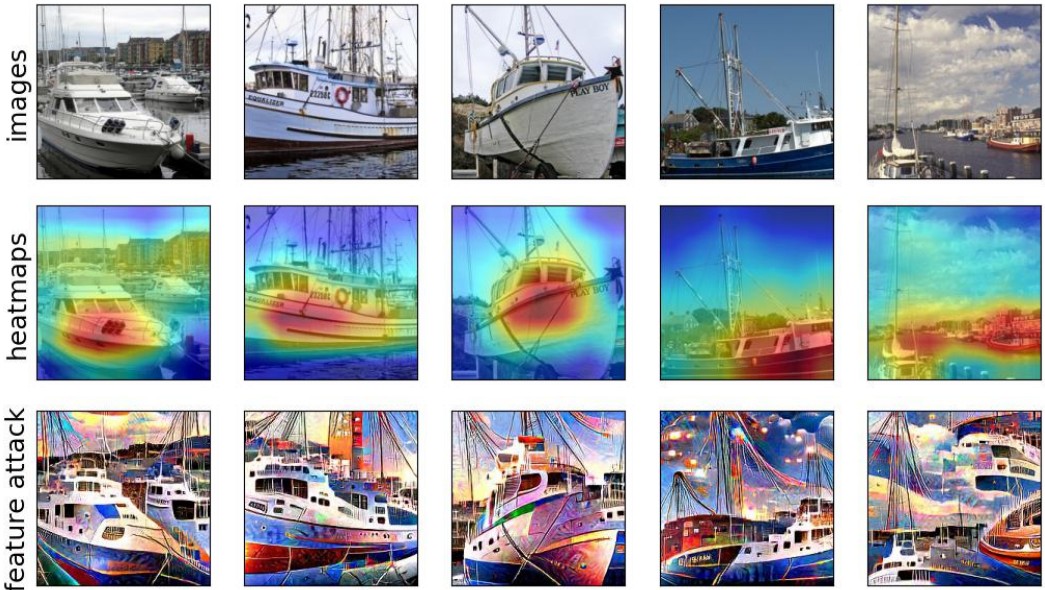

Figure 43: Visualization of feature **1536** for class **dock** (class index: **536**).
Train accuracy using Standard Resnet-50: **71.098**%.
Drop in accuracy when gaussian noise is added to the highlighted (red) regions: **-35.385**%.

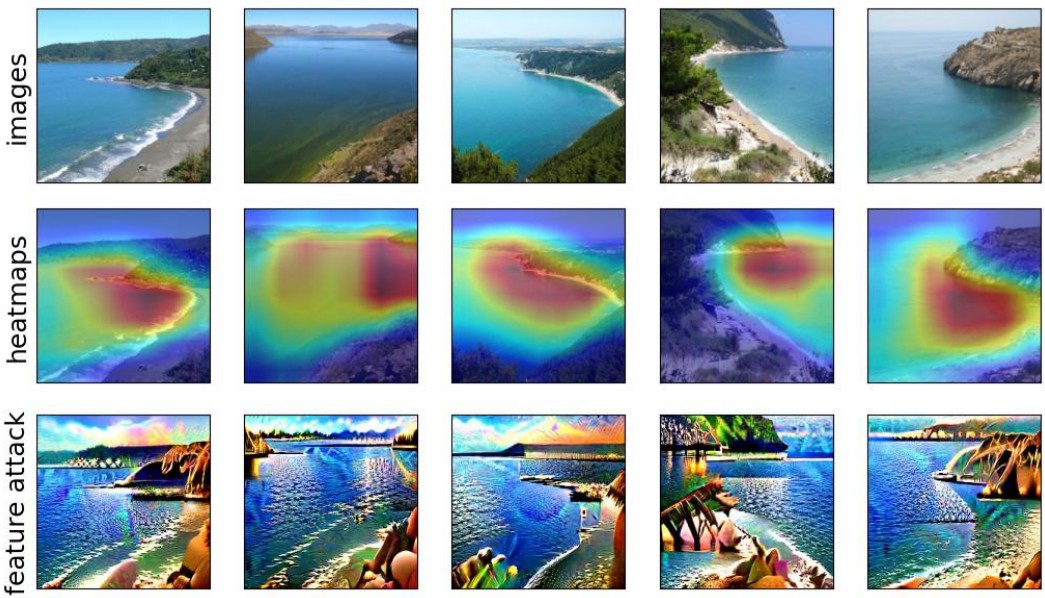

Figure 44: Visualization of feature **421** for class **promontory** (class index: **976**).
Train accuracy using Standard Resnet-50: **82.462**%.
Drop in accuracy when gaussian noise is added to the highlighted (red) regions: **-15.384**%.

### J.3.2 SPURIOUS FEATURES AT FEATURE RANK 2

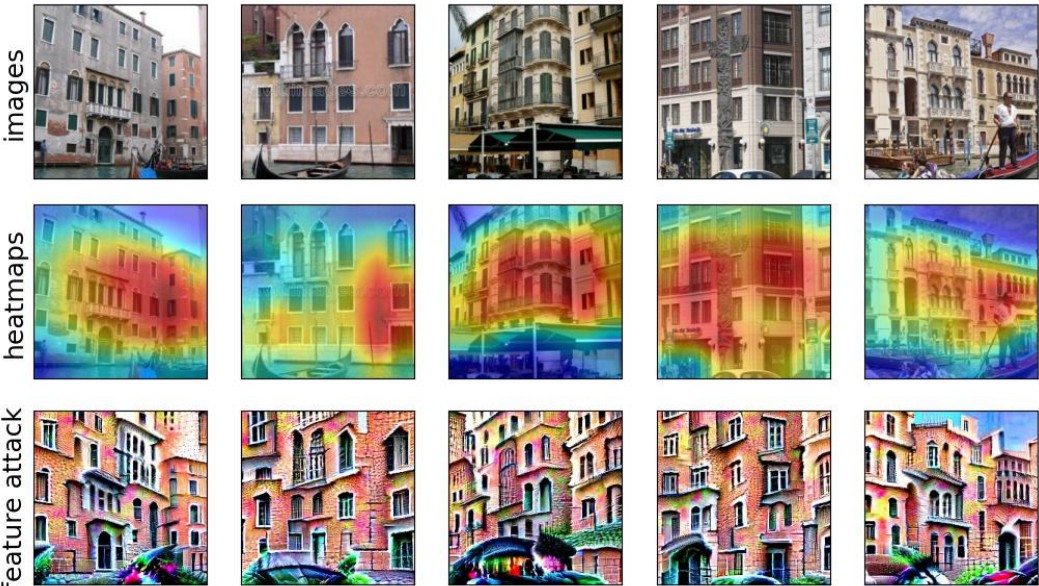

Figure 45: Visualization of feature **1772** for class **gondola** (class index: **576**).
Train accuracy using Standard Resnet-50: **98.308**%.
Drop in accuracy when gaussian noise is added to the highlighted (red) regions: **-1.538**%.

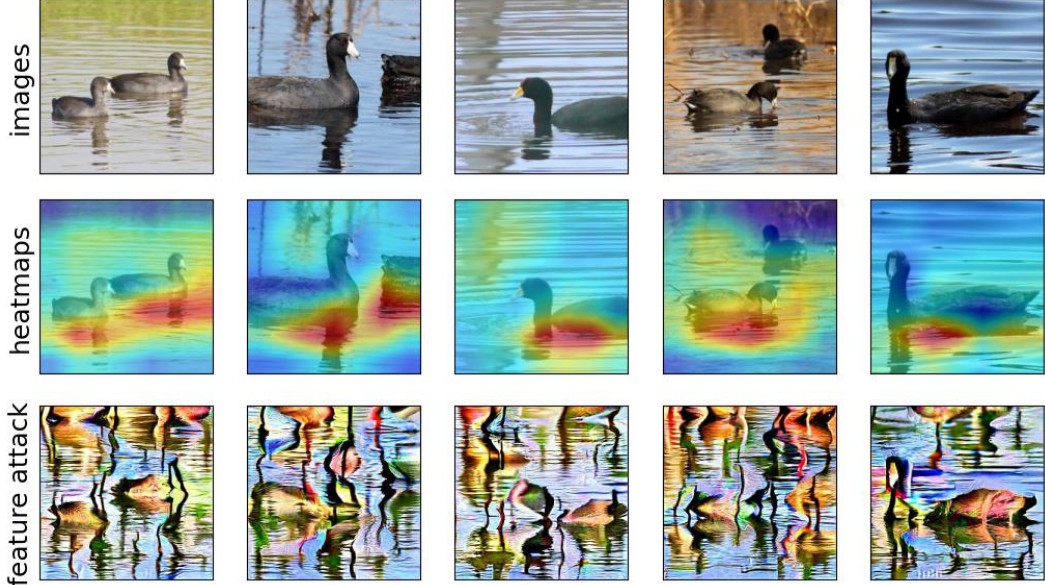

Figure 46: Visualization of feature **925** for class **american coot** (class index: **137**).
Train accuracy using Standard Resnet-50: **97.538**%.
Drop in accuracy when gaussian noise is added to the highlighted (red) regions: **-3.077**%.

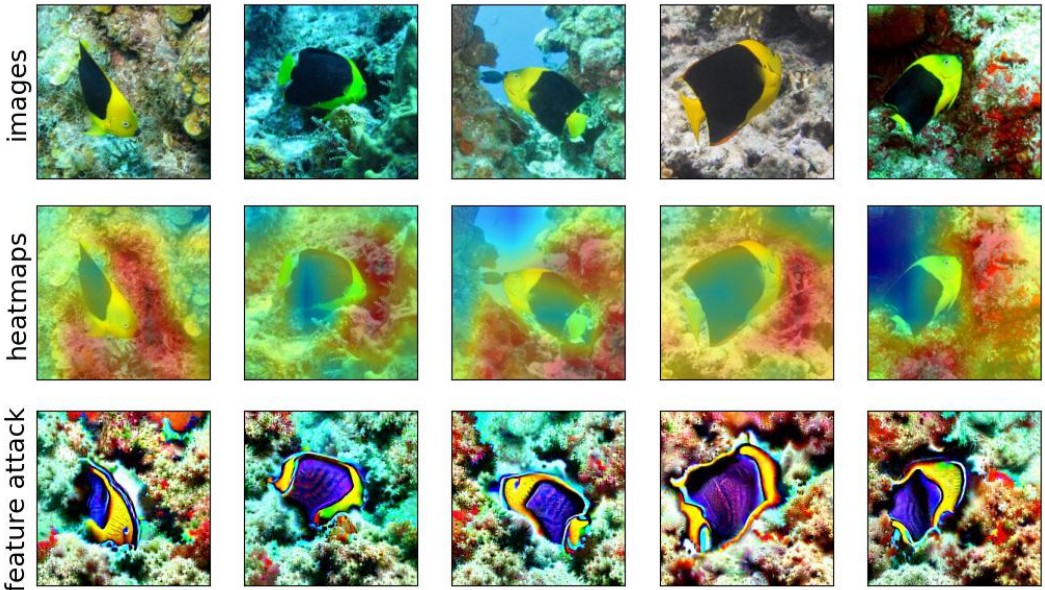

Figure 47: Visualization of feature **1753** for class **rock beauty** (class index: **392**).
Train accuracy using Standard Resnet-50: **96.285**%.
Drop in accuracy when gaussian noise is added to the highlighted (red) regions: **-12.308**%.

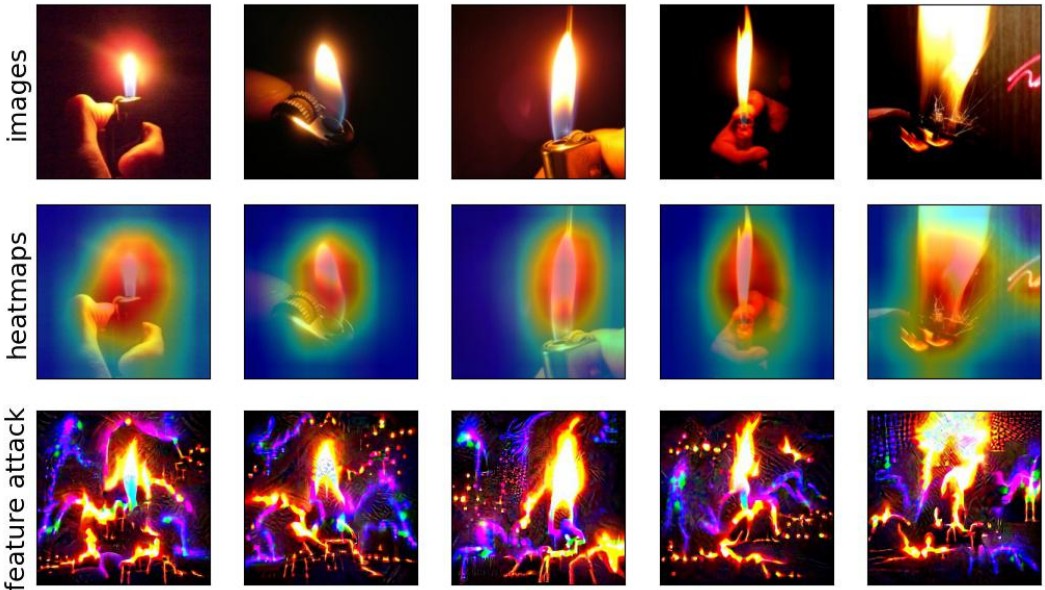

Figure 48: Visualization of feature **1986** for class **lighter** (class index: **626**).
Train accuracy using Standard Resnet-50: **86.154**%.
Drop in accuracy when gaussian noise is added to the highlighted (red) regions: **-35.385**%.

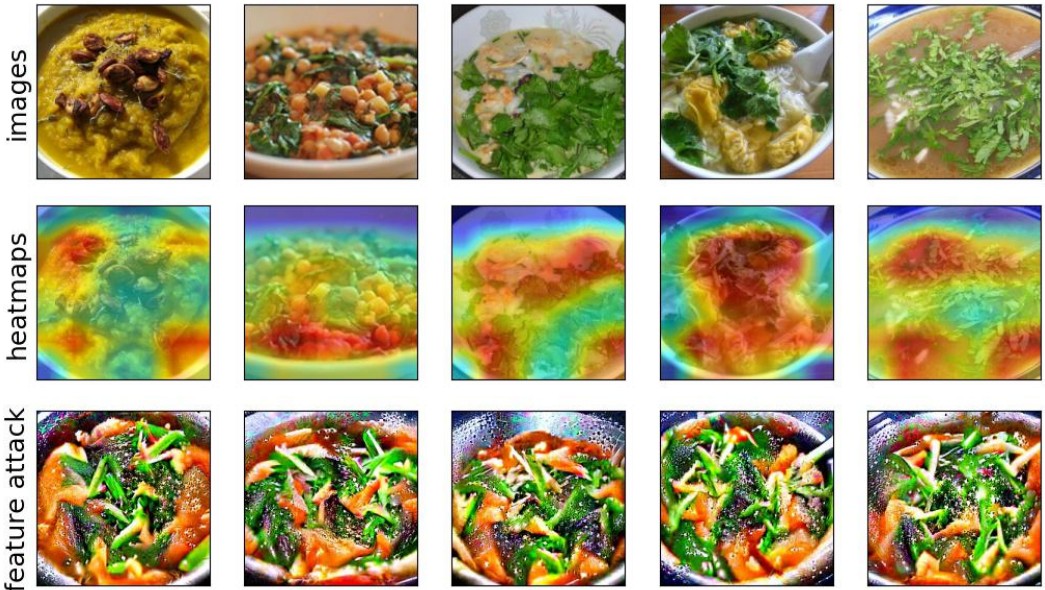

Figure 49: Visualization of feature **895** for class **soup bowl** (class index: **809**).
Train accuracy using Standard Resnet-50: **79.462**%.
Drop in accuracy when gaussian noise is added to the highlighted (red) regions: **-46.154**%.

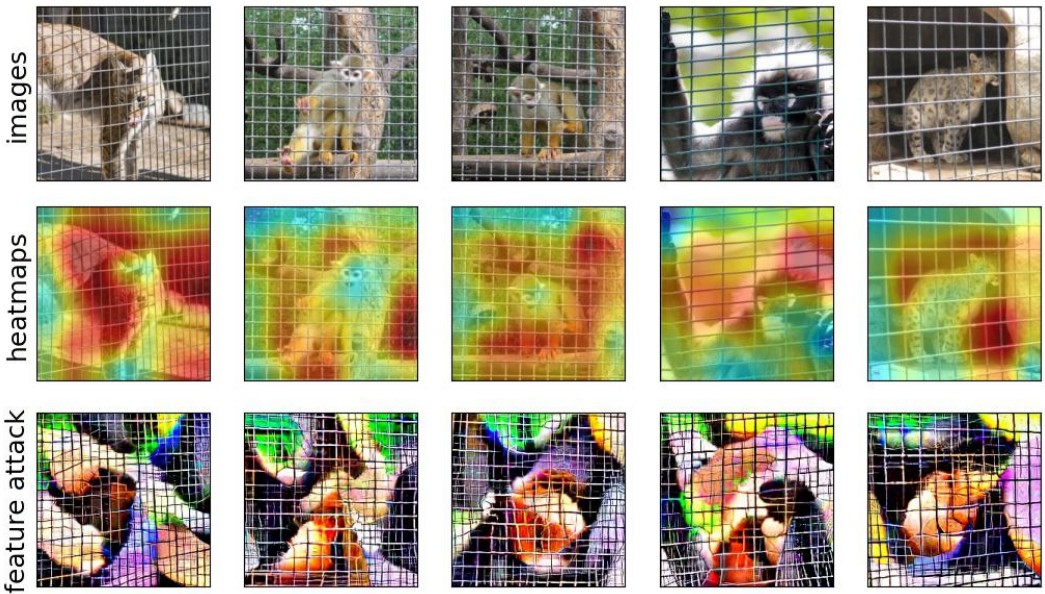

Figure 50: Visualization of feature **798** for class **spider monkey** (class index: **381**).
Train accuracy using Standard Resnet-50: **74.154**%.
Drop in accuracy when gaussian noise is added to the highlighted (red) regions: **-26.154**%.

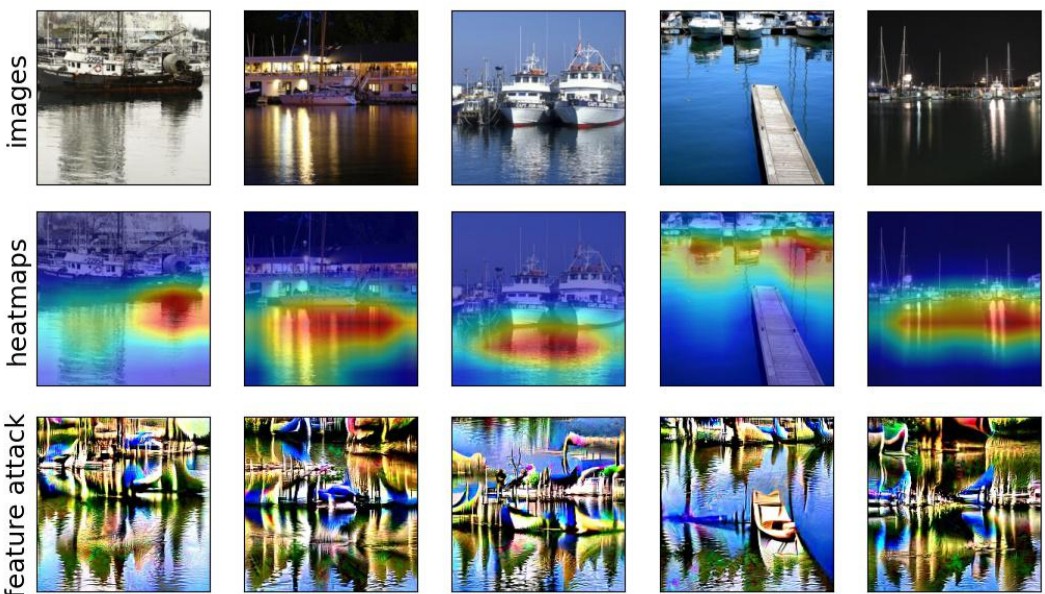

Figure 51: Visualization of feature **516** for class **dock** (class index: **536**).
Train accuracy using Standard Resnet-50: **71.098**%.
Drop in accuracy when gaussian noise is added to the highlighted (red) regions: **-15.384**%.

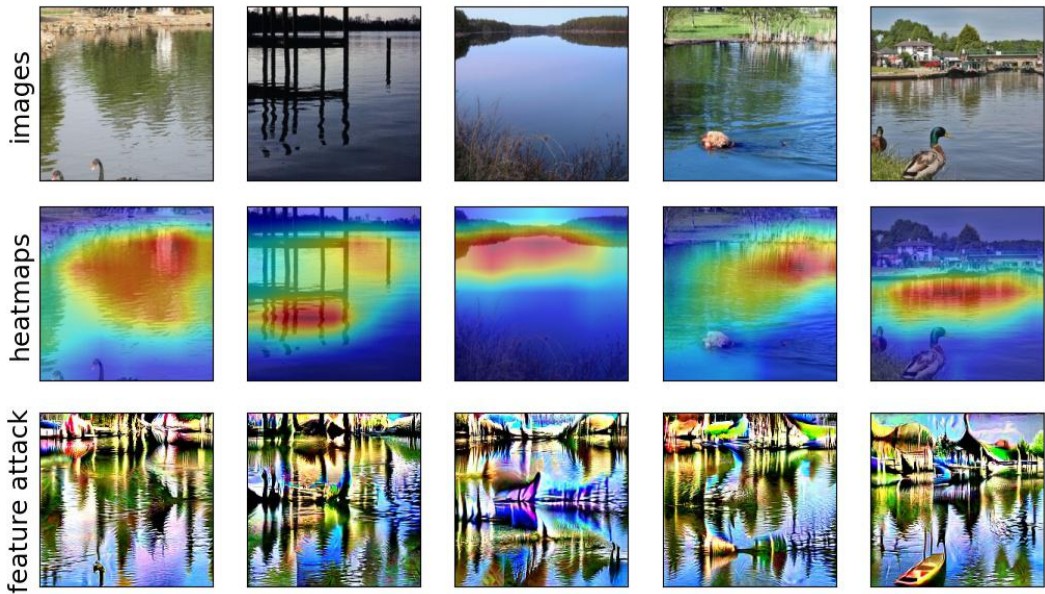

Figure 52: Visualization of feature **516** for class **lakeside** (class index: **975**).
Train accuracy using Standard Resnet-50: **61.769**%.
Drop in accuracy when gaussian noise is added to the highlighted (red) regions: **-32.308**%.

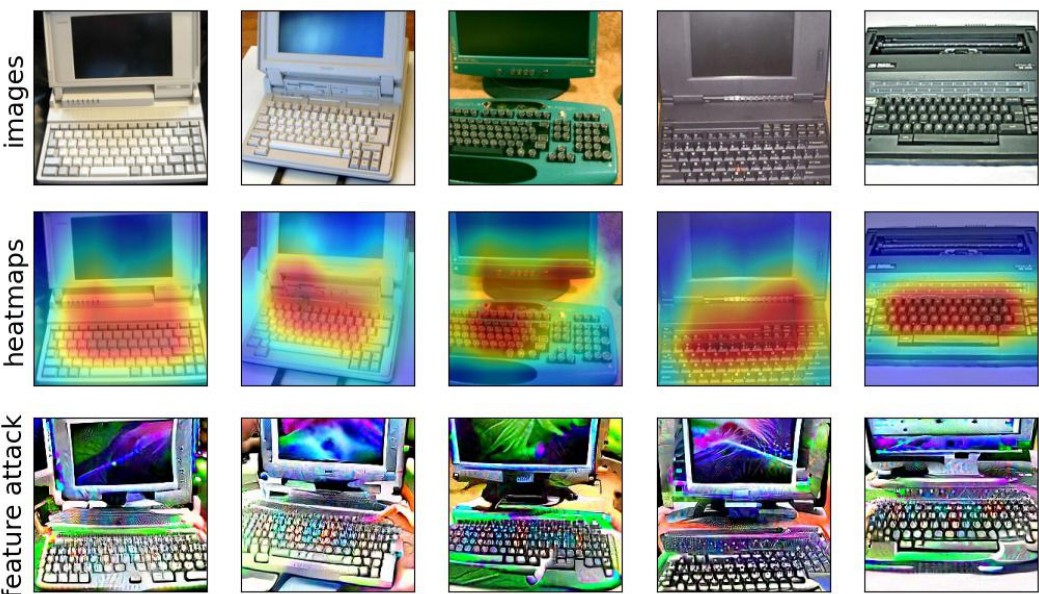

Figure 53: Visualization of feature **1832** for class **space bar** (class index: **810**).
Train accuracy using Standard Resnet-50: **59.043**%.
Drop in accuracy when gaussian noise is added to the highlighted (red) regions: **-36.923**%.

### J.3.3 SPURIOUS FEATURES AT FEATURE RANK 3

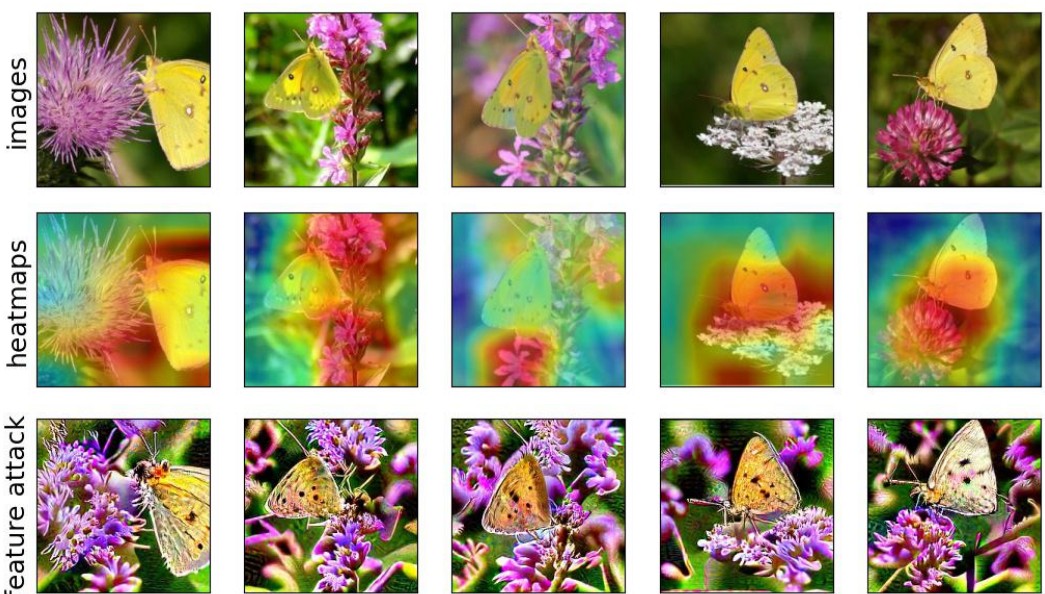

Figure 54: Visualization of feature **1797** for class **sulphur butterfly** (class index: **325**).
Train accuracy using Standard Resnet-50: **95.769**%.
Drop in accuracy when gaussian noise is added to the highlighted (red) regions: **-18.462**%.

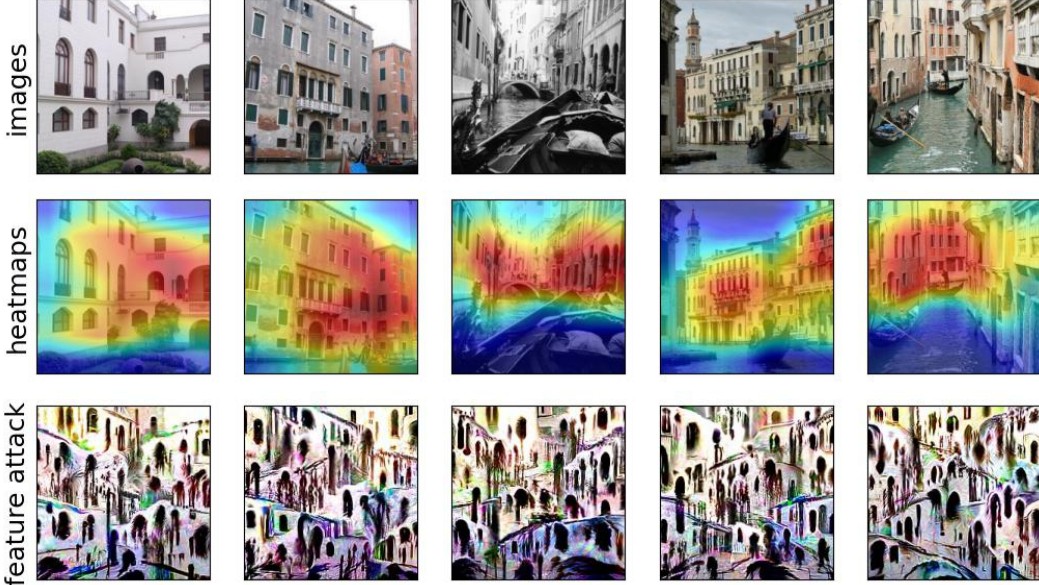

Figure 55: Visualization of feature **481** for class **gondola** (class index: **576**).
Train accuracy using Standard Resnet-50: **98.308**%.
Drop in accuracy when gaussian noise is added to the highlighted (red) regions: **-6.154**%.

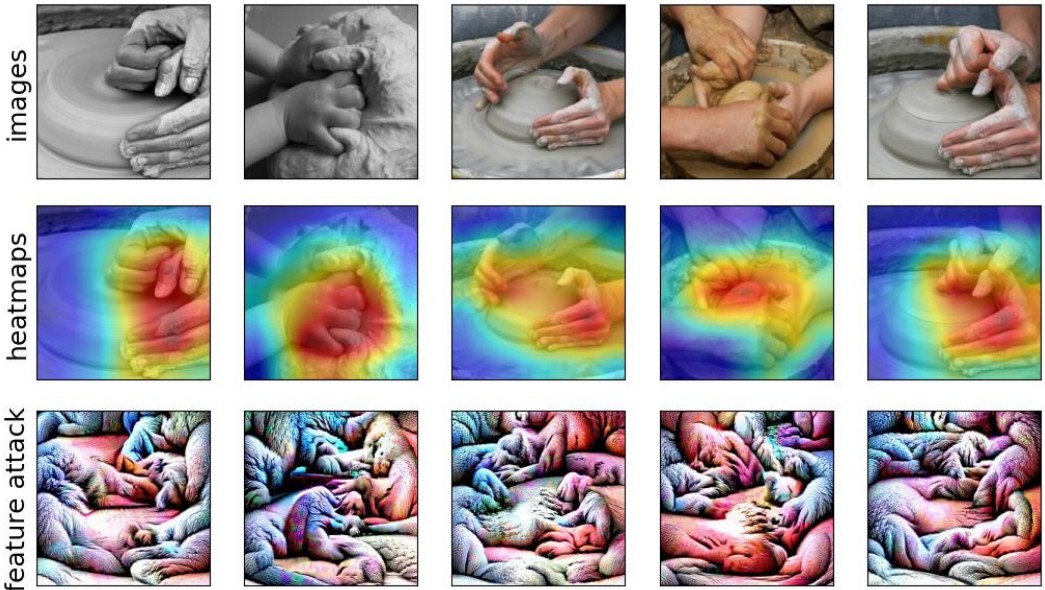

Figure 56: Visualization of feature **432** for class **potter's wheel** (class index: **739**).
Train accuracy using Standard Resnet-50: **95.615**%.
Drop in accuracy when gaussian noise is added to the highlighted (red) regions: **-20.0**%.

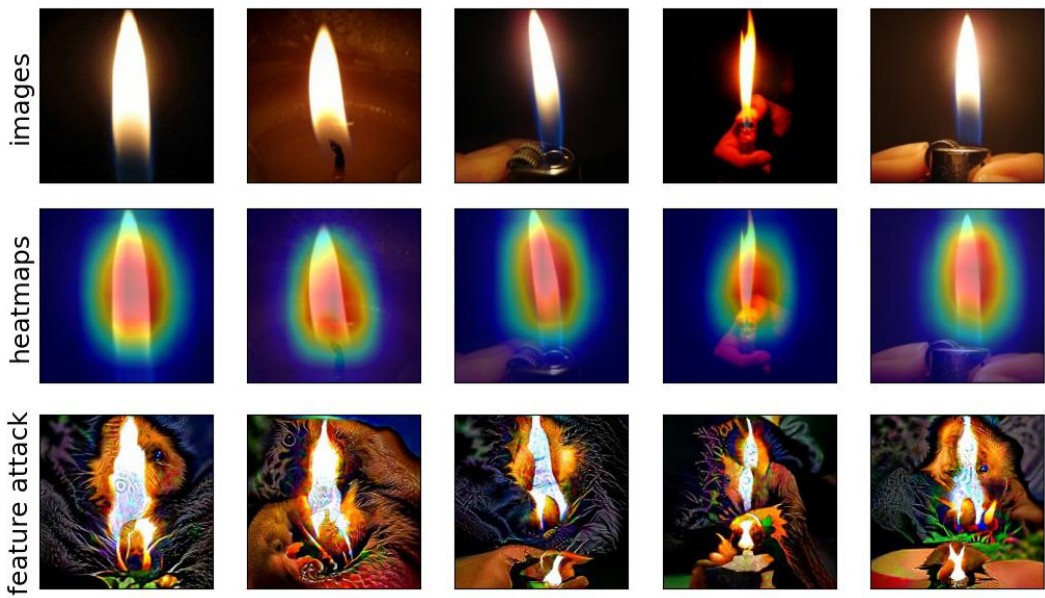

Figure 57: Visualization of feature **1287** for class **lighter** (class index: **626**).
Train accuracy using Standard Resnet-50: **86.154**%.
Drop in accuracy when gaussian noise is added to the highlighted (red) regions: **-10.77**%.

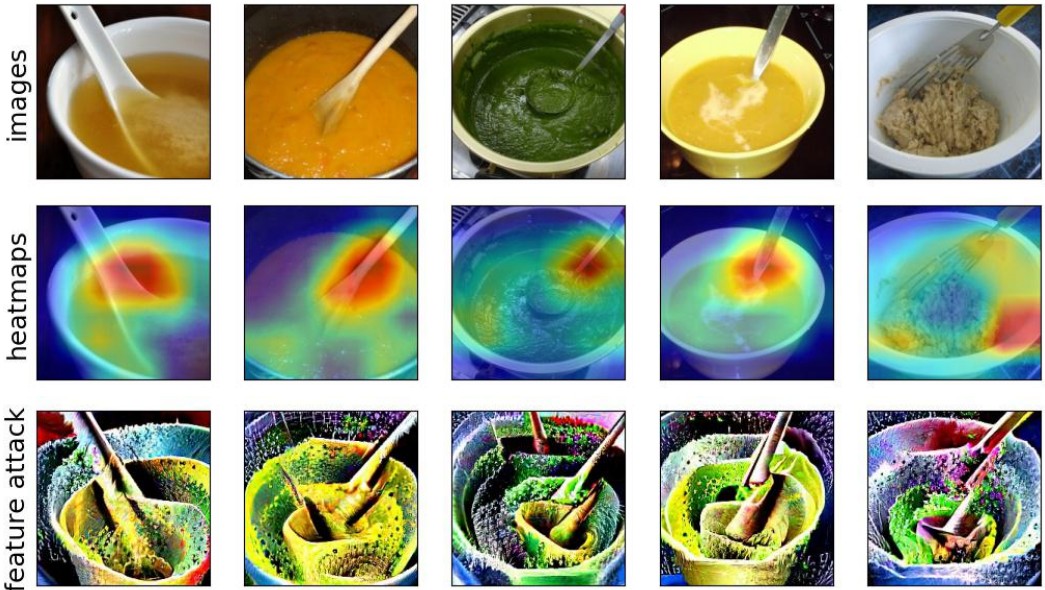

Figure 58: Visualization of feature **840** for class **soup bowl** (class index: **809**).
Train accuracy using Standard Resnet-50: **79.462**%.
Drop in accuracy when gaussian noise is added to the highlighted (red) regions: **-33.846**%.

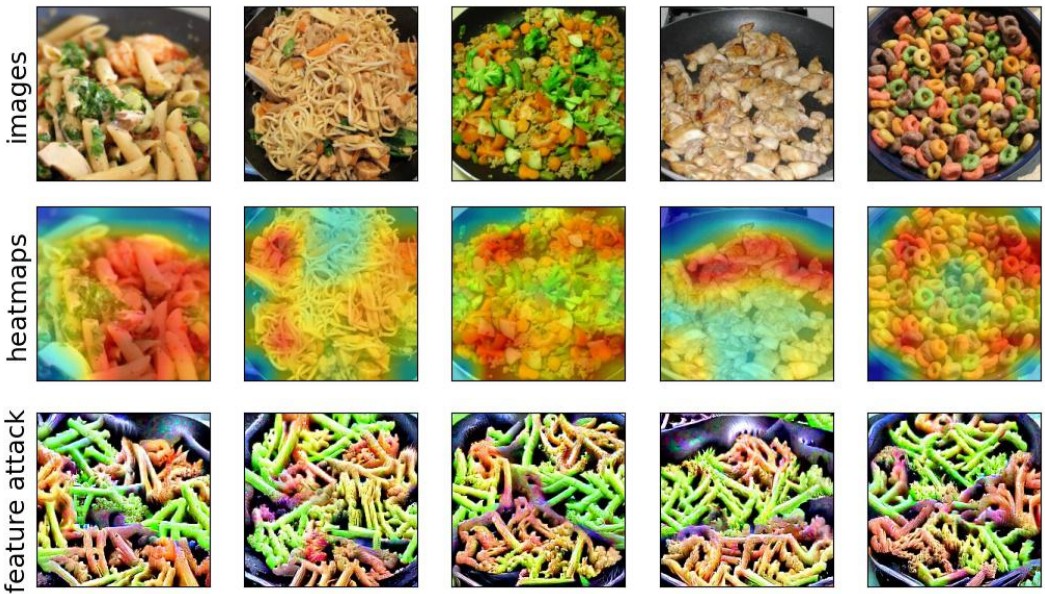

Figure 59: Visualization of feature **2000** for class **wok** (class index: **909**).
Train accuracy using Standard Resnet-50: **71.077**%.
Drop in accuracy when gaussian noise is added to the highlighted (red) regions: **-38.462**%.

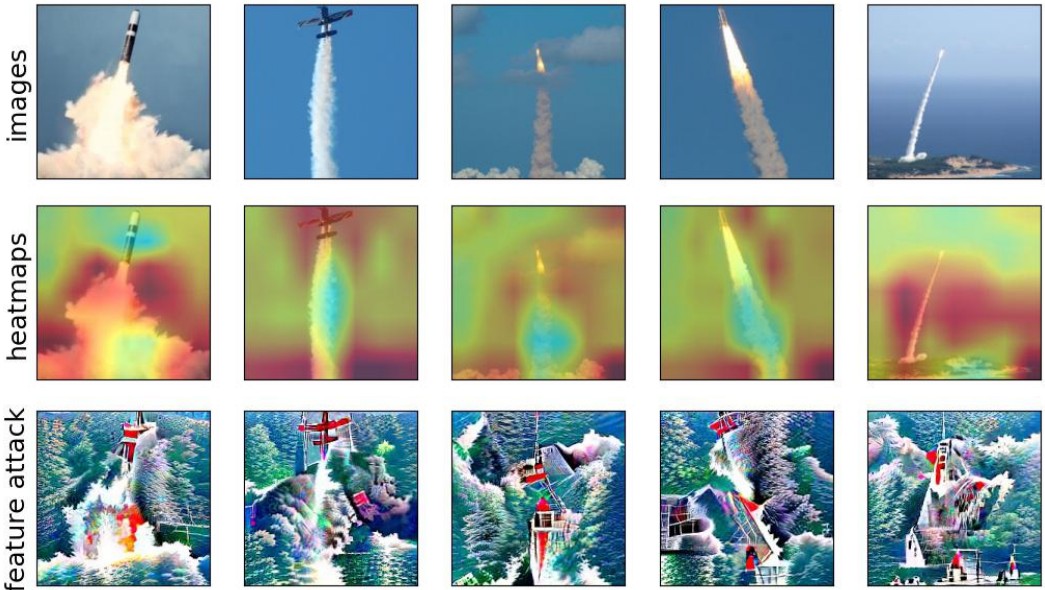

Figure 60: Visualization of feature **961** for class **missile** (class index: **657**).
Train accuracy using Standard Resnet-50: **67.846**%.
Drop in accuracy when gaussian noise is added to the highlighted (red) regions: **-13.846**%.

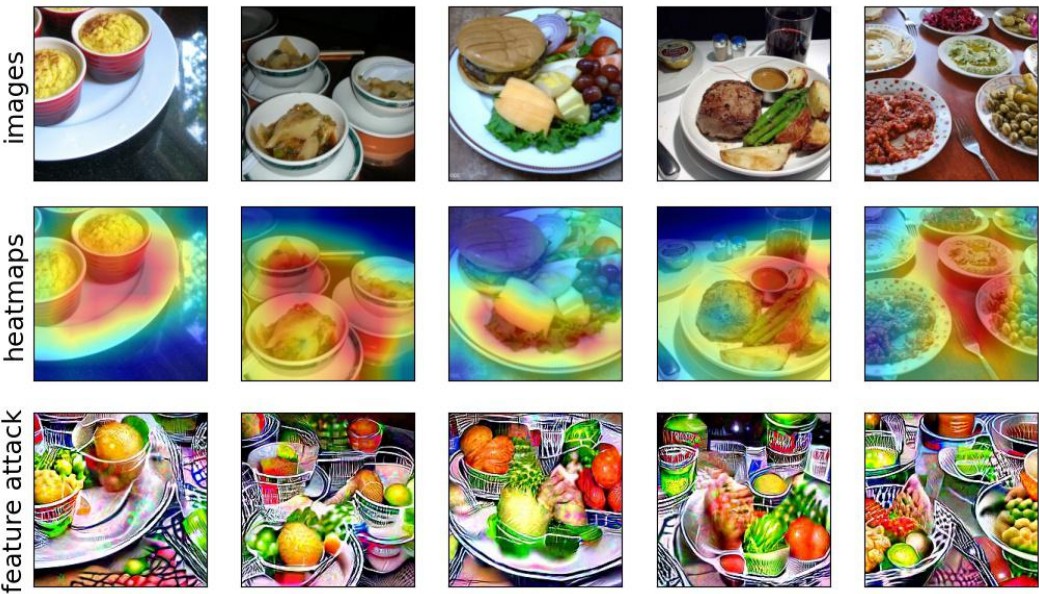

Figure 61: Visualization of feature **2025** for class **plate** (class index: **923**).
Train accuracy using Standard Resnet-50: **64.538**%.
Drop in accuracy when gaussian noise is added to the highlighted (red) regions: **-36.923**%.

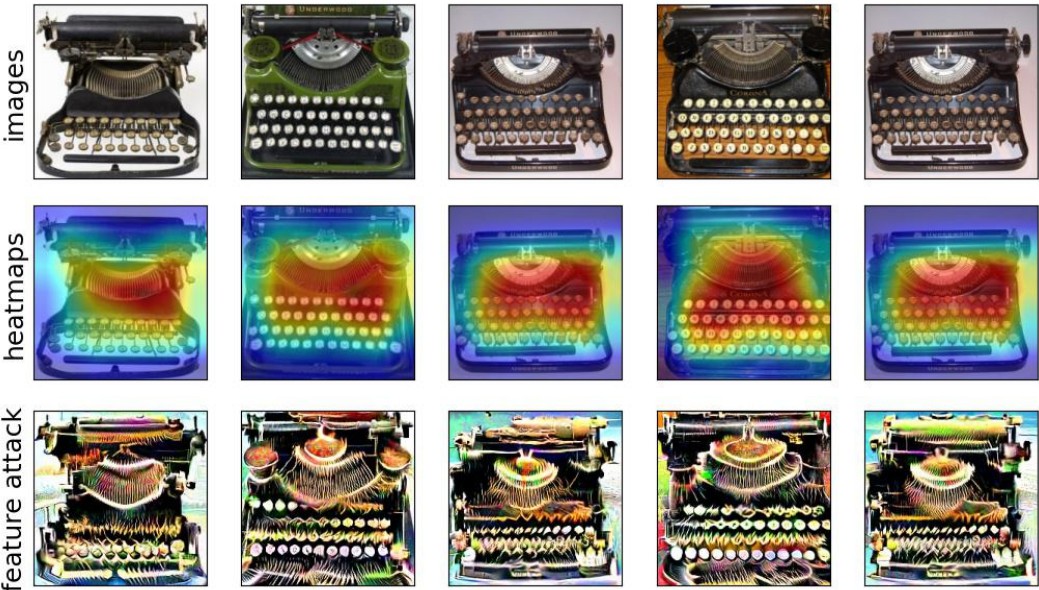

Figure 62: Visualization of feature **387** for class **space bar** (class index: **810**).
Train accuracy using Standard Resnet-50: **59.043**%.
Drop in accuracy when gaussian noise is added to the highlighted (red) regions: **-29.231**%.

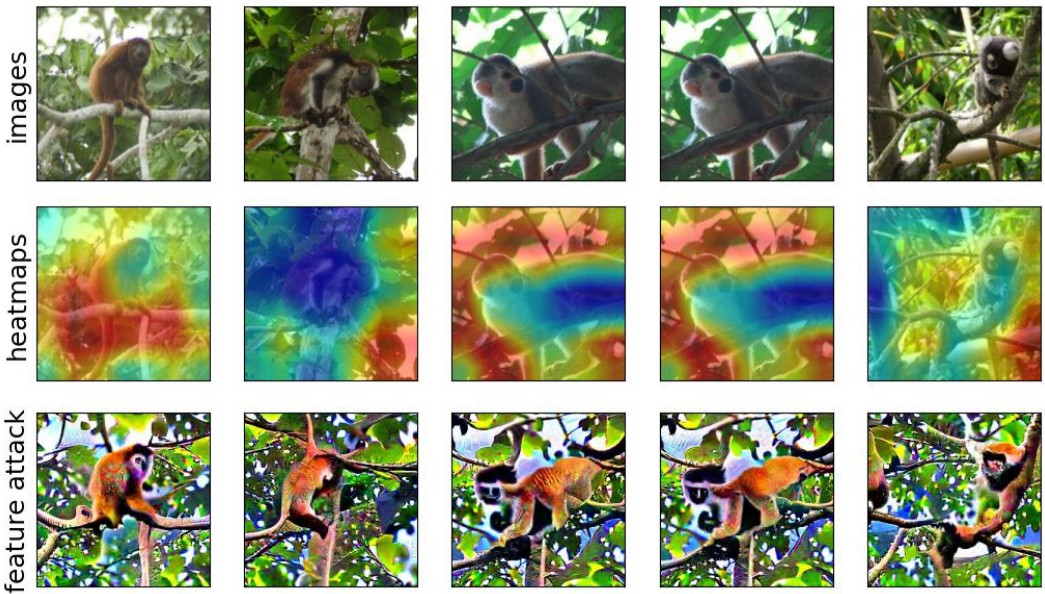

Figure 63: Visualization of feature **1120** for class **titi** (class index: **380**).
Train accuracy using Standard Resnet-50: **58.692**%.
Drop in accuracy when gaussian noise is added to the highlighted (red) regions: **-38.462**%.

### J.3.4 SPURIOUS FEATURES AT FEATURE RANK 4

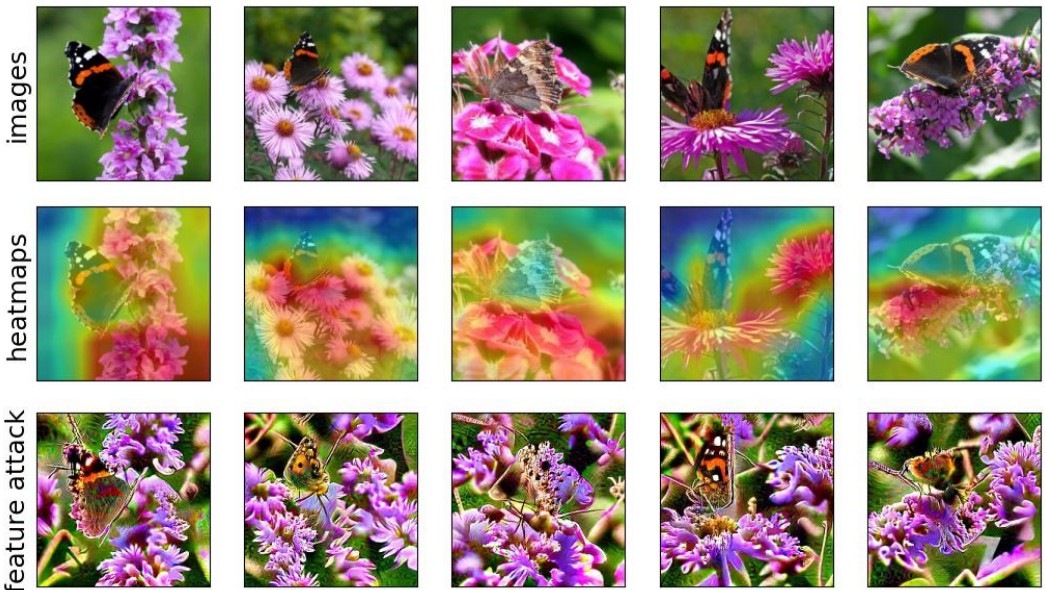

Figure 64: Visualization of feature **1797** for class **admiral** (class index: **321**).
Train accuracy using Standard Resnet-50: **99.846**%.
Drop in accuracy when gaussian noise is added to the highlighted (red) regions: **-3.077**%.

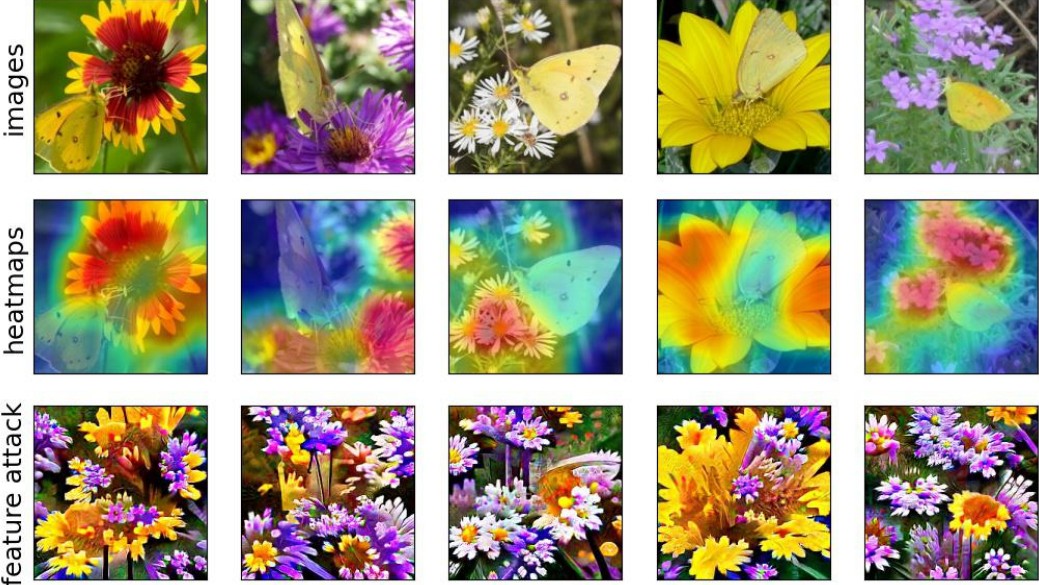

Figure 65: Visualization of feature **595** for class **sulphur butterfly** (class index: **325**).
Train accuracy using Standard Resnet-50: **95.769**%.
Drop in accuracy when gaussian noise is added to the highlighted (red) regions: **-21.539**%.

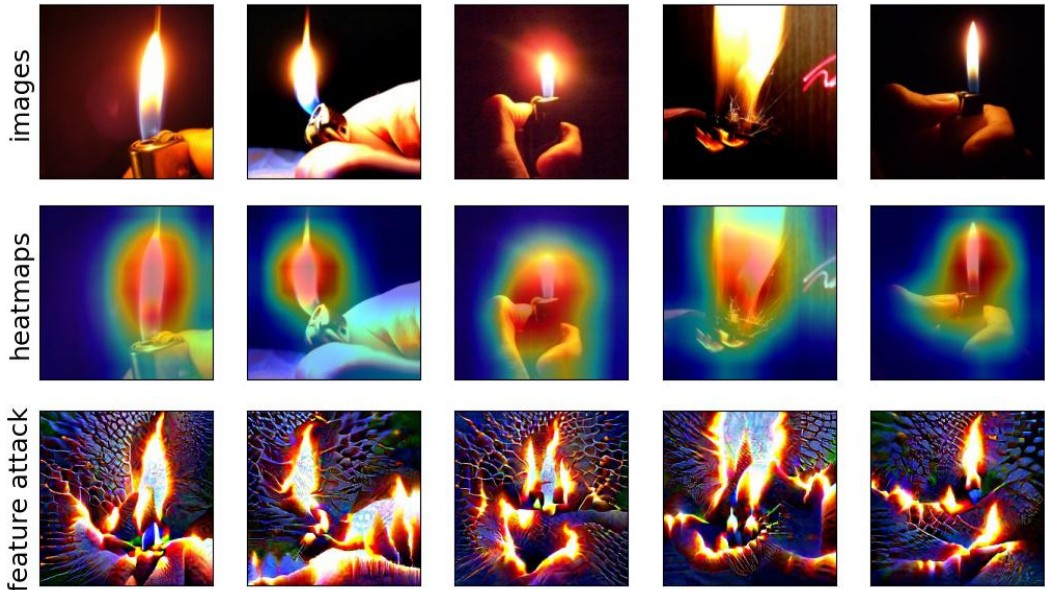

Figure 66: Visualization of feature **1642** for class **lighter** (class index: **626**).
Train accuracy using Standard Resnet-50: **86.154**%.
Drop in accuracy when gaussian noise is added to the highlighted (red) regions: **-27.693**%.

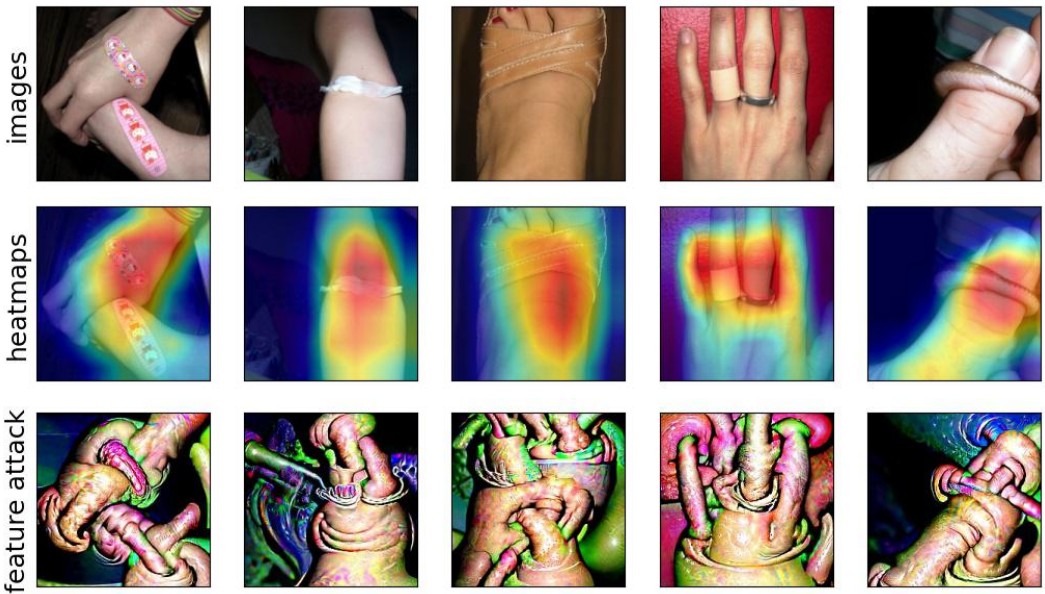

Figure 67: Visualization of feature **447** for class **band aid** (class index: **419**).
Train accuracy using Standard Resnet-50: **81.000**%.
Drop in accuracy when gaussian noise is added to the highlighted (red) regions: **-43.077**%.

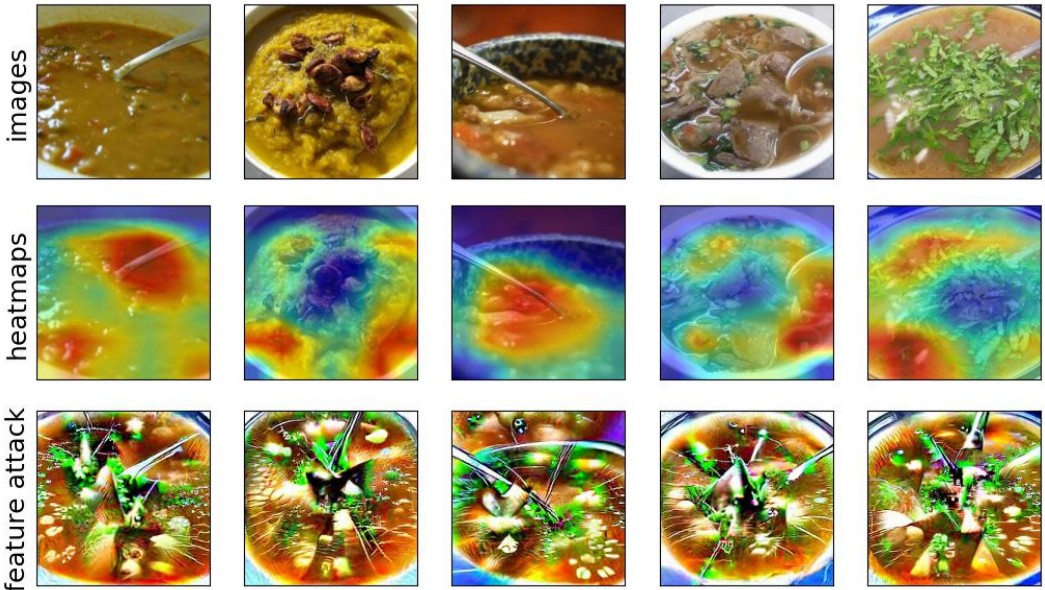

Figure 68: Visualization of feature **1296** for class **soup bowl** (class index: **809**).
Train accuracy using Standard Resnet-50: **79.462**%.
Drop in accuracy when gaussian noise is added to the highlighted (red) regions: **-49.23**%.

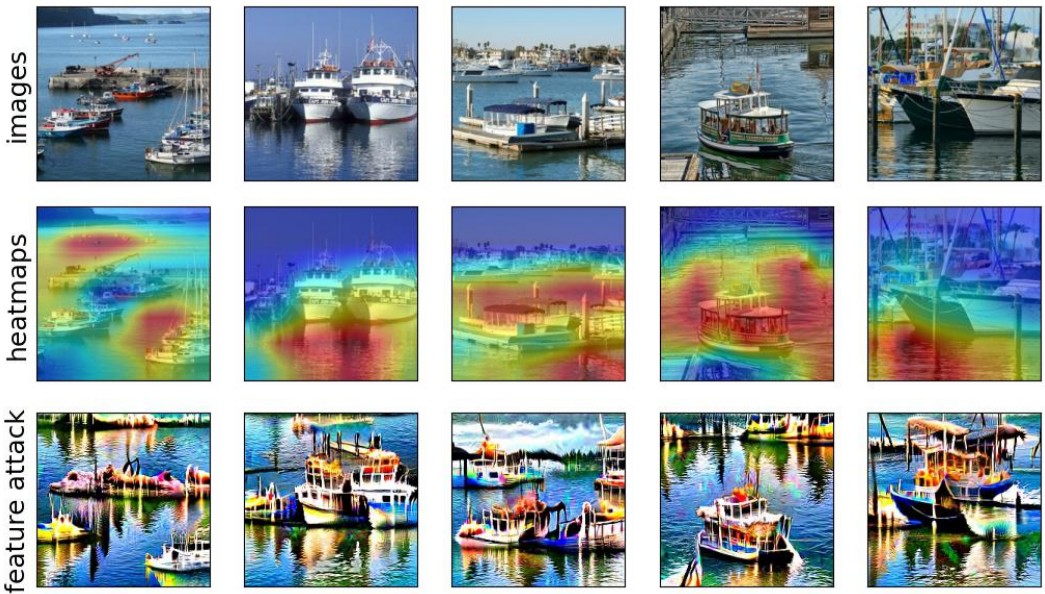

Figure 69: Visualization of feature **76** for class **dock** (class index: **536**).
Train accuracy using Standard Resnet-50: **71.098**%.
Drop in accuracy when gaussian noise is added to the highlighted (red) regions: **-24.616**%.

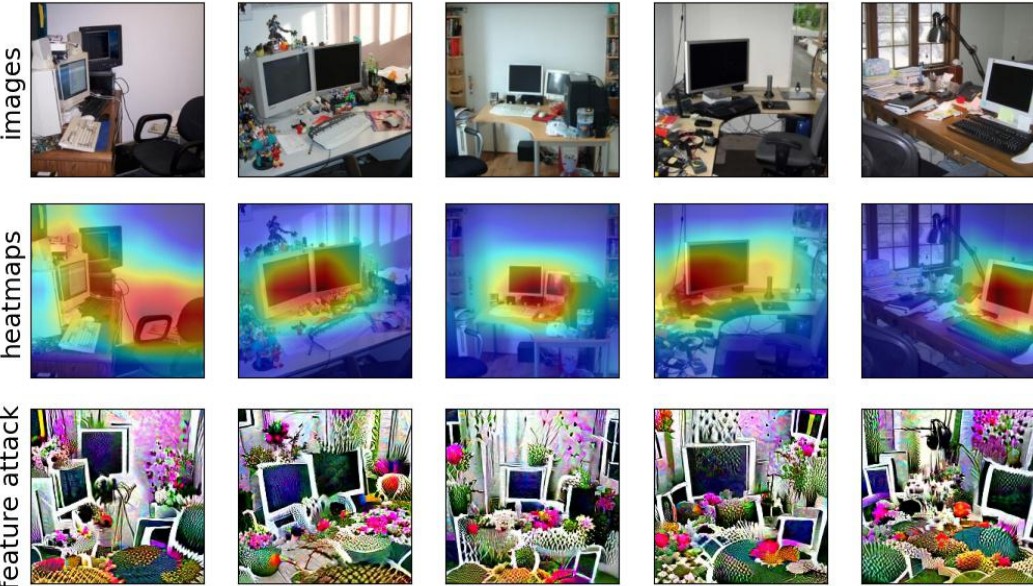

Figure 70: Visualization of feature **2** for class **desk** (class index: **526**).
Train accuracy using Standard Resnet-50: **69.769**%.
Drop in accuracy when gaussian noise is added to the highlighted (red) regions: **-16.923**%.

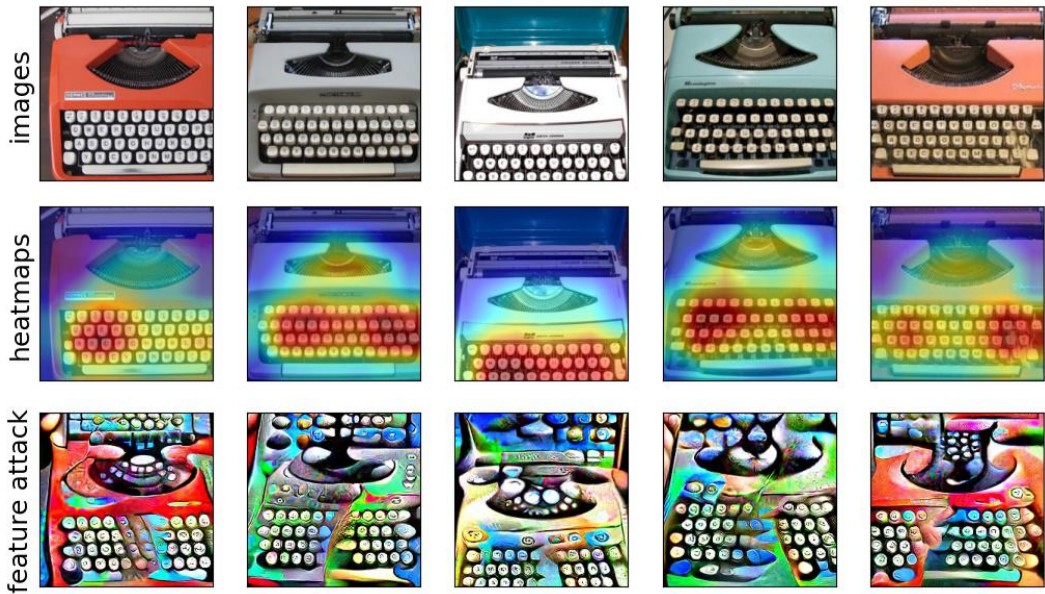

Figure 71: Visualization of feature **1469** for class **space bar** (class index: **810**).
Train accuracy using Standard Resnet-50: **59.043**%.
Drop in accuracy when gaussian noise is added to the highlighted (red) regions: **-15.384**%.

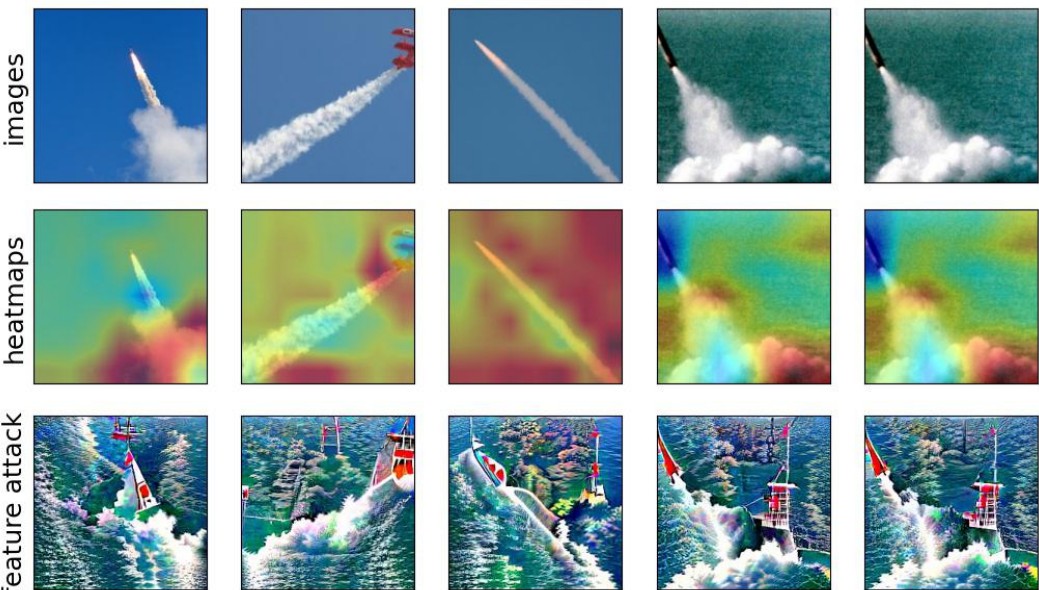

Figure 72: Visualization of feature **961** for class **projectile** (class index: **744**).
Train accuracy using Standard Resnet-50: **53.538**%.
Drop in accuracy when gaussian noise is added to the highlighted (red) regions: **-23.077**%.

### J.3.5 SPURIOUS FEATURES AT FEATURE RANK 5

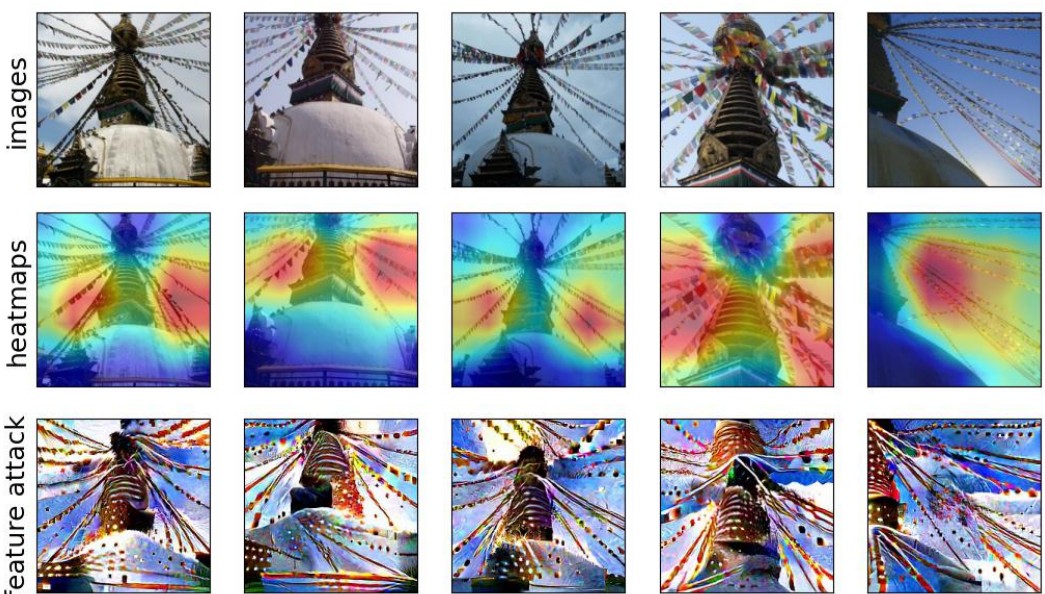

Figure 73: Visualization of feature **36** for class **stupa** (class index: **832**).
Train accuracy using Standard Resnet-50: **98.385**%.
Drop in accuracy when gaussian noise is added to the highlighted (red) regions: **-9.231**%.

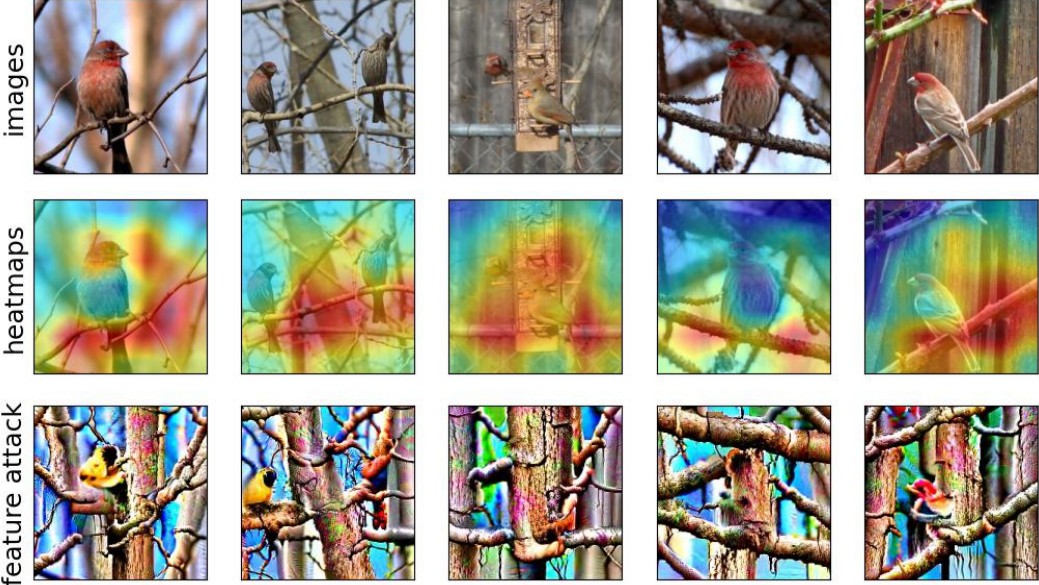

Figure 74: Visualization of feature **371** for class **house finch** (class index: **12**).
Train accuracy using Standard Resnet-50: **98.308**%.
Drop in accuracy when gaussian noise is added to the highlighted (red) regions: **-24.615**%.

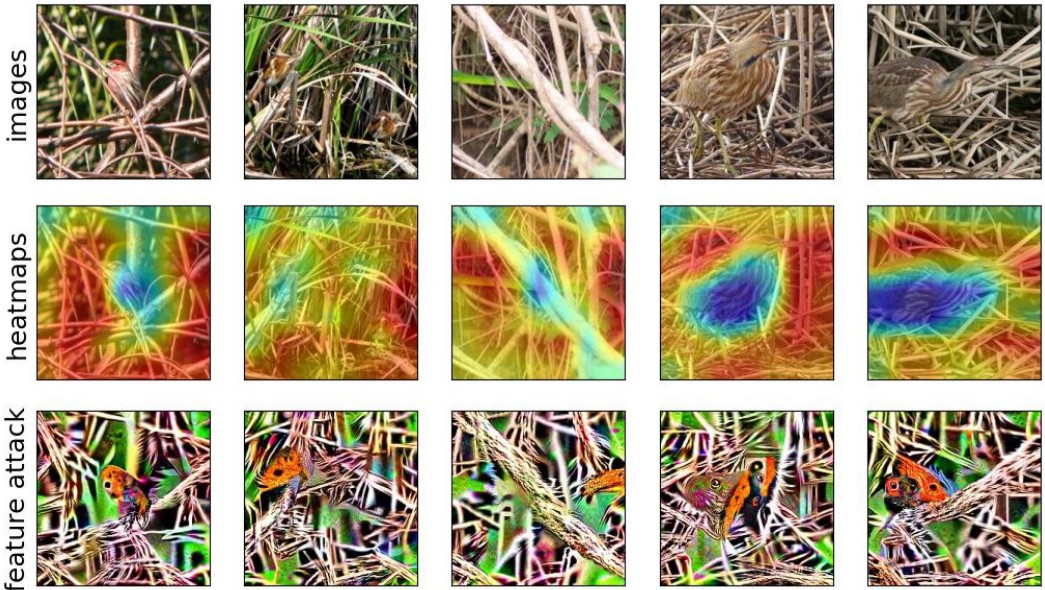

Figure 75: Visualization of feature **1556** for class **bittern** (class index: **133**).
Train accuracy using Standard Resnet-50: **97.077**%.
Drop in accuracy when gaussian noise is added to the highlighted (red) regions: **-10.769**%.

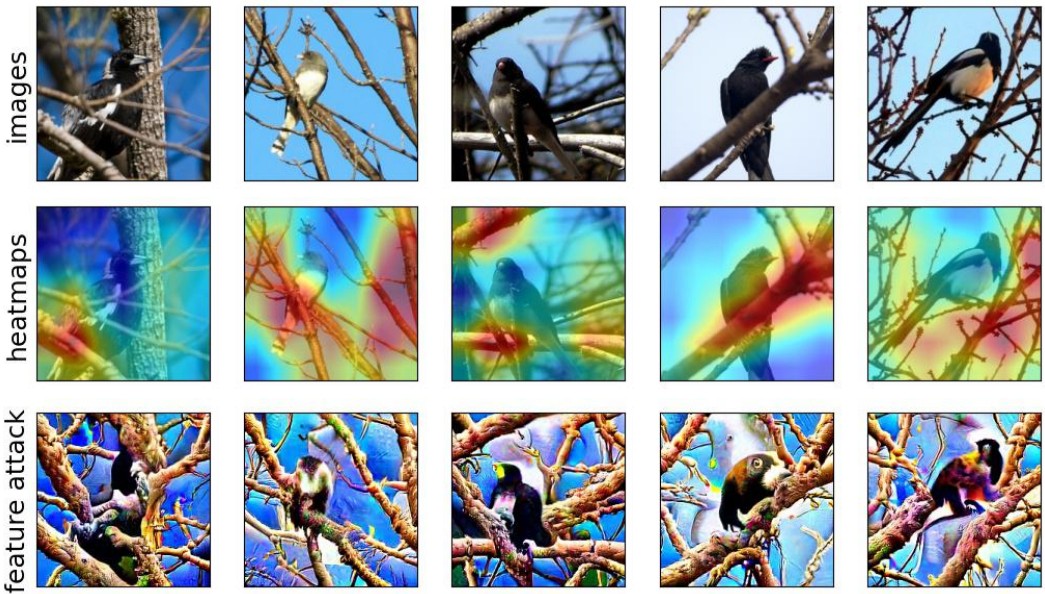

Figure 76: Visualization of feature **1541** for class **junco** (class index: **13**).
Train accuracy using Standard Resnet-50: **97.000**%.
Drop in accuracy when gaussian noise is added to the highlighted (red) regions: **-9.231**%.

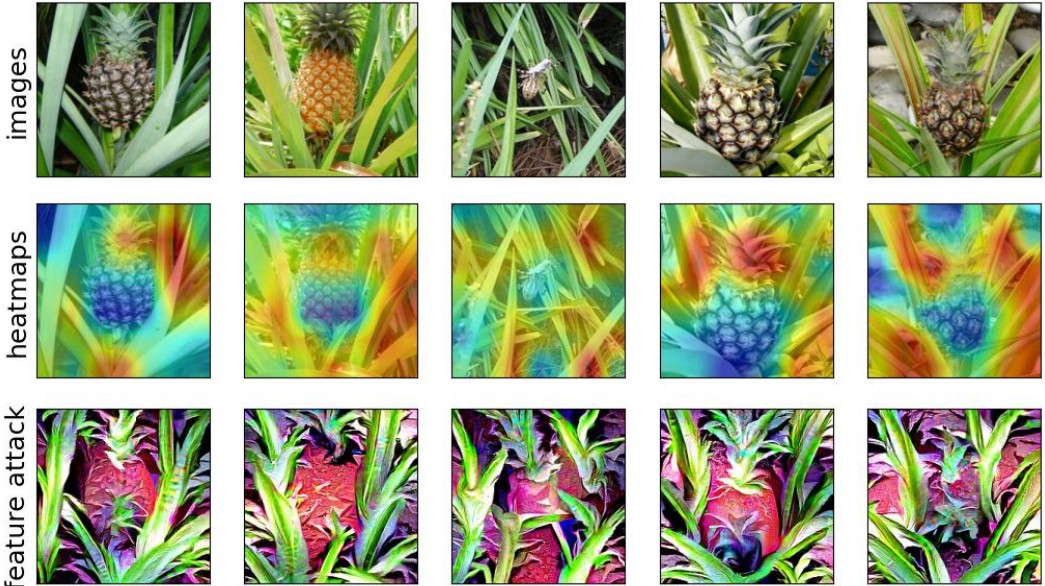

Figure 77: Visualization of feature **96** for class **pineapple** (class index: **953**).
Train accuracy using Standard Resnet-50: **94.538**%.
Drop in accuracy when gaussian noise is added to the highlighted (red) regions: **-1.539**%.

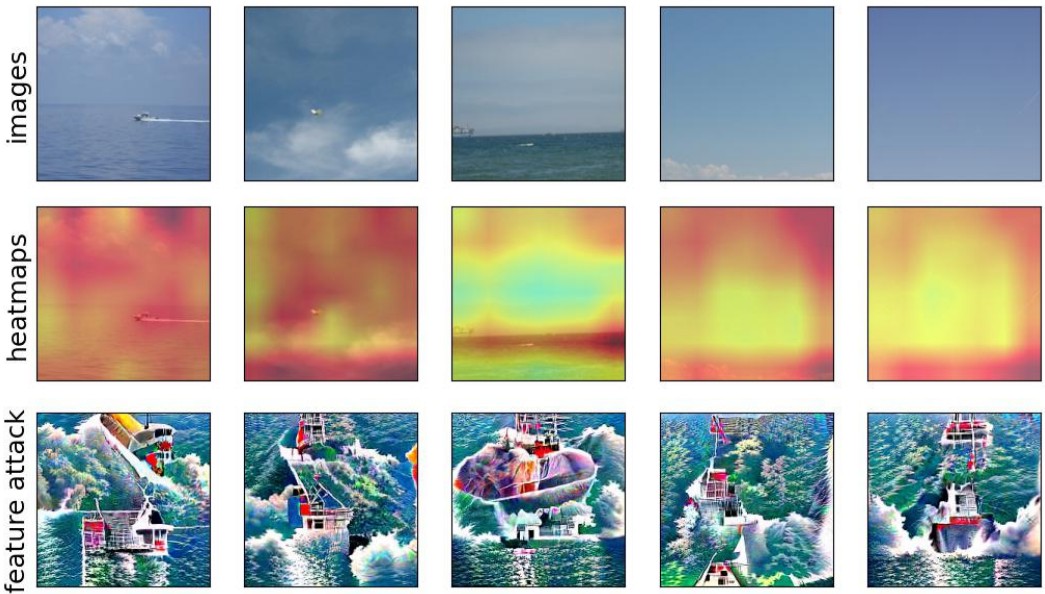

Figure 78: Visualization of feature **961** for class **sandbar** (class index: **977**).
Train accuracy using Standard Resnet-50: **71.923**%.
Drop in accuracy when gaussian noise is added to the highlighted (red) regions: **-60.0**%.

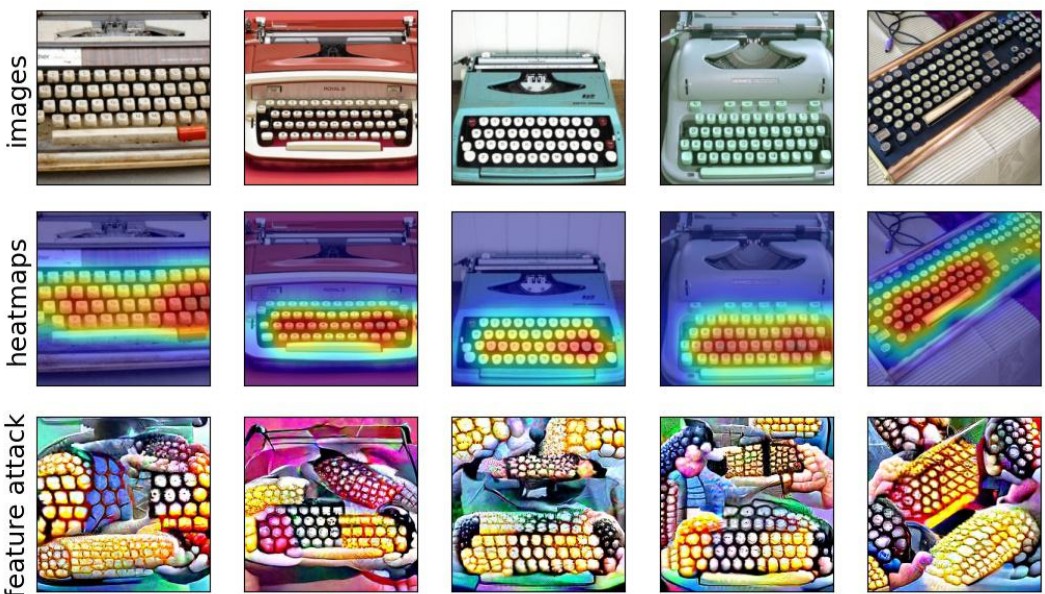

Figure 79: Visualization of feature **510** for class **space bar** (class index: **810**).
Train accuracy using Standard Resnet-50: **59.043**%.
Drop in accuracy when gaussian noise is added to the highlighted (red) regions: **-24.615**%.

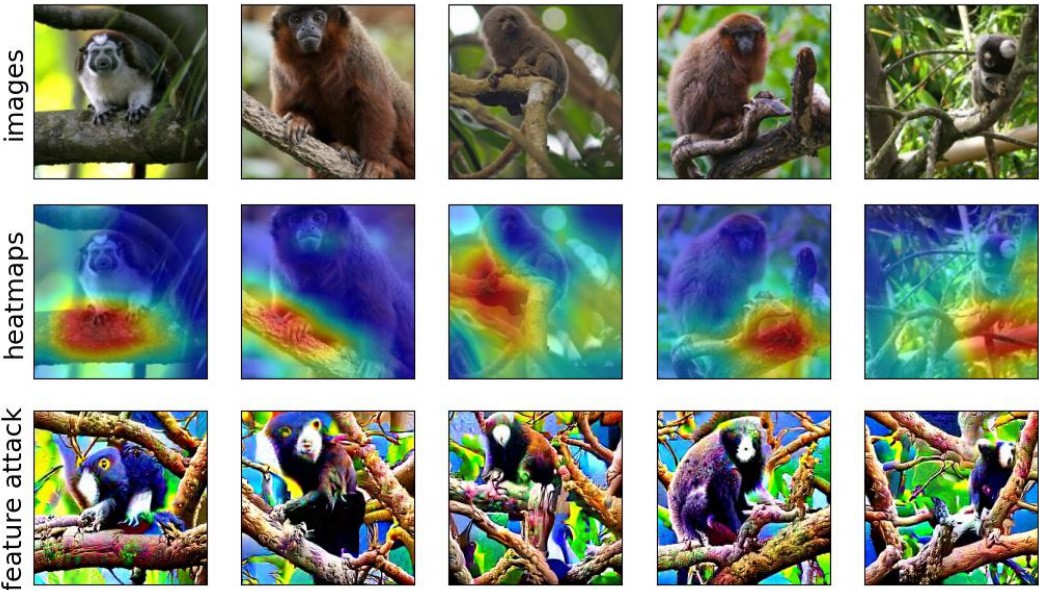

Figure 80: Visualization of feature **1541** for class **titi** (class index: **380**).
Train accuracy using Standard Resnet-50: **58.692**%.
Drop in accuracy when gaussian noise is added to the highlighted (red) regions: **-30.769**%.

# K  DIAGNOSING THE CONFUSING VISUAL ATTRIBUTE BETWEEN DIFFERENT CLASSES

## K.1  FEATURE INDEX: 1541

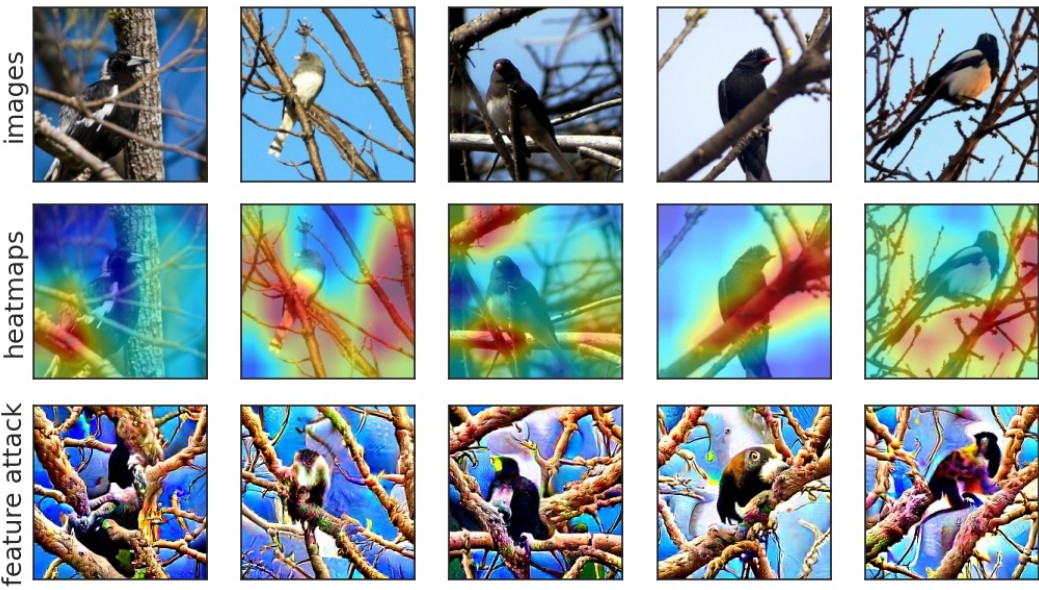

Figure 81: Visualization for class **junco** (class index: **13**).

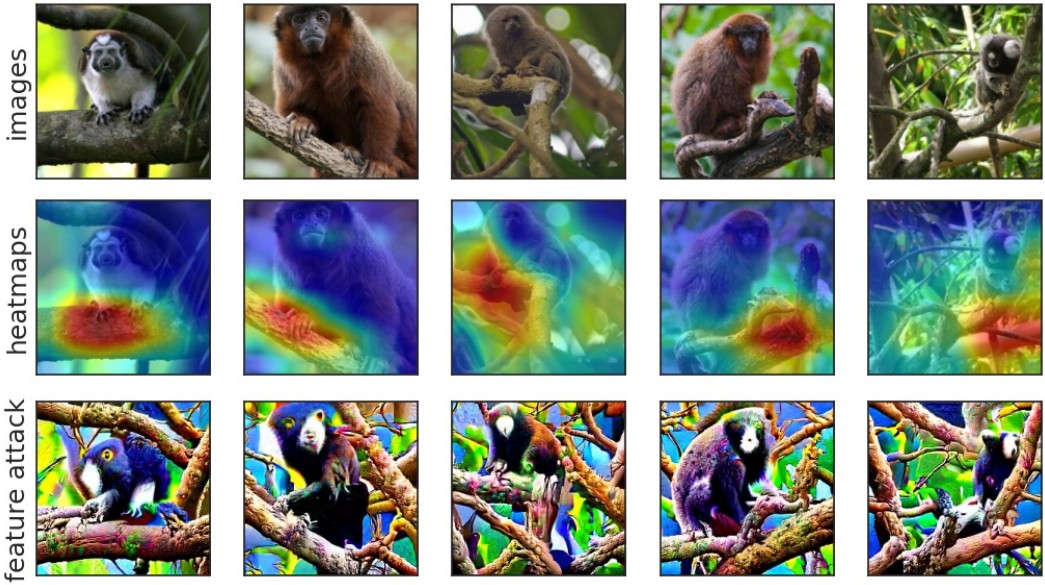

Figure 82: Visualization for class **titi** (class index: **380**).

### K.2 FEATURE INDEX: 2025

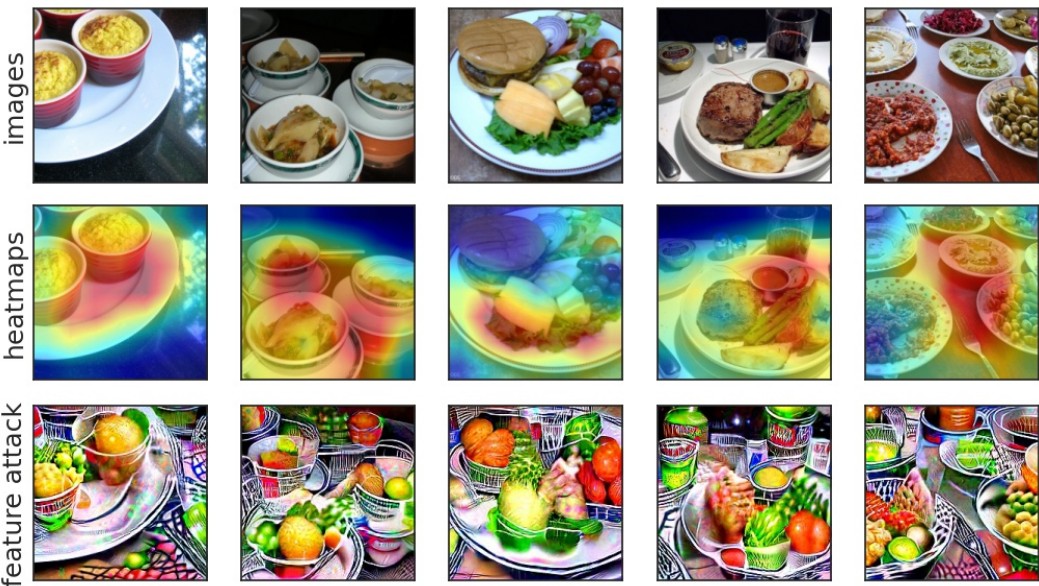

Figure 83: Visualization for class **plate** (class index: **923**).

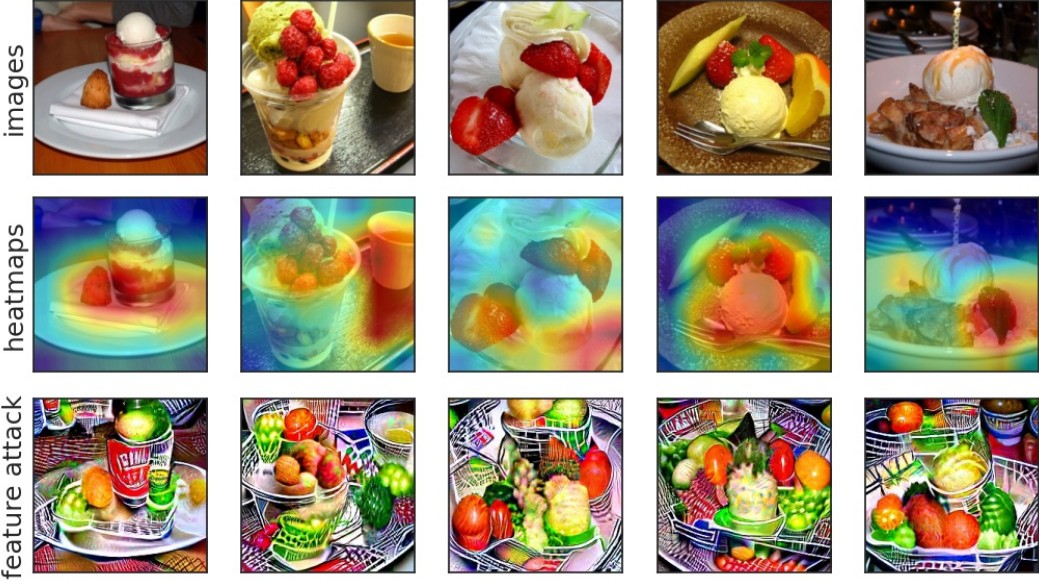

Figure 84: Visualization for class **icecream** (class index: **928**).

## L   COMPARING BETWEEN THE ACCURACY DROPS DUE TO SPURIOUS AND CORE NEURAL FEATURES

For some class $i$, we want to test the sensitivity of a trained model to the visual attribute encoded in the feature $j$ for predicting the class $i$. To this end, we add Gaussian noise to highly activating regions for images in the dataset $\mathcal{D}(i, j)$ and evaluate the drop in the accuracy of the model being inspected for failure. In particular, we show the drop in the accuracy for different classes by perturbing both core and spurious neural features using Gaussian noise with $\sigma = 0.25$.

However, for some of the classes, given a $\sigma$ value, the mean of noise perturbation magnitudes imposed on images in the dataset $\mathcal{D}(i, j)$ (referred to as "mean $\ell_2$ perturbation") can be different for different features. This is because the activation maps of different neural features can have different $l_2$ norms. Thus, for such classes, we also compute accuracy drops using multiple values of $\sigma$: $[0.30, 0.35, \dots, 0.60]$. Then, for the features whose mean $l_2$ perturbations were smaller, we choose the smallest value of $\sigma$ (referred to as $\sigma_s$) that results in a mean $l_2$ perturbation similar to (or even slightly higher than) that of the other neural feature (that initially had higher mean $l_2$ perturbation). Thus, for each neural feature, we show the accuracy drops using both $\sigma = 0.25$ and $\sigma_s$ in the figure captions in this section.

## L.1  CLASS NAME: OSTRICH, TRAIN ACCURACY (STANDARD RESNET-50): 98.615%

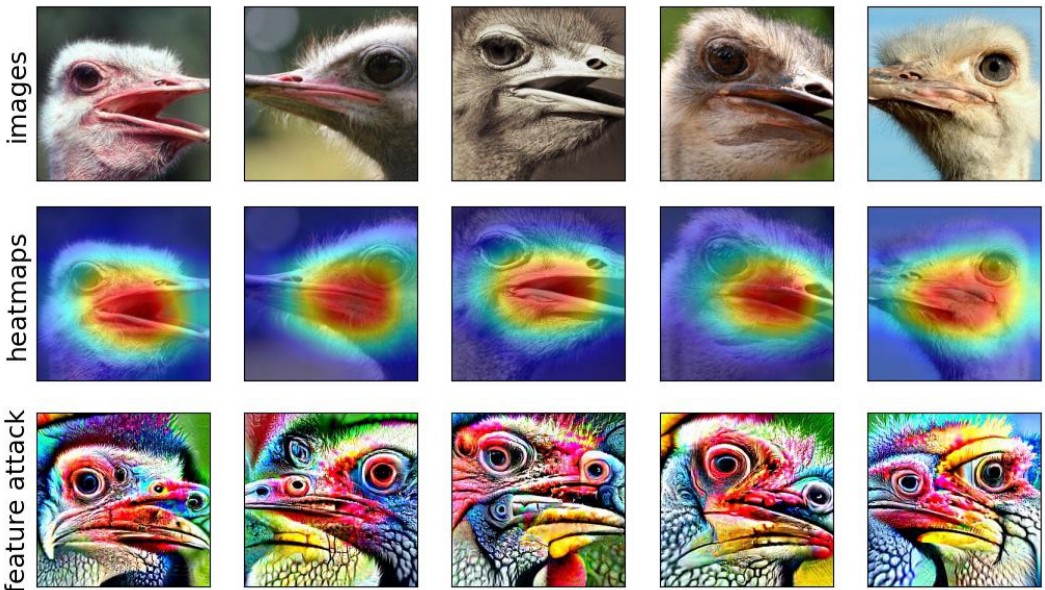

Figure 85: Visualization of feature **1964** for class **ostrich** (class index: **9**).
Feature label as annotated by Mechanical Turk workers: **core**.
Using $\sigma = 0.25$, drop in accuracy: **0.0**%, mean $l_2$ perturbation: **35.782**.
Using $\sigma = 0.35$, drop in accuracy: **-6.154**%, mean $l_2$ perturbation: **47.815**.

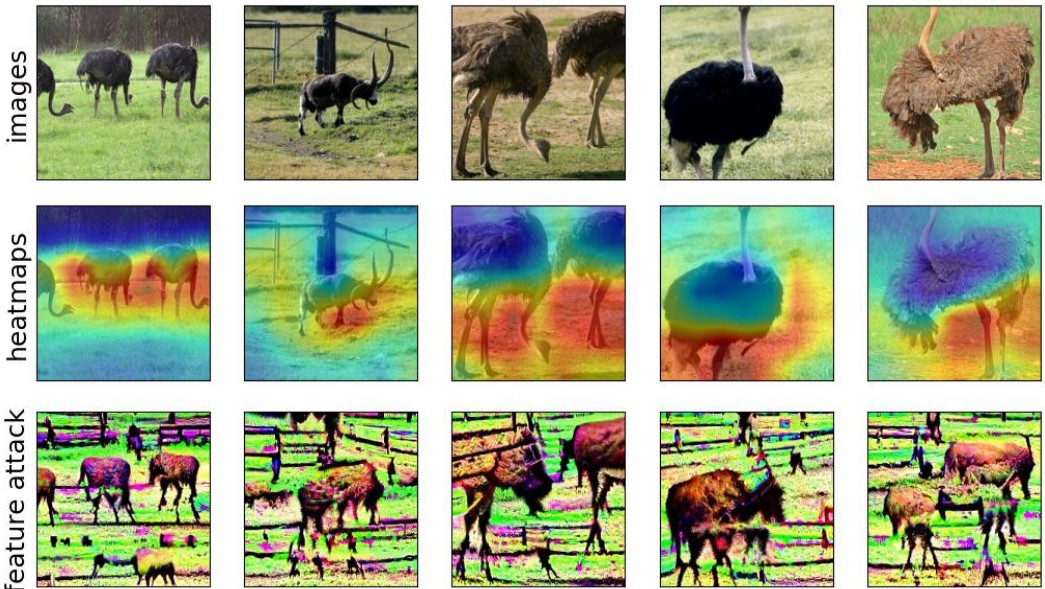

Figure 86: Visualization of feature **63** for class **ostrich** (class index: **9**).
Feature label as annotated by Mechanical Turk workers: **spurious**.
Using $\sigma = 0.25$, drop in accuracy: **-20.0**%, mean $l_2$ perturbation: **47.653**.
Using $\sigma = 0.35$, drop in accuracy: **-32.308**%, mean $l_2$ perturbation: **63.911**.

## L.2 CLASS NAME: BRAMBLING, TRAIN ACCURACY (STANDARD RESNET-50): 95.615%

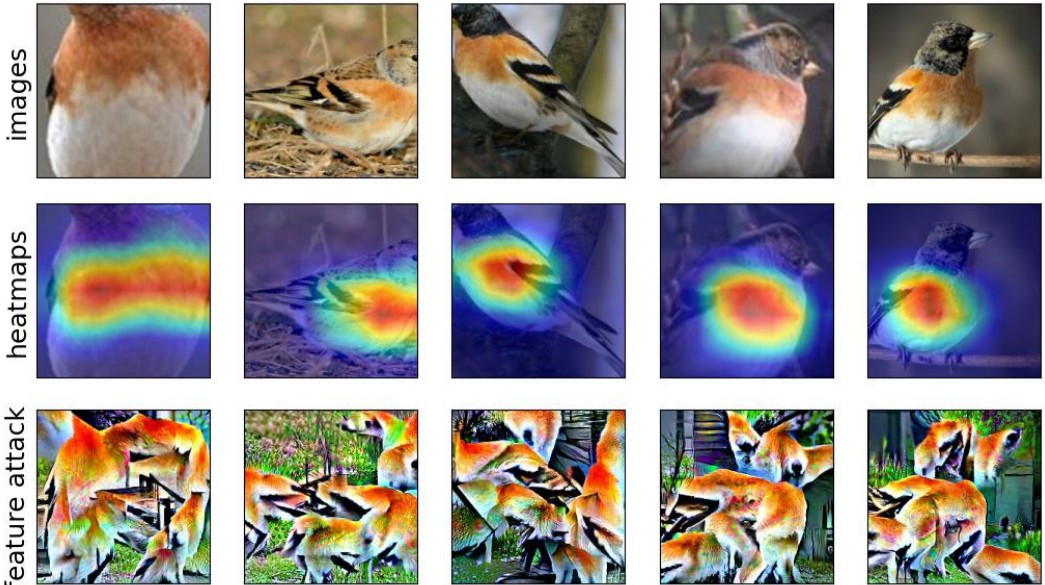

Figure 87: Visualization of feature **457** for class **brambling** (class index: **10**).
Feature label as annotated by Mechanical Turk workers: **core**.
Using $\sigma = 0.25$, drop in accuracy: **-1.538**%, mean $l_2$ perturbation: **25.337**.
Using $\sigma = 0.6$, drop in accuracy: **-21.538**%, mean $l_2$ perturbation: **51.371**.

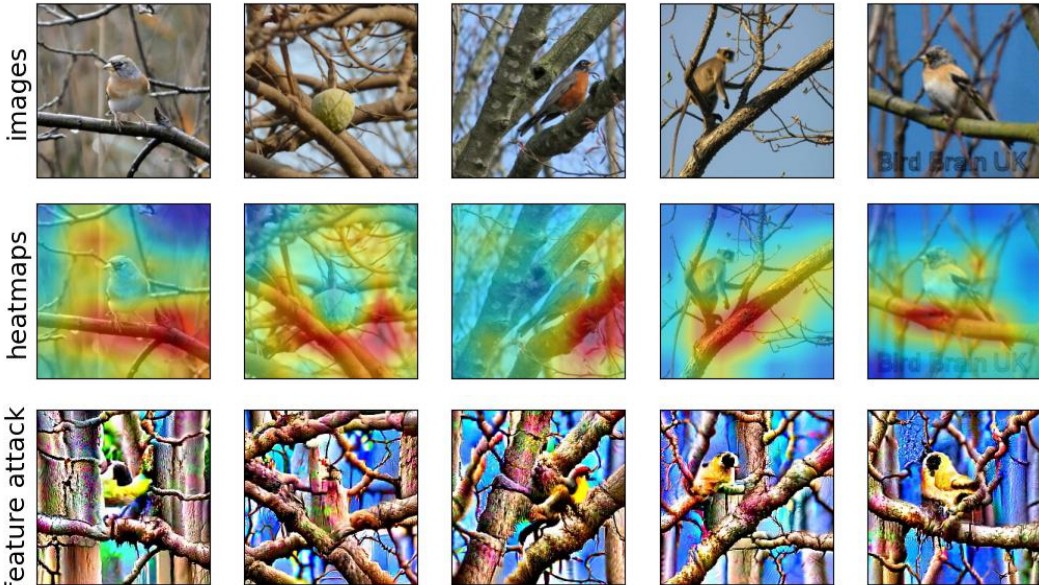

Figure 88: Visualization of feature **371** for class **brambling** (class index: **10**).
Feature label as annotated by Mechanical Turk workers: **spurious**.
Using $\sigma = 0.25$, drop in accuracy: **-33.846**%, mean $l_2$ perturbation: **50.046**.
Using $\sigma = 0.6$, drop in accuracy: **-93.846**%, mean $l_2$ perturbation: **100.689**.

L.3   CLASS NAME: HOUSE FINCH, TRAIN ACCURACY (STANDARD RESNET-50): 98.308%

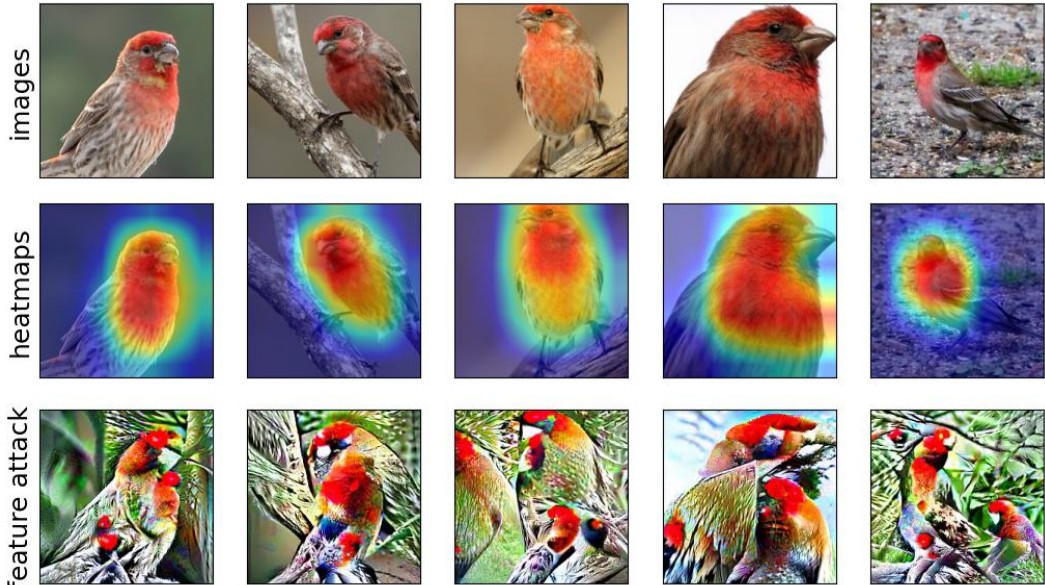

Figure 89: Visualization of feature **1667** for class **house finch** (class index: **12**).
Feature label as annotated by Mechanical Turk workers: **core**.
Using $\sigma = 0.25$, drop in accuracy: **-3.077%**, mean $l_2$ perturbation: **33.214**.
Using $\sigma = 0.4$, drop in accuracy: **-60.0%**, mean $l_2$ perturbation: **49.5**.

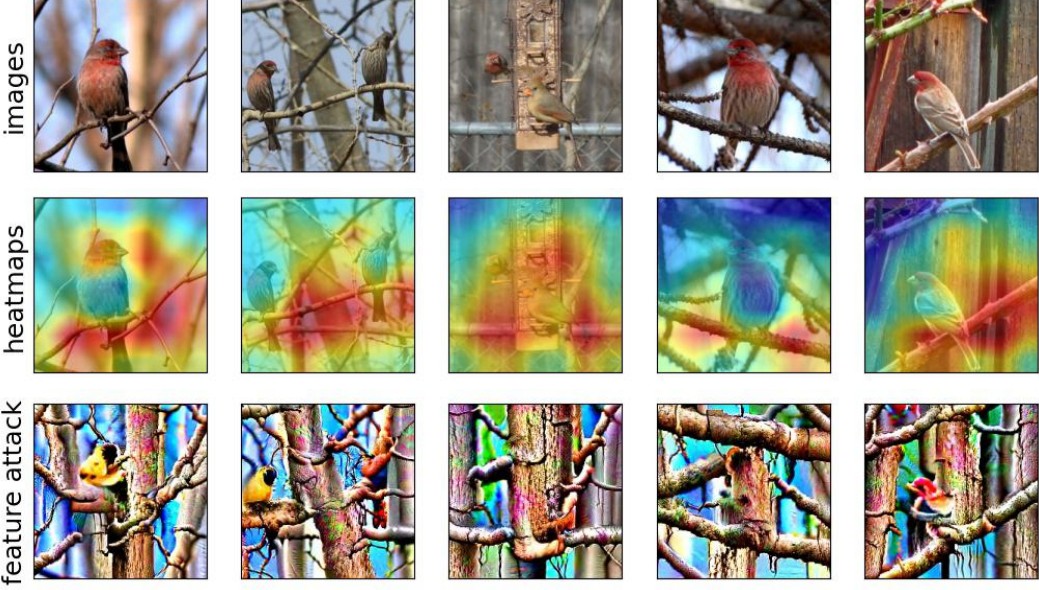

Figure 90: Visualization of feature **371** for class **house finch** (class index: **12**).
Feature label as annotated by Mechanical Turk workers: **spurious**.
Using $\sigma = 0.25$, drop in accuracy: **-24.615%**, mean $l_2$ perturbation: **49.474**.
Using $\sigma = 0.4$, drop in accuracy: **-60.0%**, mean $l_2$ perturbation: **73.833**.

### L.4 CLASS NAME: BULBUL, TRAIN ACCURACY (STANDARD RESNET-50): 98.308%

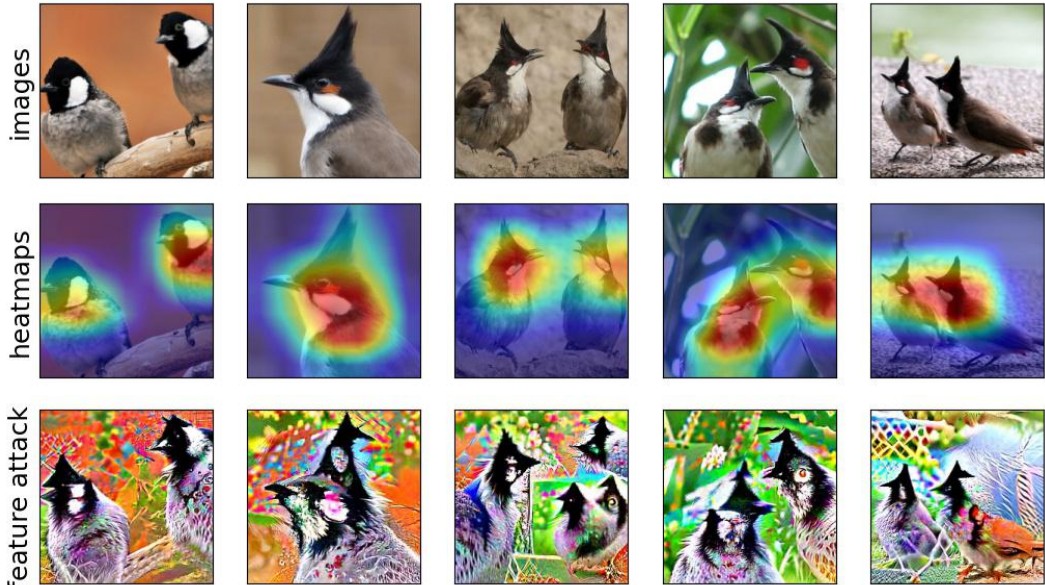

Figure 91: Visualization of feature **762** for class **bulbul** (class index: **16**).
Feature label as annotated by Mechanical Turk workers: **core**.
Using $\sigma = 0.25$, drop in accuracy: **-4.615**%, mean $l_2$ perturbation: **29.186**.
Using $\sigma = 0.45$, drop in accuracy: **-29.231**%, mean $l_2$ perturbation: **48.193**.

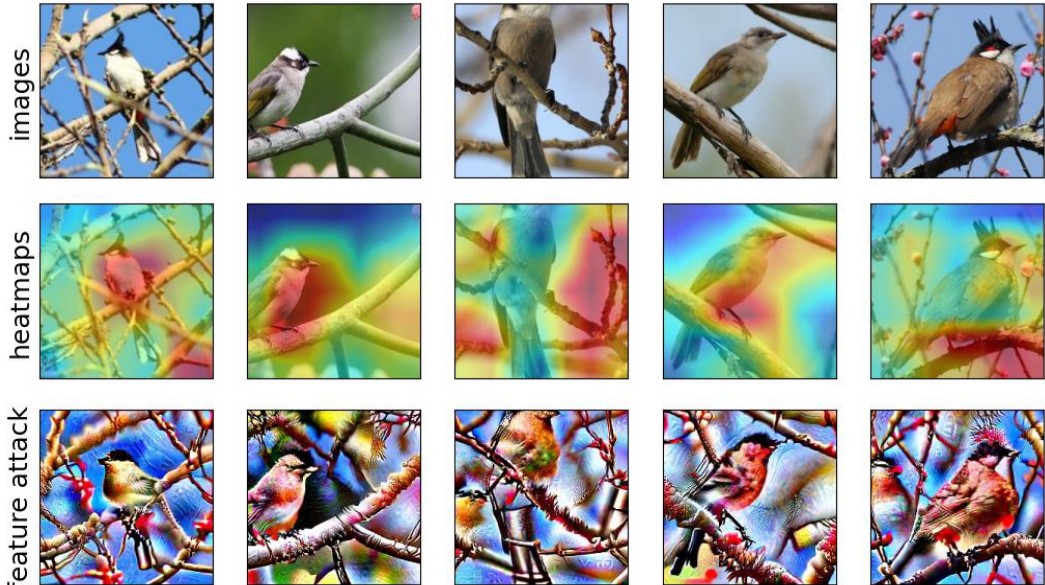

Figure 92: Visualization of feature **660** for class **bulbul** (class index: **16**).
Feature label as annotated by Mechanical Turk workers: **spurious**.
Using $\sigma = 0.25$, drop in accuracy: **-27.692**%, mean $l_2$ perturbation: **48.715**.
Using $\sigma = 0.4$, drop in accuracy: **-84.615**%, mean $l_2$ perturbation: **72.98**.

L.5  CLASS NAME: COUCAL, TRAIN ACCURACY (STANDARD RESNET-50): 97.308%

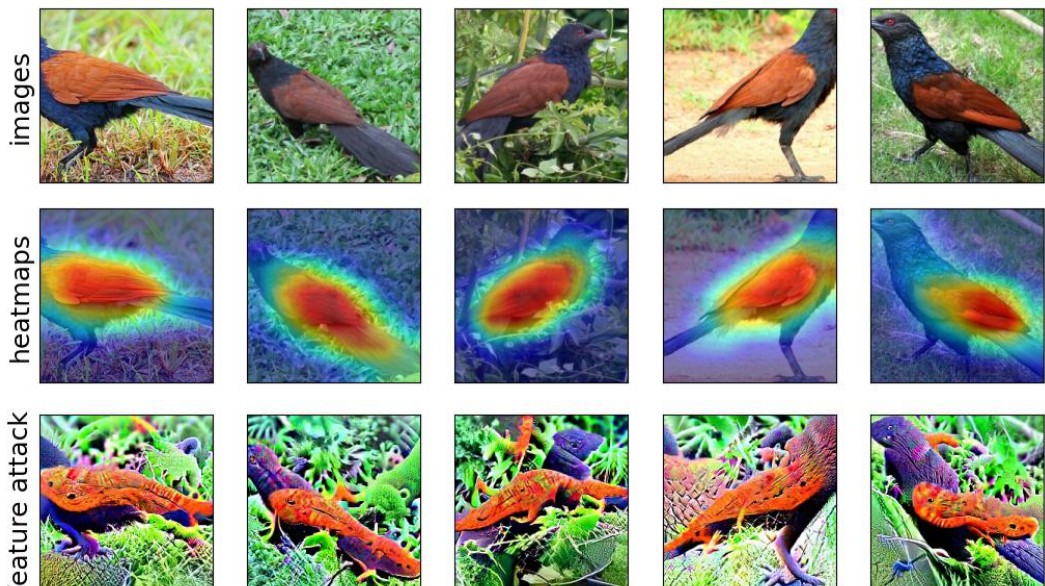

Figure 93: Visualization of feature **170** for class **coucal** (class index: **91**).
Feature label as annotated by Mechanical Turk workers: **core**.
Using $\sigma = 0.25$, drop in accuracy: **0.0**%, mean $l_2$ perturbation: **30.588**.
Using $\sigma = 0.5$, drop in accuracy: **-41.538**%, mean $l_2$ perturbation: **54.295**.

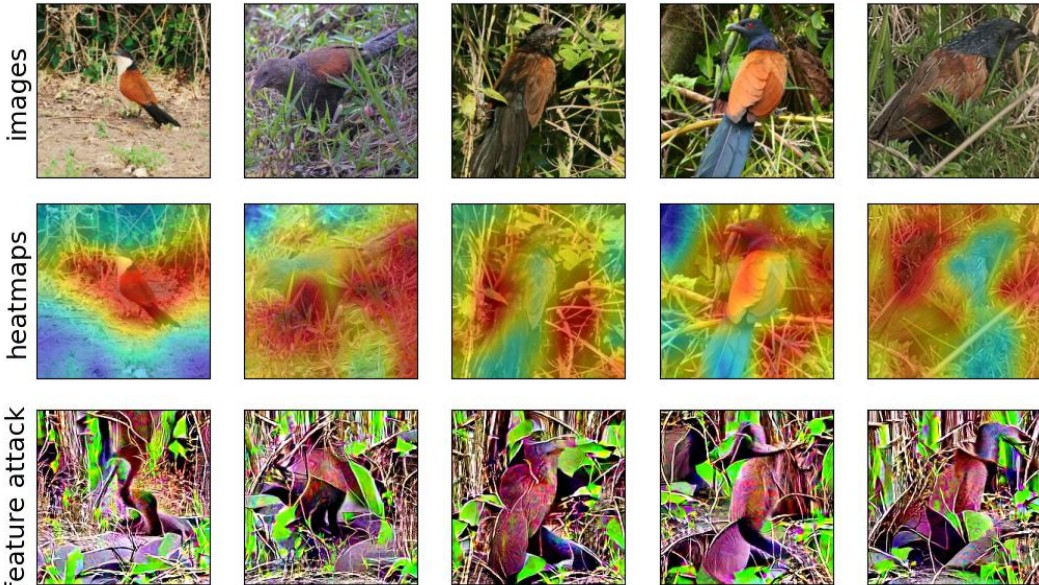

Figure 94: Visualization of feature **614** for class **coucal** (class index: **91**).
Feature label as annotated by Mechanical Turk workers: **spurious**.
Using $\sigma = 0.25$, drop in accuracy: **-27.693**%, mean $l_2$ perturbation: **54.854**.
Using $\sigma = 0.5$, drop in accuracy: **-92.308**%, mean $l_2$ perturbation: **96.12**.

L.6 CLASS NAME: JACAMAR, TRAIN ACCURACY (STANDARD RESNET-50): 97.0%

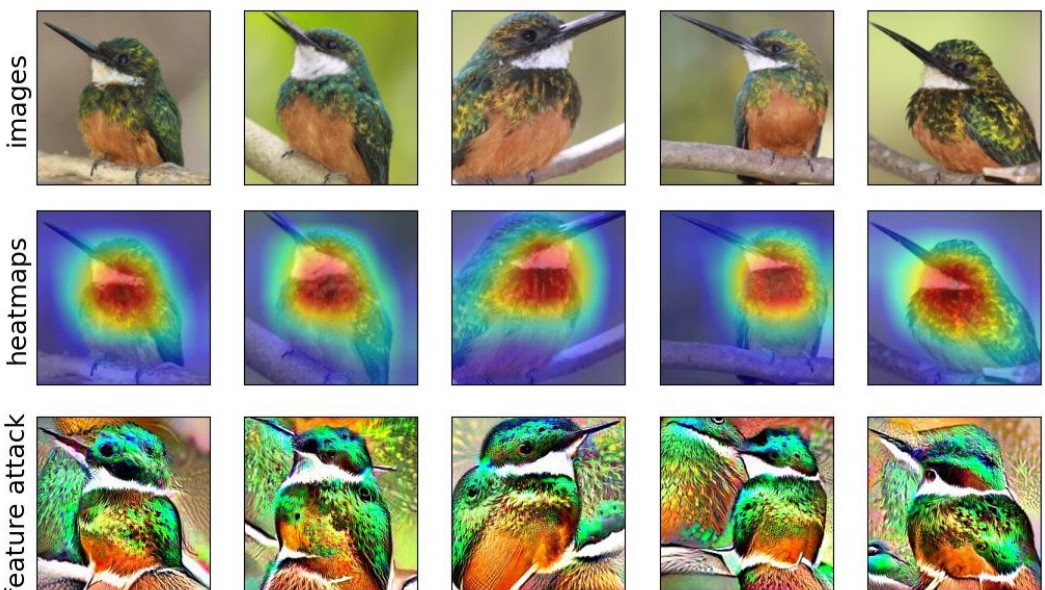

Figure 95: Visualization of feature **1339** for class **jacamar** (class index: **95**).
Feature label as annotated by Mechanical Turk workers: **core**.
Using $\sigma = 0.25$, drop in accuracy: **0.0**%, mean $l_2$ perturbation: **33.526**.
Using $\sigma = 0.45$, drop in accuracy: **-38.462**%, mean $l_2$ perturbation: **54.749**.

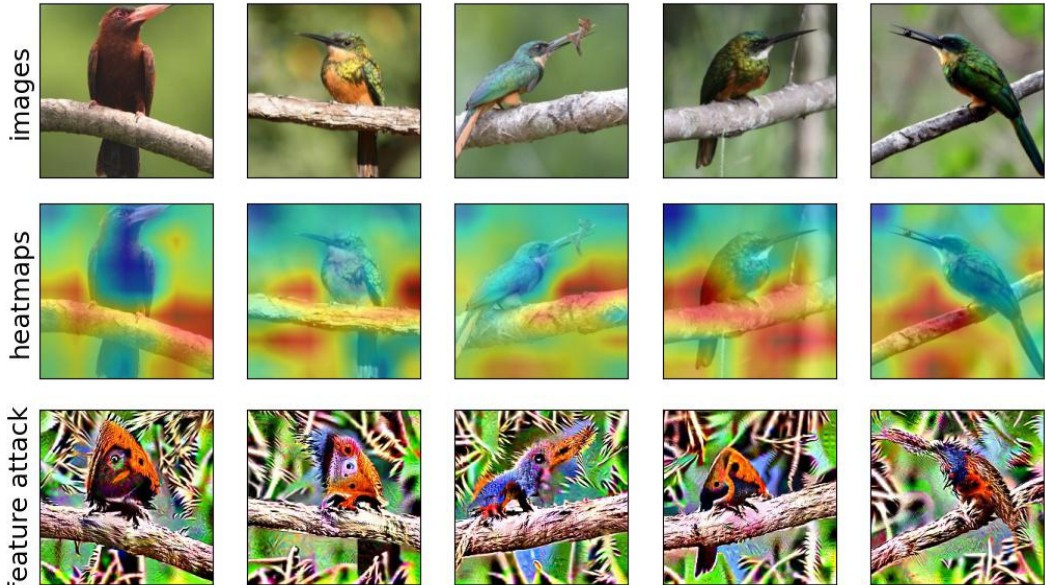

Figure 96: Visualization of feature **1556** for class **jacamar** (class index: **95**).
Feature label as annotated by Mechanical Turk workers: **spurious**.
Using $\sigma = 0.25$, drop in accuracy: **-21.538**%, mean $l_2$ perturbation: **53.542**.
Using $\sigma = 0.45$, drop in accuracy: **-76.923**%, mean $l_2$ perturbation: **87.339**.

### L.7 CLASS NAME: DRAKE, TRAIN ACCURACY (STANDARD RESNET-50): 95.462%

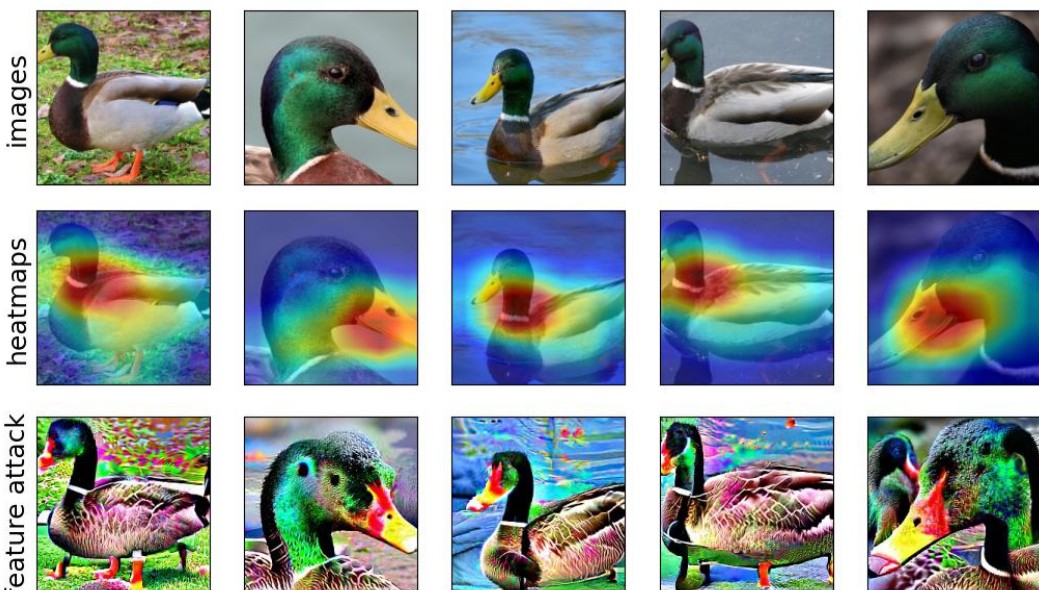

Figure 97: Visualization of feature **1113** for class **drake** (class index: **97**).
Feature label as annotated by Mechanical Turk workers: **core**.
Using $\sigma = 0.25$, drop in accuracy: **-1.538**%, mean $l_2$ perturbation: **36.558**.
Using $\sigma = 0.45$, drop in accuracy: **-16.923**%, mean $l_2$ perturbation: **60.362**.

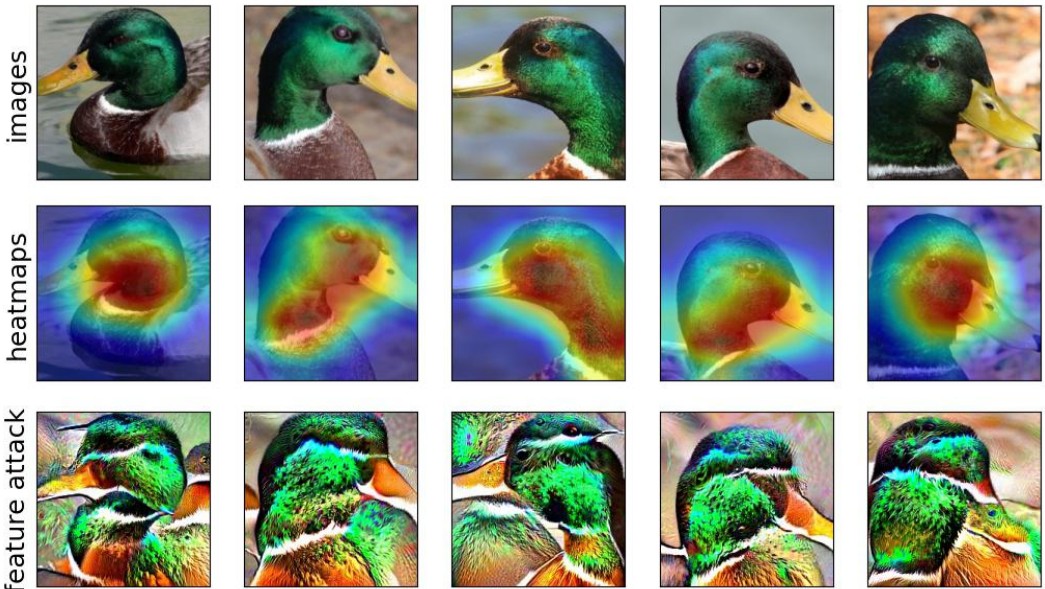

Figure 98: Visualization of feature **1339** for class **drake** (class index: **97**).
Feature label as annotated by Mechanical Turk workers: **core**.
Using $\sigma = 0.25$, drop in accuracy: **-4.615**%, mean $l_2$ perturbation: **32.946**.
Using $\sigma = 0.45$, drop in accuracy: **-16.923**%, mean $l_2$ perturbation: **54.731**.

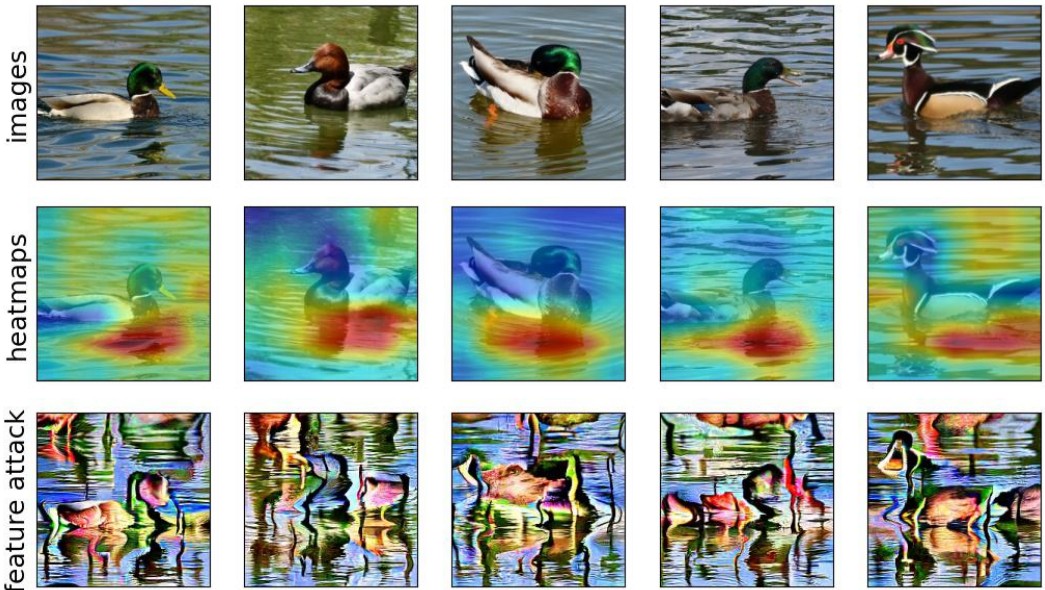

Figure 99: Visualization of feature **925** for class **drake** (class index: **97**).
Feature label as annotated by Mechanical Turk workers: **spurious**.
Using $\sigma = 0.25$, drop in accuracy: **-41.539**%, mean $l_2$ perturbation: **47.465**.
Using $\sigma = 0.45$, drop in accuracy: **-72.308**%, mean $l_2$ perturbation: **78.273**.

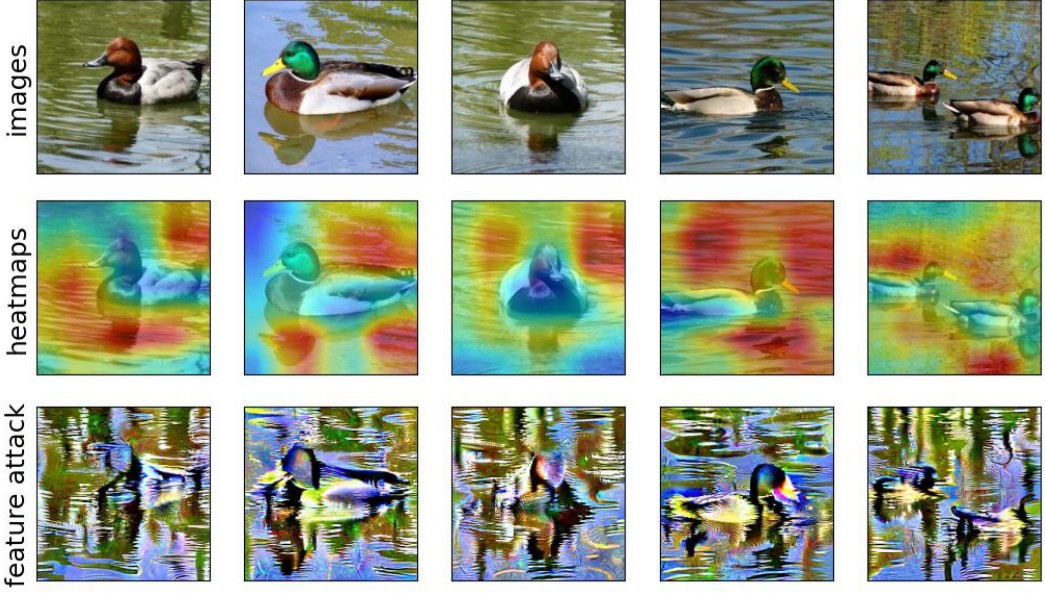

Figure 100: Visualization of feature **341** for class **drake** (class index: **97**).
Feature label as annotated by Mechanical Turk workers: **spurious**.
Using $\sigma = 0.25$, drop in accuracy: **-46.154**%, mean $l_2$ perturbation: **55.205**.
Using $\sigma = 0.45$, drop in accuracy: **-86.154**%, mean $l_2$ perturbation: **89.644**.

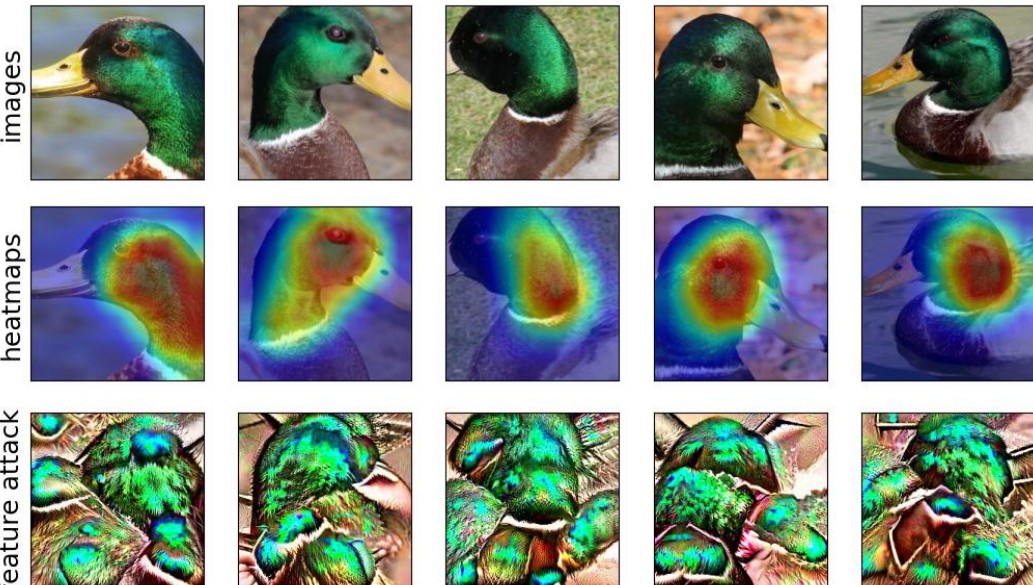

Figure 101: Visualization of feature **736** for class **drake** (class index: **97**).
Feature label as annotated by Mechanical Turk workers: **core**.
Using $\sigma = 0.25$, drop in accuracy: **0.0%**, mean $l_2$ perturbation: **28.878**.
Using $\sigma = 0.45$, drop in accuracy: **-6.154%**, mean $l_2$ perturbation: **47.661**.
Using $\sigma = 0.55$, drop in accuracy: **-7.692%**, mean $l_2$ perturbation: **55.554**.

## L.8 CLASS NAME: OYSTERCATCHER, TRAIN ACCURACY (STANDARD RESNET-50): 97.769%

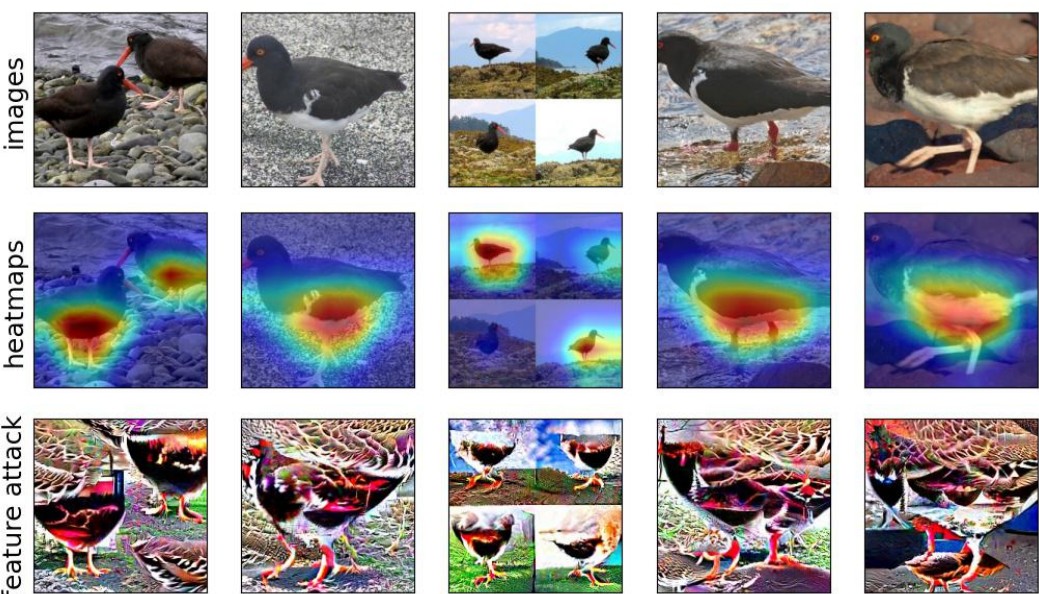

Figure 102: Visualization of feature **618** for class **oystercatcher** (class index: **143**).
Feature label as annotated by Mechanical Turk workers: **core**.
Using $\sigma = 0.25$, drop in accuracy: **-1.538**%, mean $l_2$ perturbation: **28.932**.
Using $\sigma = 0.5$, drop in accuracy: **-12.308**%, mean $l_2$ perturbation: **52.0**.

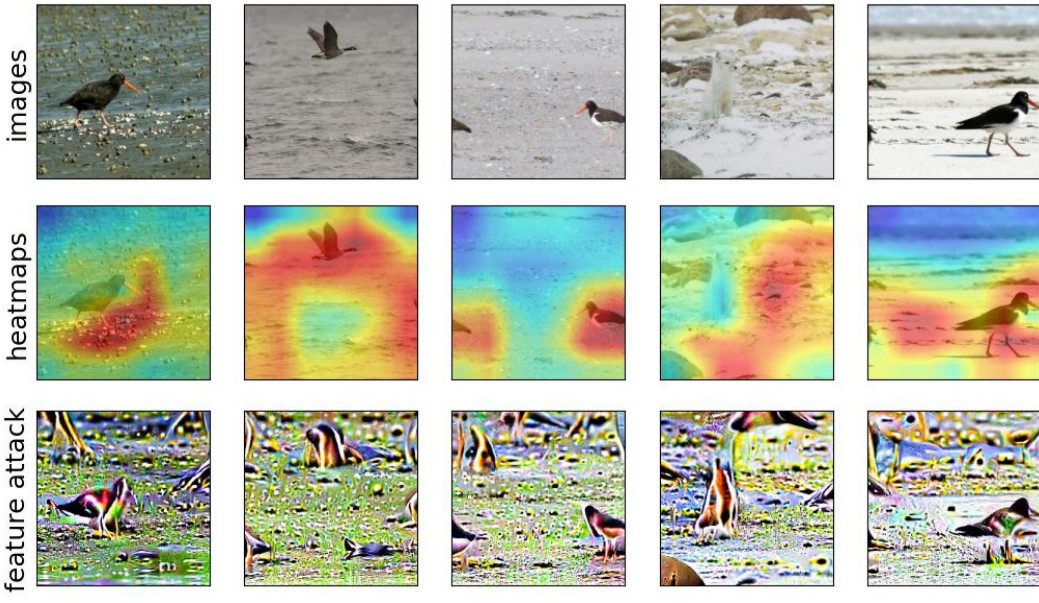

Figure 103: Visualization of feature **1468** for class **oystercatcher** (class index: **143**).
Feature label as annotated by Mechanical Turk workers: **spurious**.
Using $\sigma = 0.25$, drop in accuracy: **-21.539**%, mean $l_2$ perturbation: **51.153**.
Using $\sigma = 0.5$, drop in accuracy: **-70.77**%, mean $l_2$ perturbation: **90.219**.

### L.9 CLASS NAME: RINGLET, TRAIN ACCURACY (STANDARD RESNET-50): 95.692%

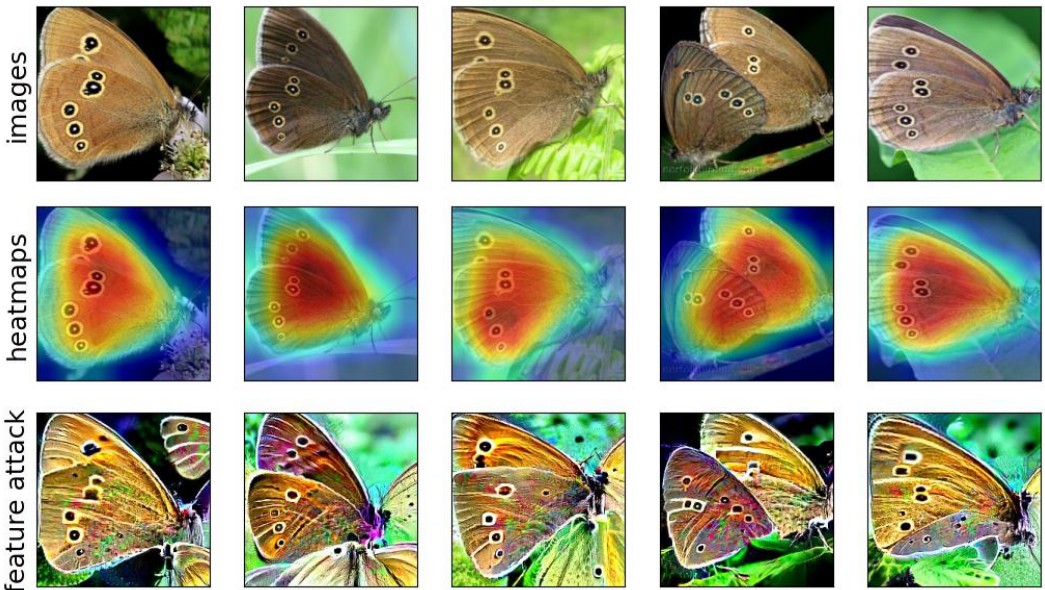

Figure 104: Visualization of feature **1305** for class **ringlet** (class index: **322**).
Feature label as annotated by Mechanical Turk workers: **core**.
Using $\sigma = 0.25$, drop in accuracy: **0.0%**, mean $l_2$ perturbation: **41.931**.
Using $\sigma = 0.35$, drop in accuracy: **-7.692%**, mean $l_2$ perturbation: **55.774**.

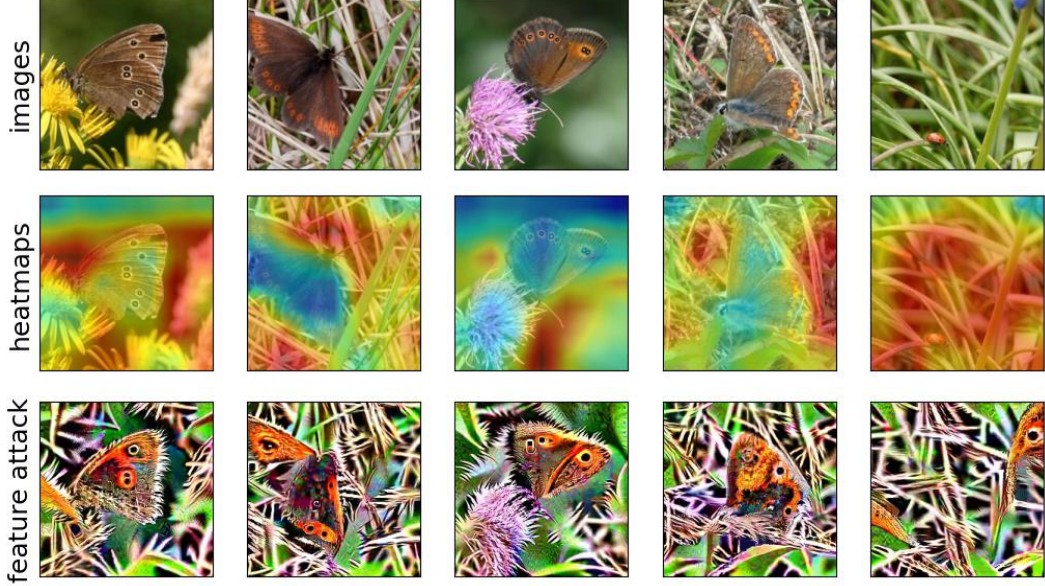

Figure 105: Visualization of feature **1556** for class **ringlet** (class index: **322**).
Feature label as annotated by Mechanical Turk workers: **spurious**.
Using $\sigma = 0.25$, drop in accuracy: **-36.924%**, mean $l_2$ perturbation: **53.174**.
Using $\sigma = 0.35$, drop in accuracy: **-70.77%**, mean $l_2$ perturbation: **71.192**.

L.10  CLASS NAME: LYCAENID, TRAIN ACCURACY (STANDARD RESNET-50): 97.846%

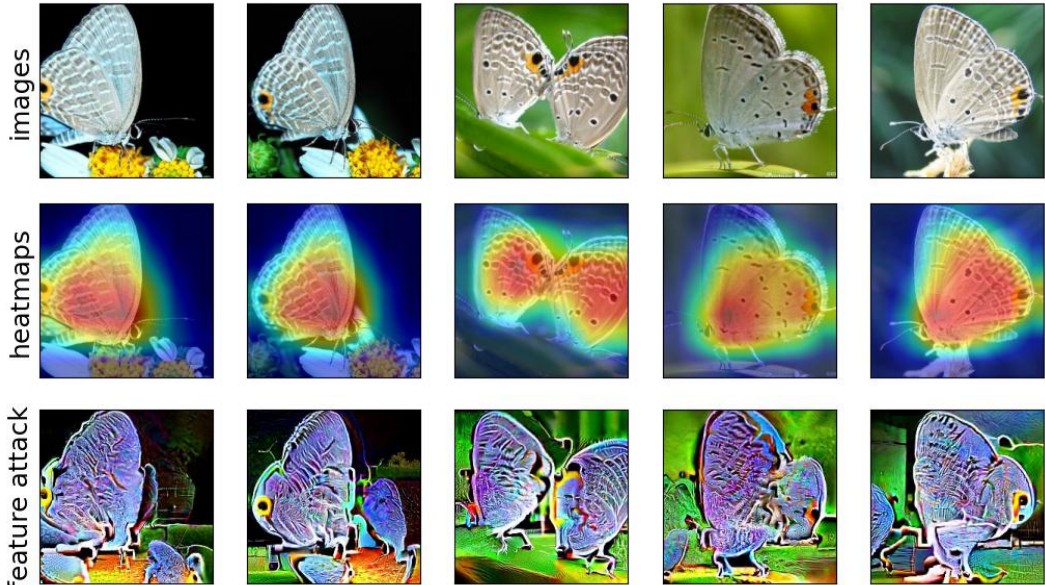

Figure 106: Visualization of feature **1041** for class **lycaenid** (class index: **326**).
Feature label as annotated by Mechanical Turk workers: **core**.
Using $\sigma = 0.25$, drop in accuracy: **-3.077%**, mean $l_2$ perturbation: **36.564**.
Using $\sigma = 0.4$, drop in accuracy: **-16.923%**, mean $l_2$ perturbation: **53.997**.

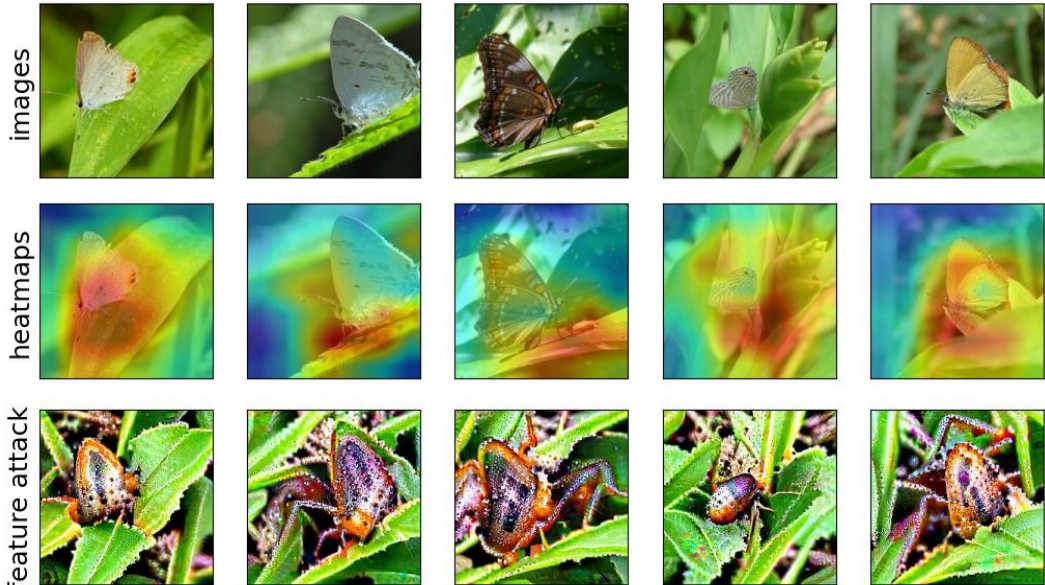

Figure 107: Visualization of feature **1390** for class **lycaenid** (class index: **326**).
Feature label as annotated by Mechanical Turk workers: **spurious**.
Using $\sigma = 0.25$, drop in accuracy: **-43.077%**, mean $l_2$ perturbation: **51.468**.
Using $\sigma = 0.4$, drop in accuracy: **-95.385%**, mean $l_2$ perturbation: **76.594**.

### L.11 CLASS NAME: PROBOSCIS MONKEY, TRAIN ACCURACY (STANDARD RESNET-50): 98.231%

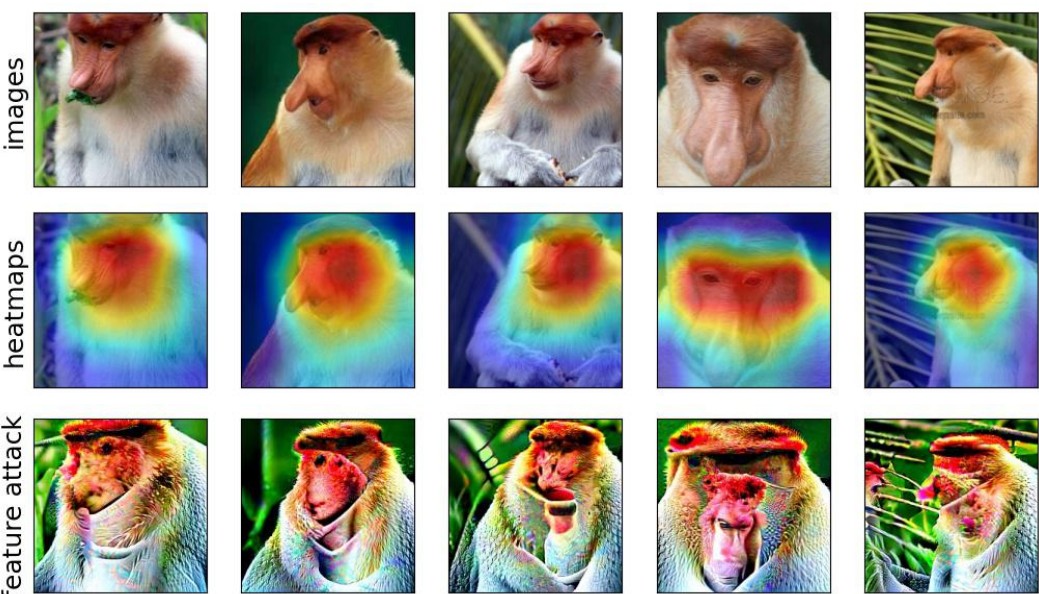

Figure 108: Visualization of feature **334** for class **proboscis monkey** (class index: **376**).
Feature label as annotated by Mechanical Turk workers: **core**.
Using $\sigma = 0.25$, drop in accuracy: **-6.154**%, mean $l_2$ perturbation: **37.884**.
Using $\sigma = 0.35$, drop in accuracy: **-15.385**%, mean $l_2$ perturbation: **50.698**.

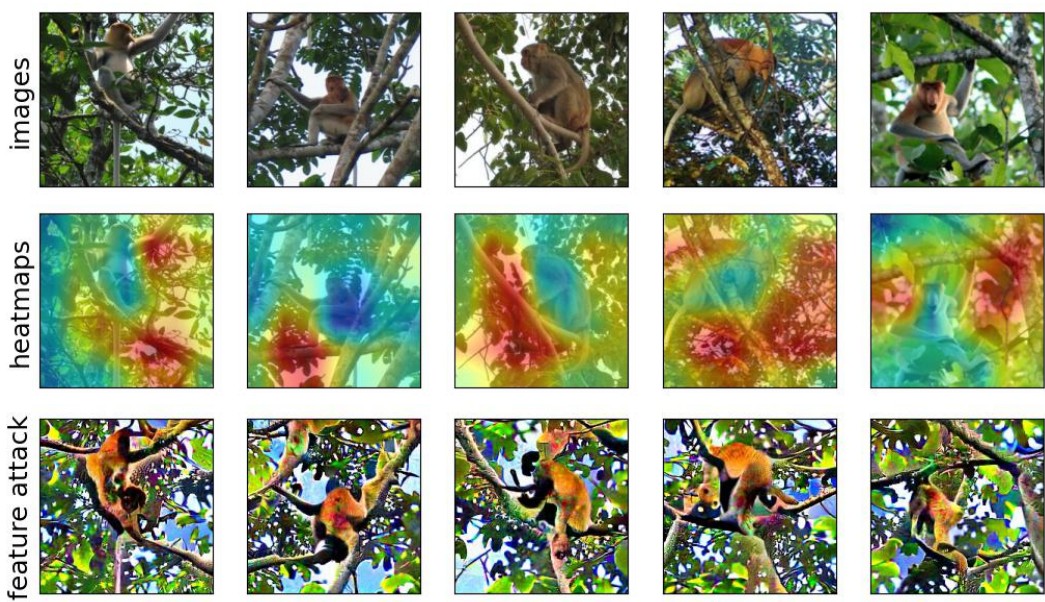

Figure 109: Visualization of feature **1120** for class **proboscis monkey** (class index: **376**).
Feature label as annotated by Mechanical Turk workers: **spurious**.
Using $\sigma = 0.25$, drop in accuracy: **-32.308**%, mean $l_2$ perturbation: **51.138**.
Using $\sigma = 0.35$, drop in accuracy: **-55.385**%, mean $l_2$ perturbation: **68.414**.

### L.12 CLASS NAME: TITI, TRAIN ACCURACY (STANDARD RESNET-50): 58.692%

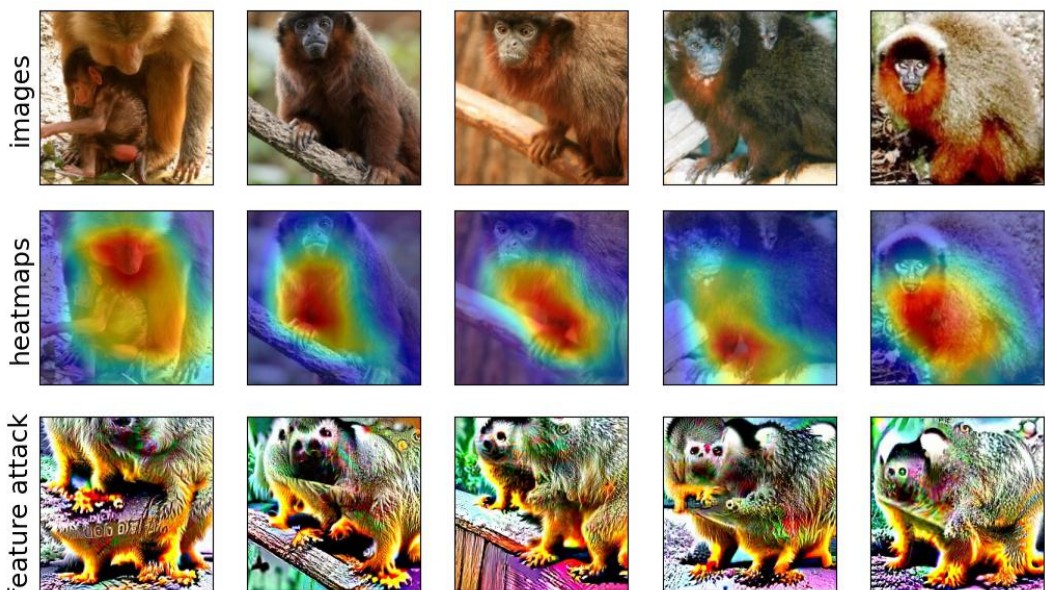

Figure 110: Visualization of feature **1003** for class **titi** (class index: **380**).
Feature label as annotated by Mechanical Turk workers: **core**.
Using $\sigma = 0.25$, drop in accuracy: **-1.539**%, mean $l_2$ perturbation: **34.703**.
Using $\sigma = 0.4$, drop in accuracy: **+12.307**%, mean $l_2$ perturbation: **51.98**.

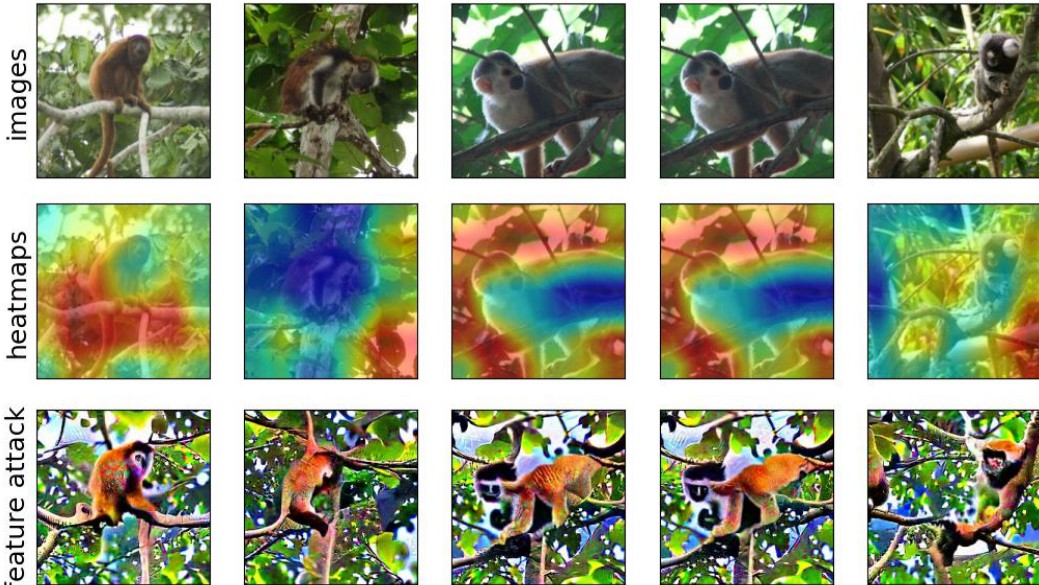

Figure 111: Visualization of feature **1120** for class **titi** (class index: **380**).
Feature label as annotated by Mechanical Turk workers: **spurious**.
Using $\sigma = 0.25$, drop in accuracy: **-38.462**%, mean $l_2$ perturbation: **50.591**.
Using $\sigma = 0.4$, drop in accuracy: **-53.846**%, mean $l_2$ perturbation: **75.361**.

## L.13 CLASS NAME: BAND AID, TRAIN ACCURACY (STANDARD RESNET-50): 81.0%

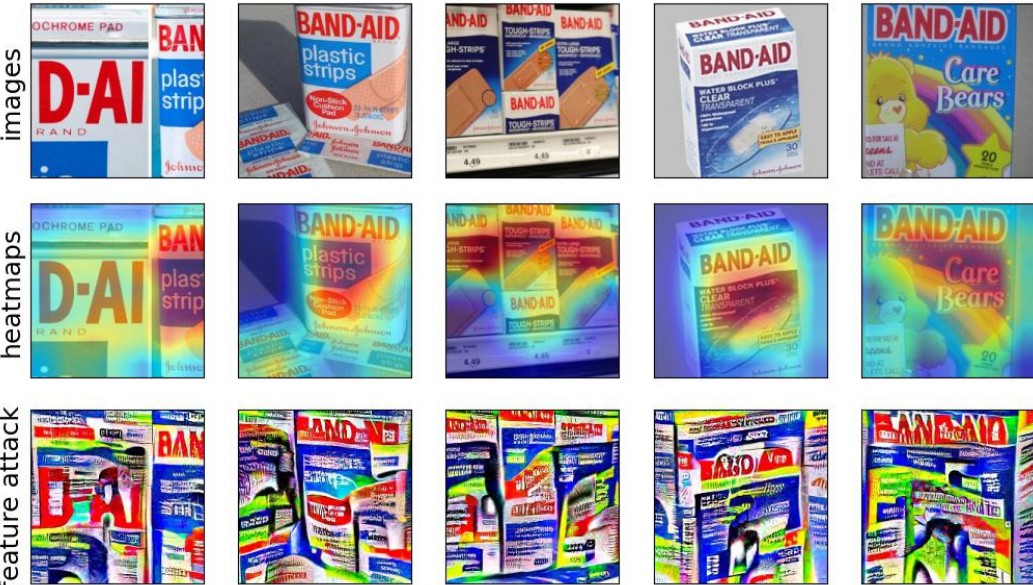

Figure 112: Visualization of feature **1164** for class **band aid** (class index: **419**).
Feature label as annotated by Mechanical Turk workers: **core**.
Using $\sigma = 0.25$, drop in accuracy: **-6.154**%, mean $l_2$ perturbation: **42.589**.

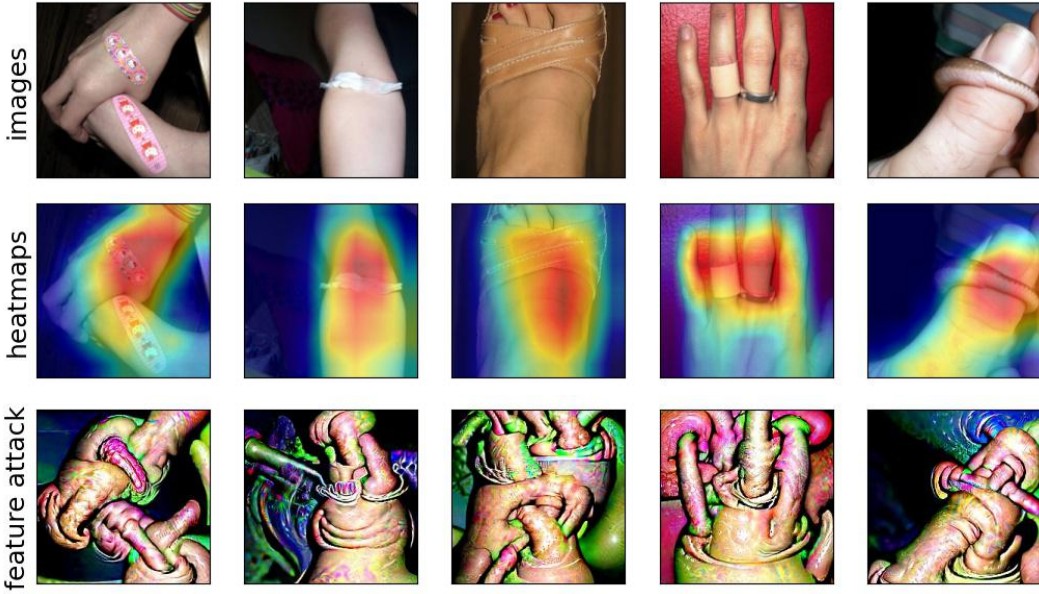

Figure 113: Visualization of feature **447** for class **band aid** (class index: **419**).
Feature label as annotated by Mechanical Turk workers: **spurious**.
Using $\sigma = 0.25$, drop in accuracy: **-43.077**%, mean $l_2$ perturbation: **41.225**.

### L.14 Class name: Projectile, Train accuracy (Standard ResNet-50): 53.538%

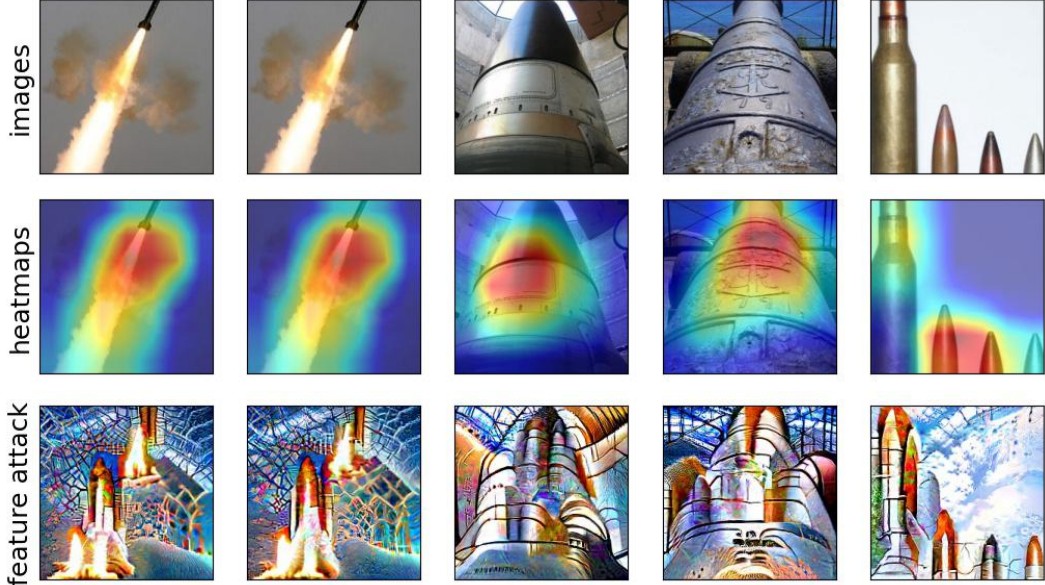

Figure 114: Visualization of feature **1606** for class **projectile** (class index: **744**).
Feature label as annotated by Mechanical Turk workers: **core**.
Using $\sigma = 0.25$, drop in accuracy: **+9.23**%, mean $l_2$ perturbation: **34.188**.
Using $\sigma = 0.45$, drop in accuracy: **+6.153**%, mean $l_2$ perturbation: **56.571**.

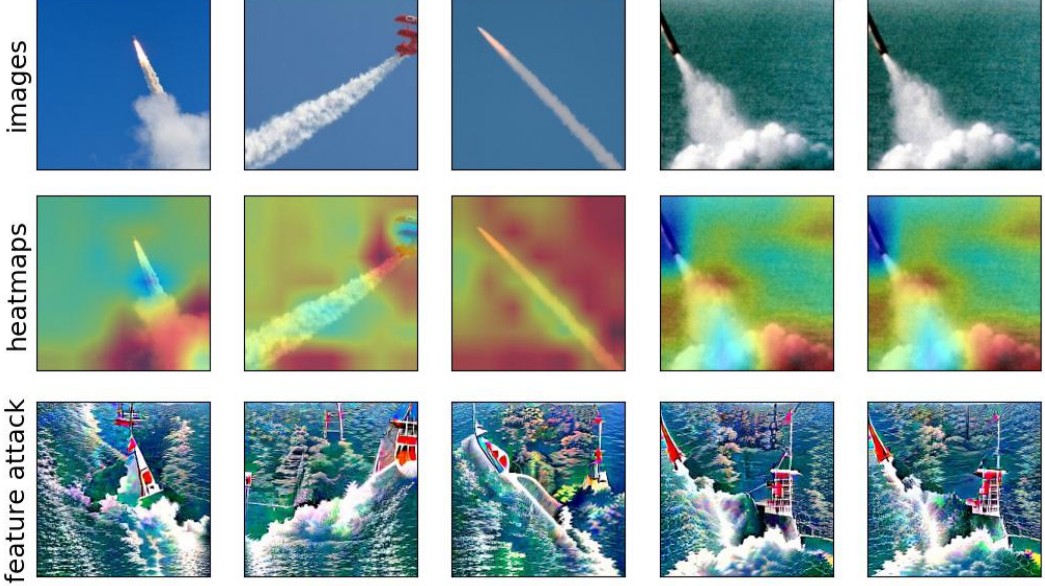

Figure 115: Visualization of feature **961** for class **projectile** (class index: **744**).
Feature label as annotated by Mechanical Turk workers: **spurious**.
Using $\sigma = 0.25$, drop in accuracy: **-23.077**%, mean $l_2$ perturbation: **56.179**.
Using $\sigma = 0.45$, drop in accuracy: **-30.769**%, mean $l_2$ perturbation: **91.036**.

L.15   CLASS NAME: WOK, TRAIN ACCURACY (STANDARD RESNET-50): 71.077%

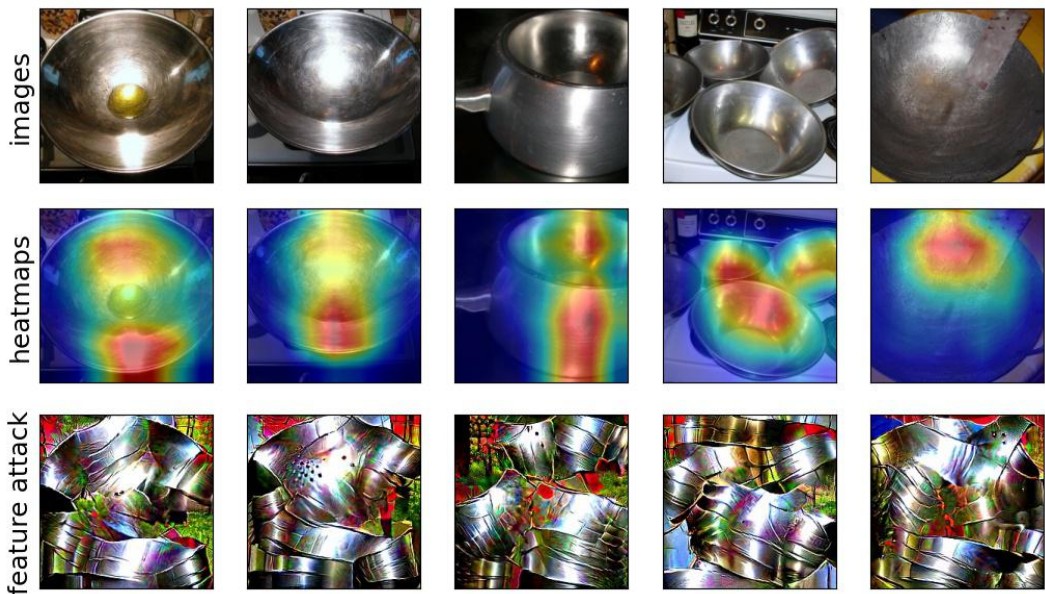

Figure 116: Visualization of feature **628** for class **wok** (class index: **909**).
Feature label as annotated by Mechanical Turk workers: **core**.
Using $\sigma = 0.25$, drop in accuracy: **-1.539**%, mean $l_2$ perturbation: **34.231**.
Using $\sigma = 0.4$, drop in accuracy: **-3.077**%, mean $l_2$ perturbation: **51.388**.

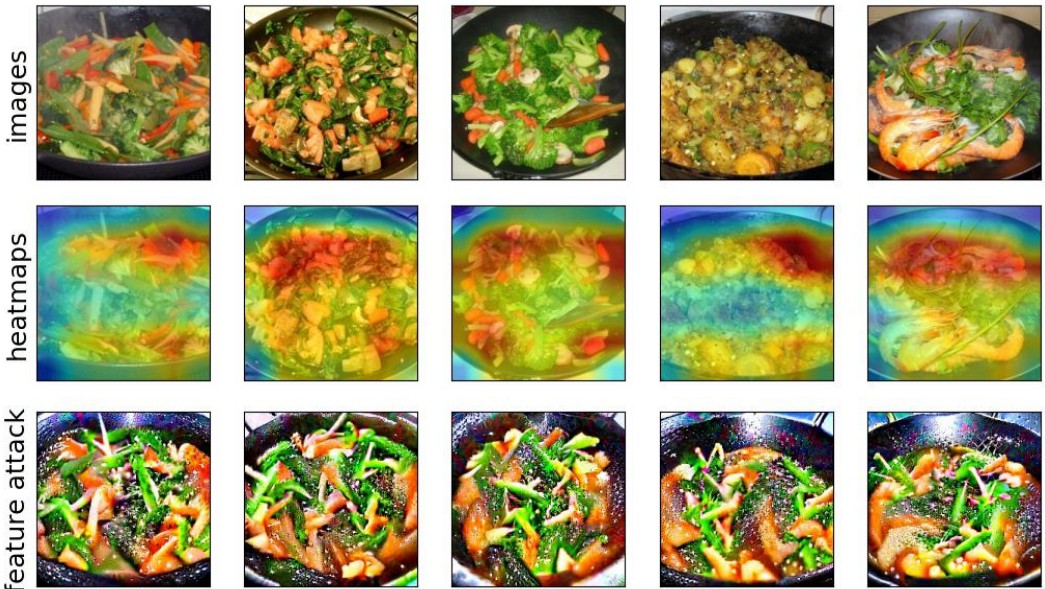

Figure 117: Visualization of feature **895** for class **wok** (class index: **909**).
Feature label as annotated by Mechanical Turk workers: **spurious**.
Using $\sigma = 0.25$, drop in accuracy: **-46.154**%, mean $l_2$ perturbation: **48.968**.
Using $\sigma = 0.4$, drop in accuracy: **-43.077**%, mean $l_2$ perturbation: **73.24**.

