# OpenReview forum: "Salient ImageNet: How to discover spurious features in Deep Learning?"
_ICLR.cc/2022/Conference — ICLR 2022 Poster_

### Official Review · Reviewer_1LYa · 2021-10-21

**Correctness:** 4
**Technical Novelty And Significance:** 3
**Empirical Novelty And Significance:** 3
**Recommendation:** 8
**Confidence:** 4

**Main Review:**


Strong (+) and weak (-) points

- (+) The need for annotations of causal/spurious features is important to support research on robust models/domains generalization/etc.
- (+) The method to propagate annotations from a small set of human labels is simple, effective, and based on observations from prior work that features of the last layer of a CNN are semantically aligned with human concepts.
- (-) There is however no theoretical basis/guarantee that this is the case.
- (+) But the authors performed a validation (Sec. 3.4) by crowd-sourcing that seems to support their assumptions.
- (-) The dataset is large but not huge (50k images, 232 classes).
- (+) The composition of the dataset (e.g. selection of the subset of 232 classes) seems to have been well thought.
- (+) The overall design of the data collection procedure generally inspires confidence that the crowd-sourced annotations are of good quality.
- (+) The authors use the new dataset to evaluate existing models and provide new observations that high accuracy on some classes is achieved by relying on spurious features.



Additional advice to improve the paper

- In the intro, the mention of cows/pastures is cited with Arjovsky et al. I'm not sure this was actually done as an experiment as part of this paper. My recollection is that it was more of a thought experiment (to be verified) and/or a citation of the Terra incognita paper of Beery et al.
- Typo: casual instead of causal
- The message in the last section that the standard accuracy metric does not tell the whole story is a very important one, IMO. It may benefit from being highlighted more prominently, possibly in the abstract, or maybe in bold in the last section as a form on conclusion/takeaway.

**Summary Of The Paper:**

- The paper describes a new benchmark. It uses images from ImageNet, which now have annotations of regions containing features spuriously correlated with the ground truth labels.
- The key contribution is a method to generate these annotations semi-automatically, by propagating a small set of human annotations to a large number of images.
- The dataset is used to evaluate existing models, showing they often rely on regions marked as spurious.

**Summary Of The Review:**

Preliminary recommendation: accept

Key reasons
- Importance of the topic/addresses a real need of the research community.
- Simple method, no technical flaw, underlying assumption are verified empirically.
- Experiments with the new dataset already provides new insights about existing models.

---

> ### Author Response · Authors · 2021-11-16
> **Author responses**
>
> We thank you for mentioning several strong points in our submission. First, we note that based on reviewers’ comment, we changed the name of the dataset: (i) We now call the provided dataset, “Salient Imagenet” (instead of Causal Imagenet) as we use saliency-based techniques to highlight visual attributes in the dataset. Please note that the title of the paper is modified accordingly in its pdf file (the openreview title cannot be changed at this stage but will be updated as soon as it is allowed), (ii) We use “core” features (instead of causal features) and we use core accuracy (instead of causal accuracy). All texts and figures (in the main text and appendix) have been updated accordingly.
>
> Below, we address the weak points you mentioned:
>
> **Lack of theoretical guarantees:** We completely agree that our work is “empirical” in nature as it aims to collect a richly annotated dataset. We believe, similar to the Imagenet dataset which paved the way for establishing and evaluating several theoretical ideas in machine learning, we hope that our “Salient Imagenet” dataset can make such an impact and also be used in future theoretical analyses.
>
> **Dataset size:** Our Salient Imagenet dataset (presented in this work) has >50K samples which, to our knowledge, is the largest available dataset with core/spurious attribute annotations. However, one of the key contributions of our paper is to propose a methodology that is more scalable than the previous methods for collecting core/spurious feature information (since our scheme requires very limited human annotations). The provided Salient Imagenet dataset is a proof of concept that our proposed methodology is successful in identifying core/spurious features used in deep learning. Upon the acceptance of this work, we plan to run and maintain an easy-to-use webpage for the Salient Imagenet dataset and also release versions of it in the future whose size is increased by multiple orders of magnitude, ultimately covering the entire Imagenet.
>
> **Correct citation for cows/pastures example:** Thank you. We clarified this and have updated the paper to include citations of both papers (Beery et al and Arjovsky et al.)
>
> **Standard accuracy metric not telling the whole story:** Thank you for this very important suggestion. We have added a sentence to the abstract (the last sentence) to emphasize this more prominently. We have also added a sentence in the last section in bold to further emphasize this message.
>
> Finally, we would like to thank you again for your detailed comments. We did our best to address them in our responses. Please let us know if you have any additional comments or questions. If you are satisfied with our responses, we would appreciate a score increase as we believe our “Salient Imagenet” dataset and our proposed methodology can be great resources for the community.

---

> > ### Comment · Reviewer_1LYa · 2021-11-16
> > **Satistfied with the authors' response**
> >
> > I appreciate the authors' response. I did not have a real issue with the use of causal terminology unlike one of the other reviewers. However I appreciate the change to "saliency" terms, given the fact that the human annotations can't be guaranteed to be actual causes of their mental processes !
> >
> > I maintain my recommendation for acceptance of this paper.

---

### Official Review · Reviewer_iQgx · 2021-10-31

**Correctness:** 3
**Technical Novelty And Significance:** 3
**Empirical Novelty And Significance:** 3
**Recommendation:** 6
**Confidence:** 3

**Main Review:**

Pros:

1. The paper is well-motivated and well-written, making its readers easier to follow the core idea delivered in the paper.
This will potentially make a good impact on the community.

2. The Visualization is very intuitive, this also helps me to quickly pinpoint the spurious vs causal features mentioned in the paper.

3. The scalability contribution is well-justified by the paper.

Cons:

1. Despite the acknowledgement of using less rigorous causal terminology, the issue remains. Take 'spurious' as an example, the author used descriptive language such as 'co-occur', it would be better later in the main content to add formal discussion related to Reichenbach’s common cause principle (Peters et al. (2017)).

2. What does causal formally mean in this paper?

The above two questions actually relate to my thought inspired Fig 5 d and e. In Fig 5 d and e, the mask highlights the suspicious attributes which could be the reflected shadow of the bird (maybe I am wrong.) But must this feature be spurious? What if the river is very quiet without any ripple and it is literally a mirror? I think a more precise discussion about causal vs spurious in vision scenario could thus be very useful.

3.  I am a little bit unclear about the concrete usage of this method. Could the authors enlighten me a little more on how we could potentially utilize this model to further improve either the interpretability or robustness of NNs.


**Summary Of The Paper:**

In the paper, the authors presented a novel scalable method to differentiate the so-called causal or non-causal attributes.
By utilizing the neural features activations of in the penultimate layer, i.e. the one adjacent to logits, the proposed method can allow efficiently annotate plausible masks indicating whether the decision-making process of an NN relies on the correct features.


**Summary Of The Review:**

Based on the pros and cons discussed above, I believe this paper bears the good potential to bring useful insights to the community, though a more thorough discussion from the theoretical side should be improved.

After reading the response from the authors, I am pleased with the paper revision and decide to raise my score.

---

> ### Author Response · Authors · 2021-11-16
> **Author responses**
>
> **Use of causal terminology:**  Thank you for the comment. We agree with you that the original use of causality in our submission was not completely accurate. Based on your and other reviewers’ comment, we have made the following updates to the paper: (i) We now call the provided dataset, “Salient Imagenet” (instead of Causal Imagenet) as we use saliency-based techniques to highlight visual attributes in the dataset. Please note that the title of the paper is modified accordingly in its pdf file (the openreview title cannot be changed at this stage but will be updated as soon as it is allowed), (ii) We use “core” features (instead of causal features) and we use core accuracy (instead of causal accuracy). All texts and figures (in the main text and appendix) have been updated accordingly.
>
> **Definitions of core and spurious features:** We have made the practical definitions more explicit in Section 3 of the paper. Additionally, we have added a section in Appendix G to discuss formal definitions of these abstract concepts based on graphical models (with connections to Reichenbach’s common cause principle) as well as to highlight challenges in these definitions. We note that we tried different definitions of causal and spurious features in our MTurk studies and settled on the definition used in the paper because we observed that other definitions often confused the Mechanical Turk workers. For example, we observed that workers would often classify graffiti as a part of a wagon (i.e., a core attribute). But we want our model to classify “wagon” correctly even when the graffiti is absent. When we instructed the workers to classify something as spurious when the said attribute could be removed without changing its identity, workers would also classify “dog limbs” as spurious (because a dog can still be classified correctly when its limbs are removed). We have added these explanations to Appendix G.
>
> **The bird example in Figures 5 d,e:** In these figures, the spurious masks highlight broadly the water (that partially overlaps with the shadow of the bird on the water). Based on our core/spurious definitions, the shadow should be considered as a spurious visual attribute because if the true object is something different from the bird but creates a similar shadow in the water, the model might still misclassify it as a bird (which is not what we would want). In the specific case when the exact mirror of the object is also present (e.g. when a river is completely quiet and without any ripple), we will have replicas of core features since both the object and its mirror will have exactly the same visual attributes.
>
> **Concrete usage of the method:** In this work, we demonstrate that even if a model achieves close to 100% accuracy on some benchmark, it is not guaranteed to work in the real world where the spurious visual attributes may be absent. Consider a setting in which a machine learning developer wants to understand the failure modes of some model before deployment. The usual method of discovering spurious features would require the developer to first hypothesize about the possible spurious attributes per class. Let us assume that the developer believes that a "green background" is the spurious feature for the class “tiger cat”. This would be followed by manual annotation of green/non-green background per image for all images in the dataset with label tiger cat. This may also require collection of additional images with cats in non-green backgrounds. Then comparing model performance on two sufficiently large subsets of data with green and non-green background would allow the developer to conclude whether green background is the spurious attribute being used by the model or not. This process will need to be repeated for every single class and their potential spurious attributes. This can be very expensive. In contrast, we show that our introduced framework can be used to discover such failure modes (i.e., reliance on spurious features) for a large number of classes for standard models that achieve close to 100% accuracy on many of these classes. Moreover, our introduced “Salient Imagenet” dataset can be used as a further stress test for models to test their reliance on visual attributes that are relevant for the object of interest. We formalize these measures as core accuracy and spurious accuracy in the paper. Finally, as we explain in Appendix E, our proposed framework can be used to discover spurious visual attributes for any other dataset (with a sufficiently large number of samples).
>
> We hope that we have fully addressed your comments. Please let us know if you have any additional questions/comments. If you are satisfied with our responses, we would appreciate a score increase as we believe our “Salient Imagenet” dataset and our proposed methodology can be great resources for the community.

---

> ### Author Response · Authors · 2021-11-19
> **Following up**
>
> Dear Reviewer iQgx,
>
> We appreciate your time reviewing our work and for your insightful comments. In our responses, we did our best to address all of your comments. Before the discussion period ends, please let us know if you have any additional comments. If you are satisfied with our responses, we would appreciate a score increase.

---

### Official Review · Reviewer_DwGM · 2021-11-02

**Correctness:** 3
**Technical Novelty And Significance:** 3
**Empirical Novelty And Significance:** 3
**Recommendation:** 8
**Confidence:** 4

**Details Of Ethics Concerns:**

None.

**Main Review:**

This paper tackles an important problem and provides a convincing demonstration that current SOTA models are susceptible to reliance on spurious correlation in the training set. I detail key strengths, weaknesses, and additional questions below.

###  Strengthens
- The focus on the ImageNet dataset is impressive and crucial to demonstrating that the reliance on spurious correlation issue is not isolated to toy datasets, but prevalent for one of the currently most important datasets in machine learning.
- The use of crowd source annotations is also helpful and interesting.
- While the pieces of the entire scheme presented in this work are not new, the combination is interesting and extends recent work that is focused on detecting whether a model relies on spurious correlation.


### Weaknesses, Concerns,  & Additional Questions
- 'Causality': This paper uses the term 'causal' inappropriately in my opinion. Nothing about the new dataset is 'causal' in any sense of the word. At this point, causality refers to a specific set of techniques that capture the data generating process. The Causal ImageNet here is a dataset that indicates which are 'core' features and which are spurious features. I would encourage the authors to completely discard the notions of causality that they refer to repeatedly across the paper. Further, the definition of 'causal' features in this work is quite vague, but acceptable. I would strongly suggest that the authors call those features 'core' features or something of that sort.
- Non spatial spurious signals: The scheme presented here would only be effective for spatial/dimension dependent features. If the DNN learns say Gender/Race or some other concept that is not visualizable by a heatmap then the approach described here will not work. I would suggest that the authors make this clear in the paper. This is not a limitation in my view, but it allows the reader to understand the scope of the work presented.
- Clarify the entire scheme and specific assumptions: As it stands, it is difficult to understand the entire scheme here. Part of the difficulty is that Section 3 should be written better to delineate each component of the entire scheme, i.e., if someone else wanted to apply this technique to a different dataset and model, what is needed? State the inputs, outputs, assumptions, etc necessary to apply the scheme discussed here to another setting. Since the authors say they are presenting a scalable framework for discovering spurious features, then it is important to completely state all the necessary requirements to apply the method.
- What prevents this method from being applied to a non-robust model? Will the approach be less effective? I don't understand whether the models used in Figure 6 are robust or standard models. I think the models for Fig. 6 are standard, i.e. non-adversarially trained, and if yes, why would annotations be collected for a robust model and used to test a non-robust one?
- Why CAM? Could any feature attribution method have worked here?
- The user study is interesting, but it is unclear what one can learn from how the crowd workers determined which feature is spurious or not. I say this because the study had the top activating images, the heat maps, the feature attack, wikipedia link, object names etc. To me, it is unclear which of these sources of information helped the crowd workers to determine that a feature is spurious or not. This is quite important information if this approach is to be replicated else where.
- Are spurious features only learned at the top layer? One assumption in this work is that the important features learned by a DNN are at the penultimate layer? What is the justification for this assumption? I am fine with this assumption, but curious if there is any past work that has made this clear. Could it be possible that a model could concentrate a spurious signal to a filter at the intermediate layer?
- Given that one can detect reliance on spurious signals, could as straightforward fix be to fine tune the model and include a penalty on the heat maps to force the network to be invariant to the spurious features? This kind of approach has been proposed in prior work.

### Minor Feedback
- The change in accuracy numbers in Figure 1 were confusing on first read. Perhaps the authors can provide additional statements in the caption to explain what the numbers mean. After reading the paper it makes sense, but it was confusing at first.
- In the introduction and the related work, the authors claim that their approach is more 'scalable' than current alternatives. I don't agree. To use the approach presented here, one needs to run a user study to collect more features, and another study to confirm that the spurious features detected will generalize. I think the spurious feature detection issue is so important that the paper doesn't need the scalability argument. Any method that requires one to run a study per dataset is not necessarily scalable, but that is also fine!
- Here is recent work that this paper should also acknowledge: https://arxiv.org/abs/2106.02112

### Post Rebuttal Updates
The author response directly addresses most of the concerns that I had. I maintain my accept rating for this paper.

**Summary Of The Paper:**

This work proposes a scheme to identify 'causal' and 'spurious' features with human supervision. More specifically, the method presented decomposes the activations that map to the logits of a supervised DNN into neural features. The neural features are then ranked and visualized with the CAM feature attribution method (called neural activation map in this work). The paper then shows the top ranked features and the corresponding heatmaps to crowd workers, who then ascertain whether a feature is spurious or not.

Given a ranking of causal and spurious features for different output classes, the work then maps these features across an entire 'audit'/validation set. Given this audit set, one can then mask out the regions in the heatmap that correspond to the casual features and observe the corresponding change in model accuracy. Similar sensitivity measurements can also be made for the spurious features as well. Overall, this paper produces an annotated version of the ImageNet dataset with causal and spurious features. This is then used to assess the performance of current state of the art models showing that these models show significant susceptibility to spurious training signals.

Overall, this paper shows how to use additional supervision for detecting a model's reliance on spurious features. The 'Causal Imagenet' dataset should be useful for future work as a stress test for models.

**Summary Of The Review:**

I recommend an accept at this time because most of the weaknesses I identified with this paper have to do with the strength of the claims made, which can be addressed in a revision. I particularly like that the paper demonstrates significant weaknesses, reliance on spurious features, of current models on the ImageNet dataset. On the strength of that contribution alone, I think this work is an important one.

---

> ### Author Response · Authors · 2021-11-16
> **Author Responses (Part 1)**
>
> **Use of causality:** Thank you for the comment. We agree with you that the original use of causality in our submission was not completely accurate. Based on your comment, we have made the following updates to the paper: (i) We now call the provided dataset, "Salient Imagenet" (instead of Causal Imagenet) as we use saliency-based techniques to highlight visual attributes in the dataset. Please note that the title of the paper is modified accordingly in its pdf file (the openreview title cannot be changed at this stage but will be updated as soon as it is allowed), (ii) upon your suggestion, we use "core" features (instead of causal features) and we use core accuracy (instead of causal accuracy). All texts and figures (in the main text and appendix) have been updated accordingly. Regarding the definitions of core and spurious features, we have made the practical definitions more explicit in Section 3 of the paper. Additionally, we have added the Section in appendix G to discuss formal definitions of these abstract concepts based on graphical models as well as to highlight the challenges in these definitions.
>
> **Non spatial spurious signals:** You are completely right that our methodology does not apply for discovering *non-spatial* spurious signals (such as gender/race) that are not visualizable using heatmaps/feature attacks. We have added this limitation of our approach on Page 6, Section 3.3 (first paragraph).
>
> **Components of the entire scheme:** We agree that the steps of our methodology were not explicitly mentioned in the original submission. We have now added specific steps of our framework at the end of the Introduction Section with proper pointers to different parts of Sections 3, 4 and 5. Moreover, regarding applications of our scheme to new datasets, we have added a new section in the appendix (Appendix E) to explain exact inputs and assumptions for different steps of our methodology.
>
> **Use of robust vs. non-robust model:** Singla et al. [1] (Appendix I, Pages 52-56) show examples in their paper where standard models do not lead to interpretable activation maps and feature attack results. That is, using standard models, activation maps often highlight completely different visual attributes and feature attack leads to noisy versions of the original images. However, for robust models we always see certain visual attributes are amplified in the feature attack visualizations and heatmaps are coherent with the visual attribute amplified in the feature attack visualizations. This is the reason why annotations are collected using a robust model. Since non-robust (standard) models are more widely deployed due to their higher clean accuracy, we use these core/spurious feature annotations to test the performance of non-robust models. The models in Figure 6 are all standard models. We have added a sentence on Page 9 (second paragraph) to make it clearer.
>
> **CAM vs other feature attribution methods:** We tried different feature attribution methods (saliency maps, smoothgrad) and observed that CAM resulted in the best visualizations. It is possible that other visualization methods may give even better results than CAM and we leave that avenue open for future research.
>
> **Reasons for MTurk worker answers:** Although understanding the reasoning behind the answers of MTurk workers is a general and challenging problem in all human studies, we have taken the following steps to address this problem: (i) workers were required to give reasons for their answers to complete each task and (ii) we also instructed the MTurk workers to use heatmaps (called highlighted visual attributes in the study) for their answers and use feature attack visualization (called amplified visual attributes in the study) for their answers when the heatmaps are unclear. The reasons provided by the workers are given in the column “reasons” in the files mturk_study_discover_sec3_3.csv and mturk_study_generalize_sec3_4.csv (provided in the supplemental material in the folder MTurk_studies). In the file mturk_study_discover_sec3_3.csv and the column “reasons”, we see 1231 mentions of the word “highlighted” and only 29 mentions of the word “amplified” suggesting that the workers use highlighted visual attributes (heatmaps) much more than the amplified visual attributes (feature attack). We have added this discussion to Appendix Section F. We have also included two figures on page 20 (Figures 11 and 12) with complete reasons provided by workers. Sample reasons for Figure 11 are: “Legs of spider are mainly focused”, “The red is on the legs of the barn spider”.  Sample reasons for Figure 12 are: “focus is highlighting the bird’s wing”, “focus is on the feathers of the bird”.
>
> [1] Understanding Failures of Deep Networks via Robust Feature Extraction. Sahil Singla, Besmira Nushi, Shital Shah, Ece Kamar and Eric Horvitz.

---

> > ### Comment · Reviewer_DwGM · 2021-11-26
> > **Read the rebuttal**
> >
> > I have read the responses and satisfied with the updates.

---

> ### Author Response · Authors · 2021-11-16
> **Author Responses (Part 2)**
>
> **Penultimate layer vs. previous layers:** Multiple previous works (e.g., Singla et al. [1], Wong et al. [2]) use penultimate layer features for analyzing failures. A main reason that we use the penultimate layer in our work is that the mapping from the penultimate layer to the logits is linear and hence it is easier to analyze which features contribute most to the logit for a certain class (i.e., as explained in “Neural Feature Importance” score in Definition 1 on Page 5). However, it might be possible to use the previous layers for analyzing spurious features and that remains a promising direction for future research.
>
> **Fixing reliance on spurious features:** Indeed it is possible that penalizing activation maps when they focus on spurious features might lead to improved models that are invariant to spurious features in their inferences. To be able to develop and study such reliable learning paradigms, one needs a dataset whose samples, in addition to the labels, are annotated by core and spurious attribute masks. This is precisely the problem we have addressed in our paper. Although it is outside of the scope of this work, we believe our “Salient Imagenet” dataset will pave the way for future research in developing such reliable learning paradigms.
>
> **Change in accuracy numbers:** Thank you for bringing this to our attention. We have edited the caption in Figure 1 to explain to the reader that the accuracy drop is only on a subset of 65 images with the shown class labels.
>
> **Scalability:** The usual method of determining whether a model is sensitive to some spurious attribute (say green background) for some class (say tiger cat) requires one to first obtain a subset of the training data where tiger cats occur in green background and another subset with tiger cats in some other background followed by computing the accuracy of models on both sets. However, obtaining these subsets would require manual annotation of all 1300 images with the label tiger cat in the Imagenet training set. In contrast, our approach only requires manual inspections of 5 images with highest and lowest activations per class. This is the reason we mentioned that our approach is more scalable than the previous ones: our framework requires significantly lower manual annotation than the existing methods. This lower human annotation effort is precisely what allows us to annotate spurious/core attributes for 232 classes in Imagenet. However, we also agree that our scheme is not “fully” scalable as it requires “some” human annotations. Based on your comment, we removed scalability phrases in the updated draft.
>
> **Acknowledgement of paper:** Thank you for bringing this recent paper to our attention. We have cited the paper in our related work section now. This work indeed makes key contributions by introducing a method for identifying spurious features and mitigating the errors. However, the major difference with our work is that it still relies on human annotations for discovering spurious visual attributes. In contrast, we use a robust model as a visual attribute detector thereby significantly reducing the need for human annotations.
>
> Finally, we would like to thank you for your insightful comments. We did our best to address them in the updated draft which we believe improved our paper significantly. Please let us know if you have any additional comments. If you are satisfied with our responses, we would appreciate a score increase as we believe our “Salient Imagenet” dataset and our proposed methodology can be great resources for the community.
>
>
> [1] Understanding Failures of Deep Networks via Robust Feature Extraction. Sahil Singla, Besmira Nushi, Shital Shah, Ece Kamar and Eric Horvitz. CVPR2021.
>
> [2] Leveraging sparse linear layers for debugging deep networks. Eric Wong, Shibani Santurkar, Aleksander Madry. ICML2021.

---

> ### Author Response · Authors · 2021-11-19
> **Following up**
>
> Dear Reviewer DwGM,
>
> We appreciate your time reviewing our work and for your insightful comments. In our responses, we did our best to address all of your comments. Before the discussion period ends, please let us know if you have any additional comments. If you are satisfied with our responses, we would appreciate a score increase.

---

### Decision · Program_Chairs · 2022-01-20

**Decision:**

Accept (Poster)

**Comment:**

The paper tackles the important problem of spurious feature detection in deep neural networks. Specifically, it proposes a framework to identify core and spurious features by investigating the activation maps with human supervision. Then, it produces an annotated version of the ImageNet dataset with core and spurious features, called Salient ImageNet, which is then used to empirically assess the robustness of the method against spurious training signals in comparison with current SOTA models.

As pointed out by the reviewers, this work is not about causality and the definitions of causal and spurious features were originally vague and inaccurate. During the revision and discussion,  the authors changed the terms "causal" features/accuracy to "core" features/accuracy. They also called the provided dataset "Salient Imagenet", instead of "Causal Imagenet", and changed the title to "Salient ImageNet: How to Discover Spurious Features in Deep Learning?". Following the prior discussion, we strongly recommend the authors discard any discussion about causality in the camera-ready version of the paper to avoid confusion. Further, we encourage the authors to consider the reviewers’ thoughts and comments in preparing the camera-ready version of their manuscript.